

# Deep water formation in the North Atlantic Ocean in high resolution global coupled climate models

Torben Koenigk[1,2], Ramon Fuentes-Franco[1], Virna Meccia[3], Oliver Gutjahr[4], Laura C. Jackson[5], Adrian L. New[6], Pablo Ortega[7], Christopher Roberts[8], Malcolm Roberts[5], Thomas Arsouze[7], Doroteaciro Iovino[9], Marie-Pierre Moine[10], Dmitry V. Sein[11]

[1] Rossby Centre, Swedish Meteorological and Hydrological Institute, Norrköping, Sweden
[2] Bolin Centre for Climate Research, Stockholm University, Sweden
[3] Istituto di Scienze dell'Atmosfera e del Clima (CNR-ISAC), Bologna, Italy
[4] Max Planck Institute for Meteorology, Hamburg, Germany
[5] Met Office, Exeter EX1 3PB, U.K
[6] The National Oceanography Centre Southampton, U.K.
[7] Barcelona Supercomputing Center – Centro Nacional de Supercomputación (BSC), Barcelona, Spain
[8] European Centre for Medium-Range Weather Forecasts (ECMWF), Reading, U.K
[9] Fondazione Centro Euro-Mediterraneo sui Cambiamenti Climatici (CMCC), Bologna, Italy
[10] CERFACS/CNRS, Toulouse, France
[11] Alfred Wegener Institute for Polar and Marine Research, Bremerhaven, Germany

*Correspondence to*: Torben Koenigk (torben.koenigk@smhi.se)



**Abstract.** Simulations from seven global coupled climate models performed at high and standard resolution as part of the High Resolution Model Intercomparison Project (HighResMIP) have been analyzed to study the impact of horizontal resolution in both ocean and atmosphere on deep ocean convection in the North Atlantic and to evaluate the robustness of the
signal across models. The representation of convection varies strongly among models. Compared to observations from ARGO-floats, most models substantially overestimate deep water formation in the Labrador Sea. In the Greenland Sea, some models overestimate convection while others show too weak convection.

In most models, higher ocean resolution leads to increased deep convection in the Labrador Sea and reduced convection in the Greenland Sea. Increasing the atmospheric resolution has only little effect on the deep convection, except in two models,
which share the same atmospheric component and show reduced convection. Simulated convection in the Labrador Sea is largely governed by the release of heat from the ocean to the atmosphere. Higher resolution models show stronger surface heat fluxes than the standard resolution models in the convection areas, which promotes the stronger convection in the Labrador Sea. In the Greenland Sea, the connection between high resolution and ocean heat release to the atmosphere is less robust and there is more variation across models in the relation between surface heat fluxes and convection. Simulated
freshwater fluxes have less impact than surface heat fluxes on convection in both the Greenland and Labrador Sea and this result is insensitive to model resolution. is not robust across models. The mean strength of the Labrador Sea convection is important for the mean Atlantic Meridional Overturning Circulation (AMOC) and in around half of the models the variability of Labrador Sea convection is a significant contributor to the variability of the AMOC.

## 1 Introduction

The Atlantic Meridional Overturning Circulation (AMOC) is one important part of the global thermohaline circulation and transports heat in the upper ocean far to the north into the northern North Atlantic, and cold water masses to the south in the deep ocean.

Many studies have discussed the potential for a weakening and even collapse of the Atlantic Meridional Overturning Circulation (AMOC) as a response to global warming (Cheng et al., 2013; Swingedouw et al., 2007; Brodeau and Koenigk,
2016; Koenigk and Brodeau, 2017). Several recent studies further suggest that the AMOC has already reduced substantially compared to the preindustrial period. For example, Thornalley et al. (2018) used paleoclimatic reconstructions to show that the AMOC in the last 150 years was lower than at any other time in the last 1600 years. Caesar et al. (2018) used a fingerprint of the AMOC on sea surface temperature (SST) as an index for the AMOC and they found that this AMOC index showed the lowest values in the last few years since 1850. A reduction of the AMOC would have fundamental impacts on
the climate in the North Atlantic region but also in adjacent regions such as Western Europe (Manabe and Stouffer 1999), the Arctic (Mahajan et al., 2011, Koenigk et al., 2012) and even for the large scale atmospheric circulation (Jackson et al., 2015). Direct observations from the RAPID array indicate an ongoing weakening of the AMOC at 26.5°N (Smeed et al.



2014; Smeed et al. 2018). However, the observational time period is short and it is difficult to know whether or not this weakening is part of a long-term response to climate change or part of internal decadal variability (e.g. Jackson et al., 2016; Roberts et al., 2014; Robson et al. 2016).


Deep convection is a key oceanic process that ventilates the lower limb of the AMOC, and contributes to the storage of heat, anthropogenic carbon and oxygen in the deep ocean. Although the idea that deep convection is accompanied by large amounts of sinking water and that this deep water formation is the main mechanism for driving the AMOC has been questioned (Sayol et al., 2019), a reduction of deep wintertime convective mixing in the northern North Atlantic will likely

have important impacts for the AMOC. Without the deep mixing, less North Atlantic Deep Water (NADW) is formed (Dickson and Brown, 1994). Kuhlbrodt et al. (2007) and Medhaug and Furevik (2011) identified wind-driven upwelling, gyre circulation, and wind and tidal vertical mixing as important processes sustaining the long-term strength of the AMOC, and a potential collapse of deep water convection in the North Atlantic would not necessarily lead to a collapse of the AMOC (Marotzke and Scott, 1999; Kuhlbrodt et al., 2007; Gelderloos et al., 2012). On the other hand, surface buoyancy

forcing exerts a very strong control on exactly where and when the overturning occurs. Thus, even if mixing ultimately sets the strength of the global overturning, the deep water formation in the North Atlantic will play a key role in the strength of AMOC on decadal timescales. In many future model simulations, deep convection in the North Atlantic declines rapidly due to surface warming and freshening (Latif et al., 2006; Deshayes et al., 2007; Koenigk et al., 2007; Frankignoul et al., 2009) but the AMOC shows a comparatively smaller reduction of 5-40% depending on model and emission scenario (Cheng,

2013). However, these studies still show the importance of deep water formation for the amplitude and variability of the AMOC (Lozier et al. 2019).

The question whether and to what extent deep water formation in the Labrador Sea is a driver for the AMOC and its variability is still discussed. While many modelling studies (Jungclaus et al., 2005; Kuhlbrodt et al., 2007; Gelderloos et al., 2012; Eden and Willebrand, 2001; Biastoch et al., 2008; Brodeau and Koenigk, 2016; Ba et al., 2014; Roberts et al., 2013)

showed a high correlation between the Labrador Sea convection and the variability of the AMOC at different time scales, no conclusive observational evidence for a link between dense water formation in the Labrador Sea and AMOC variability has emerged to date (Lozier et al. 2017, Lozier et al. 2019). The short observation period, however, might make it difficult to draw robust conclusions on the link between Labrador Sea convection and AMOC since model simulations suggest that the AMOC lags the Labrador Sea convection by several years (Brodeau and Koenigk, 2016; Roberts et al. 2013).

The deep convection in the Labrador Sea is mainly driven by wintertime buoyancy loss to the atmosphere (Latif et al., 2006; Frankignoul et al., 2009). The buoyancy loss itself is strongly governed by the North Atlantic Oscillation (NAO); during a positive NAO, cold air is advected from the Arctic southward over the sea ice to the Labrador Sea and leading to large buoyancy loss to the atmosphere and consequently strong convective mixing in the Labrador Sea. Labrador Sea convection varies strongly on interannual to decadal time scales (Yashayaev and Loder, 2016). While the convection was rather shallow

since the early 1990s, the period 2012-2016 was one of the most persistent periods with Labrador Sea convective activity ever observed since 1938 (although observations were scarce over most of the 20[th] century), and the winter of 2016 showed a



widespread convective activity down to 2200m (Yashayaev and Loder, 2017). Irminger Sea convection was also strong in recent years and reached record levels in the winter 2014/ 2015 (de Jong and de Steur, 2016; de Jong et al., 2018). The transport of freshwater from the Fram Strait along the Greenland coast into the Labrador Sea is another contributing factor,
with increased freshwater fluxes leading to reduced salinities that suppress deep convection in the Labrador Sea (Holland et al., 2001; Jungclaus et al., 2005; Koenigk et al., 2006).

Deep convection in the Nordic Seas may also play an important role (Lozier et al., 2019). Langehaug et al. (2012) linked the variability of the AMOC to the variability of the overflows across the Greenland-Scotland Ridge.

Because of the coupled nature of the involved processes, coupled climate models are in principle well suited to study the
interactions between deep convection and other climate-related processes. However, Heuze (2017) stated that "the majority of CMIP5 models convect too deeply, over too large an area, too often and too far south". Further, Heuze (2017) found that deep convection is best simulated in those models with realistic ice edges in the North Atlantic.

In this study, we analyze the impact of increasing the horizontal resolution on the deep convection in the North Atlantic. We use simulations from seven models participating in the High-Resolution Model Intercomparison Project (HighResMIP,
Haarsma et al., 2016), which have been performed in the EU-H2020-project PRIMAVERA. High resolution has been shown to improve many aspects of the ocean circulation. Gutjahr et al. (2019) showed a reduction of temperature and salinity biases in the MPI-ESM1-2 model with eddy resolving ocean resolution. Grist et al. (2018) showed a more realistic northward ocean heat transport in high-resolution models that results consequently in a more realistic representation of the sea ice in the Atlantic sector of the Arctic Ocean (Docquier et al., 2019). High resolution ocean models also substantially improve the
position of the North Atlantic Current (Chassignet and Marshall, 2008; Sein et al., 2018). Furthermore, higher horizontal resolution might lead to a more realistic simulation of freshwater exports out of the Arctic (Fuentes Franco and Koenigk, 2019) and a better representation of the properties and position of the dense overflows.

After this introduction, we proceed with describing the models, the data and the methods in section 2. Sections 3 to 5 will show the results from this study and we will conclude in section 6.

## 2 Models, data and method

### 2.1 Models and simulations

In this study we analyze seven global coupled climate models (see e.g. Vannière et al., 2019), which participated in the HighResMIP experiment within the H2020-EU-project PRIMAVERA. These models are ECMWF-IFS (Roberts et al., 2018), HadGEM3-GC31 (Roberts et al., 2019), MPI-ESM1.2 (Gutjahr et al,. 2019), CMCC-CM2 (Cherchi et al., 2019),
CNRM-CM6.1 (Voldoire et al., 2019), AWI-CM-1.0 (Sidorenko et al., 2014, HR and LR setups: Sein et al., 2016) and EC-Earth3P (Haarsma et al., 2020). We use the historical coupled simulations from 1950-2014 and the 100-year control simulations (using constant 1950-forcing) from these seven models for our analysis. All models performed the simulations in at least two different resolutions following the HighResMIP-protocol. Changes in oceanic and atmospheric parameters are





kept to a minimum between low and high resolution simulations, so that all changes can be directly attributed to the change
in resolution.

The resolution varies among models. A few of the models vary both ocean and atmosphere resolution at the same time while
others separately changed ocean or atmosphere resolution. This allows us to analyze also the effect of increasing the
resolution in only one component of the system. Five of the seven models use the NEMO-ocean model as ocean component.
While this might limit the robustness of our conclusions across models, it has to be noted that the NEMO-model
configurations differ quite substantially from each other with different sea ice models (LIM2, LIM3, GELATO, CICE) and
differences in parameters (e.g. Gent McWilliams versus Smagorinsky). AWI-CM-1-0 and MPI-ESM1.2 use the same
atmosphere component but different ocean components.

More details on the models and the simulations used in this study are provided in table 1.

### 2.2 Observational data

To compare the mixed layer depth (MLD) from the models to observations, the typical variable used to represent ocean
convection depth, we use data from ARGO-profiles (Holte et al. 2017) provided on a 1° grid. In the ARGO-data, the de
Boyer Montégut et al. (2004) variable density threshold is used to calculate the mixed layer depth. Two different ARGO data
sets of the mixed layer depth are used in this study: first, the climatological mean MLD in each grid point in March in the
years 2000-2015; second, the maximum (mean over the two largest observed values in the period 2000-2015) MLD in each
grid-point. Note that in many grid-points only a few ARGO-profiles exist and in some no profiles at all. Further, ARGO-
floats generally sample to a depth of 2000m, thus MLD extending below 2000m are not captured. In addition, the
observational time period is short given the long time scales of variability in deep water formation. Thus, the ARGO-
observations do only provide an estimate of the observed MLD.

We use turbulent latent and sensible heat fluxes from the global ocean-surface heat flux products (1958-2006) developed by
the Objectively Analyzed air-sea Heat Fluxes project at the Woods Hole Oceanographic Institution (WHOI-OAFlux) to
evaluate the surface heat fluxes in our models. We use monthly means of the WHOI-OAFlux data on a 1° grid from 1958
onwards.

### 2.3 Method

Several different indices have been defined for the deep convection in the ocean (e.g. Schott et al., 2009; Yashayaev and
Loder, 2009; Lavergne et al., 2014; Koenigk et al., 2007; L'Heveder et al., 2012). These indices take into account either the
deepest reaching convection and/ or the horizontal extent of the MLD. However, none of them excludes convective events
that are too shallow to contribute to deep water formation. To overcome this problem, Brodeau and Koenigk (2016) defined
the so-called "Deep Mixed Volume" (DMV), which only considers the convective mixing below a specific depth (critical
depth $z_{crit}$) and integrates the volume of these deep mixed water masses in different convection regions of the North Atlantic.
In our study, we use the DMV index for monitoring the deep convection. Following Brodeau and Koenigk (2016), we use a





critical depth of 1000m for the Labrador Sea and 700m for the Greenland Sea. In the Labrador Sea, convection needs to reach a depth of around 1000m to be able to sustain the renewal of Labrador Sea water and eventually become North Atlantic Deep Water (Yashayaev, 2007). In the Nordic Seas, convection needs to at least reach down to the depth of the Denmark Strait and Faroe Bank Channel, which is around 600-700m in the models. We define the Labrador Sea region as
70° W - 40° W, 45° N - 72° N and the Greenland Sea region as 20° W - 20° E, 65° N - 82° N. The main areas of Labrador and Greenland Seas convection in all models fall into these regions. Although intermittent deep convection can occur in the Irminger Sea as well, we focus here only on the two regions with deepest convection.

We use monthly mean values of the March MLD of the model simulations to calculate the DMV. Note, that short convection episodes that exceed $z_{crit}$ might thus be missed.

We also calculate the DMV from the ARGO data as comparison to the model results. We infilled grid-points with missing data in the ARGO-data by interpolating the nearest neighbours. This and the short time series of the ARGO-data lead to uncertainties in the calculations of the DMV from ARGO and therefore it only provides a rough estimate for the real world. As an additional comparison, we also calculate the DMV based on critical depths of 0m, thus considering the full mixed layer.

For correlations, we calculate the Pearson correlation coefficient (r). We call a correlation significantly different from 0, if the p-value of the Pearson correlation coefficient is 0.05 or smaller based on a two-sided student-t distribution. Assuming 98 (N-2) degrees of freedom (assuming independence of each year of data in the 100-year (N=100) 1950-control simulations), the correlation is significant if |r| exceeds 0.2. When taking the autocorrelation of the variables into account, the degrees of freedom are reduced and differ depending on model and variable.

**3 Deep Convection in the North Atlantic**

This section analyzes first the MLD in March, the month with the strongest convection in both observations and models, in the North Atlantic in the different models and in ARGO. Then, we focus on the DMV in the models, its variability and potential trends in the historical simulations.

**3.1 Mixed layer depth**

Figure 1 shows the averaged March MLD from ARGO and from all historical model simulations. In the period 2000-2015, the ARGO data suggest average MLDs of about 1000 m in the Labrador and Greenland Seas. In the models, the MLD differs greatly and shows a strong dependence on the spatial resolution. While ECMWF-IFS-LR and EC-Earth3P show no or very shallow MLD in the main convection areas of the North Atlantic, many of the other simulations strongly overestimate the MLD compared to ARGO. Some models simulate much too strong convection in the Labrador Sea but do not show any deep
convection in the Greenland Sea while other models overestimate the MLD in both seas. In contrast, in the Irminger Sea, the MLD is more consistent across models and agrees better with ARGO. Note that we here compare models' MLD averaged

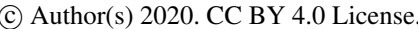



over 1950-2014 with ARGO-data from 2000-2015. As we will discuss later more in detail, some of the models show a weakening of the convection with time. Thus comparing the same time period in models and ARGO would slightly reduce the overestimation of MLD compared to ARGO in these models.


The MLD deepens with increasing ocean resolution in all models, except for AWI-CM-1-0. However, the models showing deepening MLDs are not fully independent, because they share NEMO as the ocean component, whereas AWI-CM-1-0 has FESOM as ocean component. On the other hand, even the global models with NEMO3.6 as ocean component (compare HadGEM3-GC31, CNRM-CM6.1, CMCC-CM2 and EC-Earth3P) differ considerably. This discrepancy suggests that either

the different atmospheric components or the choice of ocean parameters have a strong influence on the convection.

Differences in MLD between model versions where only the atmospheric resolution is increased are small compared to the effect of increased ocean resolution (compare ECMWF-IFS-MR and ECMWF-IFS-HR, HadGEM3-GC31-MM with HadGEM3-GC31-HM, CCCM-CM2-HR4 with CCCM-CM2-VHR4) except for the MPI-ESM1-2, where increased atmospheric resolution leads to reduced MLD. This can likely be linked to too weak wind forcing in MPI-ESM1-2-XR

(Putrasahan et al., 2019).

To investigate the impact of natural variability on the mean March MLD in the historical period and to verify the potential contribution of natural variations to the differences in MLD with changing resolution, we use an ensemble of historical simulations with the ECMWF-IFS model. The MLD in the low resolution version ECMWF-IFS-LR is very shallow in all 6

ensemble members and there is no deep convection in the historical and control simulations (not shown). Thus, we concentrate in the following on the four members of ECMWF-IFS-HR, which all exhibit pronounced deep mixing, particularly in the Labrador Sea. These four ECMWF-IFS-HR members indicate a considerable natural variability (Figure 2 a-d). The averaged March MLD (1950-2014) deviates in individual ensemble members up to about 200m from the ensemble mean MLD. Although, this is a considerable amount, given the relatively long averaging period, the MLD differences due to

increased resolution from 1° to a 0.25° in the NEMO-models are larger. Figure 2f shows the DMV in the Labrador Sea for the ensemble mean and the four ECMWF-IFS-HR members. There is substantial spread across members but no generally different behavior in amplitude and time-scales of variability and trends across model members can be seen.

Even though four members are not sufficient to capture the total natural variability, these results suggest that natural variability cannot explain the differences in MLD due to a change in spatial resolution.

**3.2 Deep Mixed Volume**

In the following, in order to consider the horizontal extension of convection patterns and discard shallow convection events that have limited impact on the oceanic circulation such as the AMOC, we will concentrate on the DMV index to investigate the deep convection in the Labrador and Greenland Seas in more detail.





### 3.2.1 Labrador Sea

Figure 3 shows the DMV in the Labrador Sea in March in the historical model simulations. In agreement with Figure 1, increasing the resolution from around 1° to 0.25° in the ocean generally leads to an increased DMV in all models using NEMO, while the opposite is true for AWI-CM-1-0. A further increase in ocean resolution to 1/12 ° in HadGEM3-GC31-HH does not further increase the DMV. The DMV varies strongly among models: ECMWF-IFS-LR does not show any deep convection events in the entire historical period, CNRM-CM6.1 and EC-Earth3P simulate only a few events with deep

convection and AWI-CM-1-0-LR and CMCC-CM2 simulate strong deep convection every winter.

Table 2 compares the average DMV in the historical model simulations with that of ARGO in the period 2000-2015. The only simulation that shows similar values in the Labrador Sea as ARGO is EC-Earth3P-HR. As discussed above, EC-Earth3P and ECMWF-IFS-LR show no or rather little deep convection in the Labrador Sea while the other simulations (except for CNRM-CM6.1) overestimate the ARGO-based DMV with factors of four to almost 40. Despite the uncertainties

in the ARGO data, it is clear that the models seem to have problems to realistically simulate the convection in the Labrador Sea. If deep convection occurs, the ocean is often mixed down to the bottom, while in-situ observations indicate that deep convection rarely exceeds 2000m (Yashayaev and Loder, 2016; Yashayaev and Loder, 2017).

If we use a critical depth of $z_{crit}$=0 m instead of 1000 m in the Labrador Sea and thus consider the total mixed layer depth, the relative deviation of the DMV in the models from ARGO is reduced as expected (not shown). However, AWI-CM-1-0-LR

and CMCC-CM2 still overestimate the DMV based on ARGO by a factor of three and two, respectively. On the other hand, ECMWF-IFS-LR simulates only 20% of the mixed volume compared to ARGO. The comparison between $z_{crit0}$ and $z_{crit1000}$ reveals also some non-linearites in the deep convection. While CNRM-CM6.1-HR has a nine times higher DMV ($z_{crit1000}$) compared to ARGO, it is only 16% higher for $z_{crit0}$, whereas the DMV ($z_{crit1000}$) for MPI-ESM1-2-XR is 4.6 times higher compared to ARGO but 14% smaller for $z_{crit0}$.


The interannual variability of the DMV is large in all models (Figure 3). Some of the simulations also indicate substantial variability at decadal or longer periods. Despite missing continuous long-term observations of convection in the Labrador Sea, in situ observations from different observational campaigns suggest that deep convection occurs only intermittently with large interannual to decadal variations (Lazier et al., 2002; Yashayaev and Loder, 2016). Such intermittent deep convection

is partly visible in the DMV-time series of EC-Earth3P, HadGEM3-GC31-LL, MPI-ESM1-2, CNRM-CM6.1 and ECMWF-IFS-MR. However, in most of the model simulations, deep convection occurs in almost every winter or not at all.

The strength of the deep convection in March is reflected in the vertical density distribution in the Labrador Sea. Naturally, the models with more frequent and deeper convection show a much weaker vertical stratification than the models that do not exhibit deep convection. More interesting as the density distribution during the convection period itself is the vertical

stratification at the beginning of the winter, which indicates the preconditioning of the ocean for convection events later in winter. Figure 4 shows the vertical density stratification of the upper 600m in the Labrador Sea. All models show a near



surface low density layer, mainly due to a combination of low surface salinity and relatively (compared to late winter) warm water near the surface in November. Generally, the models with lower ocean resolution show a stronger stratification in the upper ocean than models with higher resolution (except for AWI-CM-1-0). The two model simulations, which do not

simulate any deep convection, ECMWF-IFS-LR and EC-Earth3P, show particularly strong upper ocean density gradients. Consequently, a large buoyancy flux would be needed during winter until deep convection could set in in these two models. MPI-ESM1-2 and AWI-CM-1-0 show a more stratified upper ocean in November with increased atmospheric resolution. This agrees with a weaker convection in their higher resolution versions. The density profiles of the high ocean resolution models agree relatively well with the observed one from ARGO although the near surface low density layer is too shallow in

most of these models. This might contribute to the overestimation of the deep convection in late winter in these models (compare Figures 1 and 3) but is probably not the only reason as will be further discussed in section 4.

Twelve of 19 simulations indicate a significantly negative trend of the DMV in the historical period (Figure 3, Table 3). To investigate whether this trend is really due to external forcing and not to model drift due to the rather short spinup period, we

compared the DMV in the historical simulations with that from the 100-year 1950-control simulations (Figure 5 and Table 3). Most of the control simulations do not show any large trends and in 9 out of 17 historical simulations, the DMV trends in the historical simulations are significantly more negative compared to the first 65 years of the control simulations indicating that external forcing is the major cause for the DMV reduction. As found with the mean DMV, the negative trends are larger with higher ocean resolution in the historic simulations.

A reduction of DMV in the historical period would be in line with some recent studies by Caesar et al. (2018), Thornalley et al. (2018) and Brodeau and Koenigk (2016).
We calculated the power spectrum of the DMV in the Labrador Sea in order to investigate the predominant variability of the DMV in each simulation in more detail (Figure 6). For better comparison, we detrended and normalized (using the standard deviation of the normalized time series) the 100-year 1950-control simulations. The dominant time scale varies across

simulations and models. Many of the models with NEMO-ORCA025 as the ocean model (ECMWF-MR, HadGEM3-GC31-MM, HadGEM3-GC31-HM, CMCC-CM2-HR, EC-Earth3P-HR) and also MPI-ESM1-2-HR show a dominant peak in the spectrum at around 10 years. In ECMWFS-IFS-HR, HadGEM3-GC31-HH and MPI-ESM1-2-XR with further increased resolution in either ocean or atmosphere, the main peak in the spectrum seems to shift towards somewhat longer time periods. The AWI-CM-1-0-HR shows a very pronounced variation at a period of around 7 years. The lower ocean-resolution

models, AWI-CM-1-0-LR, HadGEM3-GC31-LL and ECMWF-IFS-LR (but very weak DMV), show less dominate peaks than their higher resolution versions. CNRM-CM6.1 shows peaks at similar periods as CNRM-CM6.1-HR but the amplitudes differ.



### 3.2.2 Greenland Sea

The DMV in the Greenland Sea shows also a large spread across models (Figure 7, Table 2). As for the Labrador Sea, AWI-CM-1-0-LR and the two CMCC-CM2 simulations show the strongest deep convection while ECMWF-IFS and EC-Earth3P simulate rather weak convection (note the different order of magnitude in the vertical scales of Figure 7). Only three out of seven models simulate Greenland Sea DMV's of a similar magnitude compared to ARGO observations (HadGEM3-GC31, MPI-ESM1-2, CNRM-CM6.1). EC-Earth3P and ECMWF-IFS strongly underestimate deep water formation in the
Greenland Sea while it is strongly overestimated by AWI-CM-1-0 and CMCC-CM2.

The resolution dependency of Greenland Sea DMV differs substantially from the Labrador Sea. The NEMO-models do not show any deepening of convection with increased ocean resolution. In contrast, the high resolution versions of ECMWF-IFS, EC-Earth3P and CNRM-CM6.1 (after 1980) show very shallow or no deep convection. Moving from HadGEM3-GC31-LL to HadGEM3-GC31-MM leads to substantially smaller DMV. However, when increasing the atmospheric resolution from
HadGEM3-GC31-MM to HadGEM3-GC31-HM, DMV increases, and increases further in HadGEM3-GC31-HH, where both ocean and atmosphere resolution is increased. Thus, unlike for the Labrador Sea, there is no robust relation between the resolution in ocean or atmosphere and DMV in the Greenland Sea in the global models with NEMO as ocean component. The two other models, MPI-ESM1-2 and AWI-CM-1-0 show, as for the Labrador Sea, a reduction of DMV with increased resolution.


The trends of DMV in the Greenland Sea in the models do not agree in the historical period (Figure 7, Table 3). While HadGEM3-GC31-MM, MPI-ESM1-2-XR and the CMCC-CM2 simulations show a significantly negative trend, HadGEM3-GC31-HM and EC-Earth3P show positive trends. This could be due to the competing effects that the global warming trend has on deep convection in the region, which might be represented differently in each model: on one hand, by reducing sea
ice and thus enabling a larger surface for deep convection, and on the other hand by releasing melting heat water fluxes and warming the ocean surface, both contributing to lighter surface conditions and therefore increasing density stratification. The high DMV in CNRM-CM6.1-HR before 1970, and the decrease thereafter occurs similarly in the first decades of the 1950-control-simulation indicating that this trend is caused by a model balance adjustment and not due to external forcings (Figure 8, Table. 3). Similarly, the strong negative trends in CMCC-CM2 can partly be explained by similar drifts in the control
simulations, although the reduction in the historical runs is significantly larger than in the control runs.

Some of the historical and control simulations show strong decadal or longer-term variations (Figure 9). HadGEM3-GC31-HM, MPI-ESM1-2-HR, AWI-CM-1-0-HR and EC-Earth3P-HR show dominant variability at periods of around 20-25 years; ECMWF-IFS-MR, HadGEM3-GC31-HH, CMCC-CM2 and CNRM-CM6-1 simulations at time scales of 10-15 years. In addition, some of the simulations indicate variability at time-scales below 8 years. Overall, there is no clear dependence of
the variability on the resolution.

Figure 10 summarizes the resolution dependency of the deep convection across all seven models. Each single model shows a clear dependence of the DMV in the Labrador Sea on the oceanic resolution. The differences across models are large, and as discussed before, even models using the same version of the NEMO as ocean model exhibit a wide range of solutions. Only

models with coarse ocean resolution fail to produce any deep convection or produce only very shallow convection. All models with an ocean resolution of 50 km or higher (except for EC-Earth3P-HR) overestimate the convection compared to ARGO.

Increasing the atmosphere resolution has a minor impact on the DMV in the Labrador Sea, except for MPI-ESM1.2 and AWI-CM-1-0, where DMV is reduced with increased resolution.

The resolution dependency of the DMV in the Greenland Sea in single models is smaller than in the Labrador Sea. However, all the models, except for CNRM-CM6-1, show a decreased DMV when increasing the resolution to around 0.25°. The response to increased atmosphere resolution is not robust across models.

## 4 The impact of heat and freshwater fluxes on the deep convection in the North Atlantic

Deep convection depends strongly on the buoyancy of the ocean surface layer in the convection regions - the heat loss to the
atmosphere and the influx of fresh water into the convection regions.

### 4.1 Surface heat fluxes

Brodeau and Koenigk (2016) showed that the turbulent surface heat flux (SHF) is the main driver for interannual variability in the DMV. Thus, in the following, we will mainly focus on the SHF.

Figure 11 shows the winter (January, February, March) SHF in each of the model simulations. The WHOI-OAFlux data
show the largest SHF up to more than 200 W/m$^2$ from the ice edge in the Labrador Sea extending to the southern part of the subpolar gyre, south of Iceland and along the southeast coast of Greenland, and in the northern Norwegian-Greenland Seas and Barents Sea. The large-scale features of this pattern are reproduced by most of the models. ECMWF-IFS-LR and to a lesser degree EC-Earth3P, both simulating too weak convection, strongly underestimate the SHF in the Labrador Sea. The high-resolution models show a better representation of the observed SHF pattern. In particularly, they represent more
realistically the extension of high SHF from the Labrador Sea into the southwestern branch of the sub-polar gyre and the high SHF in the northern Greenland and Norwegian Seas. A number of models, in particular both CMCC-CM2 versions, HadGEM3-GC31-HH and CNRM-CM6.1 overestimate the SHF in the sub-polar gyre. In addition, the SHF west and northwest of Scotland is too high in most of the models.

In the Labrador Sea, all high-resolution models with NEMO as the ocean component simulate increased SHF compared to
their lower-resolution counterparts (Table 2). In contrast, MPI-ESM1-2 shows a reduced SHF with increased atmospheric



resolution in line with the reduced convection. This dependence shown by most models agrees well with the resolution dependency of the DMV in the Labrador Sea. In all models, the interannual variations of winter SHF (averaged over the same box as used for calculation of the DMV) is significantly correlated with the DMV in March. The correlation coefficient varies from 0.48 in CNRM-CM6.1 to slightly above 0.7 in ECMWF-IFS-MR, EC-Earth3P and CMCC-CM2-HR4. The

relation between SHF and DMV is neither resolution nor model dependent.

The winter SHF in the Labrador Sea itself is governed by the atmospheric circulation (not shown). In all model simulations northerly to northwesterly winds, which advect cold air from the Arctic sea ice towards the Labrador Sea, lead to strong surface heat fluxes, which overcome the stratification of the ocean (Ortega et al. 2011), and increased convection. These north-to-northwesterly winds are further linked to a large scale atmospheric circulation pattern, which is similar to the

positive phase of the North Atlantic Oscillation (NAO). The NAO-index itself, which we define here as the difference of the normalized winter sea level pressure anomalies over the Azores and Iceland, is significantly correlated with the $DMVz_{crit1000}$ in the Labrador Sea in all simulations except for the low resolution simulations with EC-Earth3P and ECMWF-IFS. These are the simulations with no or only little deep convection and which have a strongly stratified ocean. The other model simulations show correlations between 0.38 (HadGEM-GC31-LL) and 0.67 (HadGEM-GC31-HM and CMCC-CM2-HR4).

The NAO is not only important for interannual variations of the DMV but also on the decadal scale. Correlations of 10-year running means of NAO and DMV reach between 0.3 and 0.57. A spectral analysis of the NAO resembles most of the peaks in the spectrum of the DMV (not shown) although the NAO shows relatively more energy at shorter time scales compared to the DMV in the Labrador Sea.

As in the Labrador Sea, northerly winds are the main cause for large oceanic surface heat loss to the atmosphere in the

Greenland Sea. The northerly winds are connected to low pressure anomalies over northern Scandinavia and the Barents Sea. The DMV in the Greenland Sea is correlated to the SHF as well. However, here we find a stronger model dependency of this relation. The correlation is weak to moderate in HadGEM3-GC31 (r=0.22 for LL; r=0.5 for HH) and in CNRM-CM6.1 (r=0.35; r=0.5 for HH) but high correlation is found for ECWMF-IFS (r=0.64 for HR; r=0.85 in LR) and EC-Earth3P (r=0.61; r=0.69 for HR). As for the Labrador Sea, the relation between SHF and DMV shows no clear resolution-

dependency.

### 4.2 Freshwater and sea ice exports

A number of studies have previously discussed the effect of Arctic freshwater export, especially through Fram Strait, as a potential source of variability of the deep water convection in the Labrador Sea (Holland et al., 2001; Jungclaus et al., 2005; Koenigk et al., 2006). Here, we analyze the correlation between freshwater transports across different sections (Fram Strait,

Denmark Strait, northern Baffin Bay) and deep convection in the Labrador Sea and in the Greenland Sea (transports through Fram Strait) in the historical simulations of the models. We consider both the vertical integrated liquid freshwater transports

and solid (sea ice) transports. Generally, the correlations between deep convection and both solid and liquid transports across all sections are relatively small although in some of the models significant. In all model simulations, the annual mean southward transport of both liquid and solid freshwater across Fram Strait and the liquid transport across Denmark Strait are

negatively correlated with the deep convection in the Labrador Sea in March. The correlation coefficients range between -0.1 and -0.4 and reach highest values when the freshwater transport through Fram Strait (and Denmark Strait) leads the convection by one to two years (zero to one year). Increased southward transport of sea ice and liquid freshwater transports through Fram Strait along Greenland's east coast and through Denmark Strait leads to more freshwater input into the Labrador Sea, which tends to reduce the convection. Figure 12 shows for the two model simulations with the highest

correlation between freshwater transport through Denmark Strait and DMV in the Labrador Sea (HadGEM-GC31-LL, EC-Earth3P-HR) that increased freshwater transport leads to a substantial reduction of the MLD in the Labrador region and thus contributes to the variability of the DMV. For most other model simulations, the effect is rather small compared to the impact of SHF-variability on the DMV.

In some models, the southward transport of liquid freshwater through Baffin Bay is positively correlated with the deep

convection in the Labrador Sea (up to r=0.35 in HadGEM3-GC31-LL). This may seem counterintuitive, but northerly winds in the Baffin Bay cause strong SHF in the Labrador Sea and dominate the convective conditions and simultaneously lead to increased fresh water transports to the south.

We do not find any resolution dependency of the correlation between freshwater exports and convection in the Labrador Sea. This result is in contrast to a recent study from Fuentes Franco and Koenigk (2019) where they analyzed a set of HadGEM3-

GC2 simulations at different resolutions and found larger correlations with increased resolution.

Overall, there is only a weak relationship between freshwater export through the Fram Strait and convection in the Greenland Sea, although it shows some dependency on the respective model (not shown). In some of the simulations, more freshwater export out of the Arctic is associated with reduced deep convection in the Greenland Sea, but in the majority of the simulations larger exports occur at the same time as increased convection. In the latter case, the increased convection is

driven by northerly winds, which at the same time increase the freshwater exports through Fram Strait.

## 5 The linkage of the DMV to the AMOC

The effect of high resolution on the AMOC in the HighResMIP model simulations has been studied in more detail in a parallel study to ours (Roberts et al. submitted). They found that "the AMOC tends to become stronger as model resolution is enhanced, particularly when the ocean resolution is increased from non-eddying to eddy-present and eddy-rich". Roberts et

al., (submitted) also analysed the relation between temporal mean values of the DMV and the average AMOC. As shown in section 3.2, only few models simulate a DMV that is consistent to observed estimates. However, these models underestimate



the AMOC (except for CNRM-CM6-1) compared to the RAPID-observations whereas some of the models (HadGEM3-GC31-MM and -HM, MPI-ESM1.2-HR, AWI-CM-1-0-LR) markedly overestimate the DMV in the Labrador Sea but simulate a realistic AMOC. Thus, the observations show a stronger AMOC with a lower DMV compared to the models, indicating other shortcomings in the representation of processes that govern the AMOC in the models.

There is in general a strong relationship between DMV in the Labrador Sea and the AMOC strength across models; models with more deep water production in the Labrador Sea have a stronger AMOC. Also for all single models (apart from AWI-CM-1-0), simulations with larger DMV are linked to a stronger AMOC. This relationship is less robust between DMV in the Greenland Sea and the AMOC as expected from the reduced DMV with increased resolution in most of the models (see sections 3.2.2).

To investigate the impact of variability in the deep water formation on the variability of the AMOC, we performed cross-correlation analyses between the DMV in Labrador and Greenland Seas and the AMOC (at 26°N) for lags between -/+ 10 years. In agreement with results by Brodeau and Koenigk (2016), annual values are only rather weakly correlated with each other. We thus focus here on correlations of linearly detrended and 10-year low pass filtered values of DMV and AMOC (Figures 13 and 14) in the 100-year 1950-control simulations. Positive lags mean that the AMOC leads DMV, negative lags mean that the DMV leads AMOC. In the case of Labrador Sea (Figure 13), maximum values of the normalized cross-correlations vary between around 0.3 (AWI-CM-1-0-HR, ECMWF-IFS-MR, CMCC-CM2-HR, EC-Earth3P-HR) and 0.8 (HadGEM3-GC31-LL, CMCC-CM6-1-VHR). Also CNRM-CM61-HR and MPI-ESM1-2-XR and HadGEM3-GC31-HM (HH) show high correlations. Most model simulations with high correlations reach their maximum correlation when the DMV leads the AMOC by 0-4 years.

The cross-correlations between DMV in the Greenland Sea and the AMOC index are generally positive for lags around year 0 but somewhat lower compared to the Labrador Sea. Relatively high values (exceeding 0.6) are obtained for the HadGEM3-GC31 model versions and by CNRM-CM6-1-HR. The time lag where correlations are highest differ among models. While in AWI-CM-1.0-models, CMCC-CM2-VHR and MPI-ESM1.2-HR, the DMV in the Greenland Sea leads the AMOC by a few years, in HadGEM-GC31 simulations and EC-Earth3P-HR, the AMOC leads the DMV, and in the other simulations no clear lead-lag relation can be identified.

## 6 Conclusions

We analyzed historical and 1950-control simulations in different resolution from seven global climate models following the HighResMIP protocol and investigated the impact of increasing the resolutions in ocean and atmosphere on deep convection in the North Atlantic Ocean.

The main results are summarized as follows:


- In general, global models mostly fail to simulate a realistic deep convection in the North Atlantic. This is critical since a realistic simulation of deep convection is important for the large scale ocean circulation, in particular the AMOC, the northward heat transport in the ocean and related impacts on the atmosphere. It also raises serious questions of the future

behaviour of the AMOC in climate models and its consequences for local and global climate.

- The ocean resolution clearly affects the deep water formation in the Labrador Sea. Convection activity enhances with increasing ocean resolution in four out of five models in this study. However, all these models use NEMO3.6 (although in somewhat different configurations) as their ocean component. It remains therefore unclear whether global models with other

ocean models respond differently to an increased resolution since the reduced convection in the fifth model (AWI-CM-1-0) results very likely from the simultaneously increased atmosphere resolution.

- Increasing the ocean resolution from 1° to 1/4° in the models with NEMO as the ocean component has a larger impact on the convection than increasing the atmosphere resolution in these models. In contrast, MPI-ESM1-2, in which only the

atmosphere resolution has been increased, and AWI-CM-1-0 (increased resolution in both atmosphere and ocean) show substantially reduced convection in the Labrador Sea at high resolution. Both models (AWI-CM-1-0, MPI-ESM1-2) use the same atmospheric component (ECHAM6.3) and the reduction of DMV with increased atmospheric resolution can likely be linked to reduced atmospheric winds in ECHAM6.3 in the high resolution version (Gutjahr et al., 2019; Putrasahan et al., 2019). The models with higher ocean resolution show more dominate variability at the decadal time scale in the Labrador

Sea compared to their lower resolution counterparts.

- In the Greenland Sea, increasing the ocean model resolution to around 1/4° reduces the convection in most models. Increasing the atmosphere resolution tends to reduce the convection but the result is not robust across models. Many models show dominant variability between 10 and 25 years but no clear dependence on the resolution could be found.


- The turbulent surface heat fluxes are strongly related to the deep convection in both the Labrador and Greenland Seas and seem to be more important for the variability of the DMV than freshwater exports out of the Arctic. In the Labrador Sea, we find that higher resolution leads to increased ocean heat release to the atmosphere in all the NEMO models but to reduced heat release in MPI-ESM1-2. This is in close agreement with the resolution dependency of the deep convection. Thus,

increased turbulent surface heat flux with high resolution is the main explanation for increased DMV in the Labrador Sea. The correlation between surface heat fluxes and DMV in the Labrador and Greenland Seas does not show any robust resolution-dependency.

- The 10-year low pass filtered DMV in the Labrador Sea is highly positively correlated (r=0.6-0.8) with the AMOC at 26 °N

in around half of the model simulations. In these simulations, the DMV leads the AMOC by a few years. In the other



simulations, the correlations are also positive but much lower (0.3-0.4) and time lags of the highest correlations are not robust across these simulations. The DMV in the Greenland Sea and the AMOC are only significantly correlated in few simulations and no clear lead/ lag relationship can be established. The correlations between DMV and AMOC are not dependent on the resolution.


- Increasing the resolution improves the vertical stratification of the upper ocean in late autumn but it does not generally improve the representation of the deep convection. In a few of the low-resolution models, the convection is overestimated compared to ARGO and this positive bias becomes even larger with higher resolution. However, the high resolution models have not been tuned and the main purpose of HighResMIP is to investigate the impact of increasing the resolution rather than

to improve existing biases.

**Acknowledgements**

This work has been funded by PRIMAVERA project, which is funded by the European Union's Horizon 2020 programme, Grant Agreement no. 641727PRIMAVERA.

The global ocean heat flux and evaporation products were provided by the WHOI OAFlux project (http://oaflux.whoi.edu) funded by the NOAA Climate Observations and Monitoring (COM) program.

**Data:** The data are stored on the Jasmin infrastructure, http://www.ceda.ac.uk/projects/jasmin/. The simulations are part of the High Resolution Model Intercomparison project (HiResMIP) and will be uploaded to the ESGF: https://esgf-

node.llnl.gov

**Scripts** for analyzing the data will be available from the corresponding authors upon reasonable request.

**Authors' contributions:** TK is first author of the manuscript and performed the largest part of the analysis, RFF calculated the freshwater exports out of the Arctic, VM calculated the AMOC and contributed with the analysis between DMV and

AMOC. OG, LJ, AN, PO, CR, MR, TA, DI, MM and DS contributed with discussions on the processes governing the DMV and linking DMV and AMOC as well as with performing the model simulations.



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





| Model | Ocean Model resolution | Atmosphere Model resolution | hist runs 1950-2014 | 100-yr ctrl-1950 |
|---|---|---|---|---|
| | NEMO3.4/ LIM2 | IFS cycle 43r1 | | |
| ECMWF-IFS-LR | ORCA1 - 1° | 50 km | 6 | 1 |
| ECMWF-IFS-MR | ORCA025 – 1/4° | 50 km | 1 | 1 |
| ECMWF-IFS-HR | ORCA025 – 1/4° | 25 km | 4 | 1 |
| | NEMO3.6/ CICE5.1 | UM | | |
| HadGEM3-GC31-LL | ORCA1 - 1° | 130 km | 1 | 1 |
| HadGEM3-GC31-MM | ORCA025 – 1/4° | 60 km | 1 | 1 |
| HadGEM3-GC31-HM | ORCA025 - 1/4° | 25 km | 1 | 1 |
| HadGEM3-GC31-HH | ORCA12 - 1/12° | 25 km | 1 | 1 |
| | MPIOM1.6.3 | ECHAM6.3 | | |
| MPI-ESM1-2-HR | TP04 - 0.4° | T127 | 1 | 1 |
| MPI-ESM1-2-XR | TP04 – 0.4° | T255 | 1 | 1 |
| | NEMO3.6/CIC4.0 | CAM4 | | |
| CMCC-CM2-HR4 | ORCA025 -1/4° | 100 km | 1 | 1 |
| CMCC-CM2-VHR4 | ORCA025 - 1/4° | 25 km | 1 | 1 |
| | NEMO3.6/GELATO | ARPEGE6.3 | | |
| CNRM-CM6-1 | ORCA1 - 1° | T127 | 1 | 1 |
| CNRM-CM6-1-HR | ORCA025 - 1/4° | T359 | 1 | 1 |
| | FESOM | ECHAM6.3 | | |
| AWI-CM-1-0-LR | 50 km | T63 | 1 (-2010) | 1 |
| AWI-CM-1-0-HR | 25 km | T127 | 1 (-2010) | 1 |
| | NEMO3.6/LIM3 | IFS cycle 36r4 | | |
| EC-Earth3P | ORCA1 - 1° | T255 | 1 | 1 |
| EC-Earth3P-HR | ORCA025 - 1/4° | T511 | 1 | 1 |

**Table 1: Overview on the model configurations and the simulations used in this study.**







| Normalized DMV, SHF | $DMV_{norm}$ Lab-Sea | $SHF_{norm}$ Lab-Sea | $DMV_{norm}$ Green-Sea | $SHF_{norm}$ Green-Sea | Corr SHF-DMV Lab-Sea | Corr SHF-DMV Green-Sea |
|---|---|---|---|---|---|---|
| ARGO/ WHOI Observations absolute values | $3.95e+13$ $m^3$ | $129.2$ W/m$^2$ | $6.5e+13$ $m^3$ | $125.7$ W/m$^2$ | not enough data | not enough data |
| ECMWF-IFS-LR ECMWF-IFS-MR ECMWF-IFS-HR | 0 - 0.03 8.9 9.9 - 11.7 | 0.40-0.65 1.15 1.17-1.26 | 0-0.008 0.0002 0-0.002 | 0.45-0.68 0.93 0.83-0.87 | 0.62 0.71 0.59 | 0.85 0.65 0.64 |
| HadGEM3-GC31-LL HadGEM3-GC31-MM HadGEM3-GC31-HM HadGEM3-GC31-HH | 4.3 17.1 19.6 17.8 | 0.98 1.28 1.39 1.48 | 4.0 0.6 1.7 6.5 | 1.22 1.05 1.05 1.12 | 0.63 0.70 0.64 0.59 | 0.22 0.48 0.50 0.36 |
| CMCC-CM2-HR4 CMCC-CM2-VHR4 | 24.4 24.8 | 1.22 1.34 | 13.0 15.0 | 1.22 1.14 | 0.72 0.59 | 0.52 0.58 |
| CNRM-CM6.1 CNRM-CM6.1-HR | 1.09 9.3 | 1.15 1.18 | 1.10 2.47 | 1.07 1.35 | 0.53 0.48 | 0.45 0.35 |
| MPI-ESM1-2-HR MPI-ESM1-2-XR | 10.6 4.6 | 1.14 0.98 | 1.2 0.6 | 1.25 1.16 | 0.61 0.64 | 0.44 0.64 |
| AWI-C-1-0-LR AWI-CM-1-0-HR | 39.5 12.8 | no data | 20.9 6.1 | no data | no data | no data |
| EC-Earth3P EC-Earth3P-HR | 0.26 0.95 | 0.63 1.07 | 0.24 0.004 | 1.02 0.79 | 0.72 0.50 | 0.69 0.61 |

**Table 2: Observed and modeled DMV and SHF in the Labrador and Greenland Seas, their ratio and their correlations. Row 2: DMV and SHF in observations. Rows 3-9: Ratio of modeled and observed DMV and SHF (Model values divided through observational values). Columns 6, 7: Correlation between SHF and DMV in the respective boxes of the Labrador and Greenland Sea. For the correlations $z_{crit0}$ has been used to avoid complications with periods without any deep convection; the correlations based on $z_{crit1000}$ and $z_{crit700}$ in LAB and GIN Seas are**
**generally similar for models with deep water formation in every winter but much lower in the models with no or very few deep convection events (ECMWF-LR, EC-Earth3P).**







| Model | Trend/year historical 1950-2014 DMV-Labrador | Trend/year control-1950 year 1-65 DMV-Labrador | Trend/year historical 1950-2014 DMV-Greenland | Trend/year control-1950 year 1-65 DMV-Greenland |
|---|---|---|---|---|
| **ECMWF-IFS-LR** | 2.54e+09 | -1.42e+10 | 1.11e+10 | 0.5e+10 |
| **ECMWF-IFS-MR** | ***-3.93e+12*** | -5.37e+11 | -5.35e+08 | 4.83e+09 |
| **ECMWF-IFS-HR** | ***-3.40e+12*** | -4.26e+11 | 1.90e+09 | 1.34e+10 |
| **HadGEM3-GC31-LL** | 4.09e+11 | 1.58e+12 | *6.62e+12* | *5.59e+12* |
| **HadGEM3-GC31-MM** | ***-4.43e+12*** | -1.11e+12 | ***-1.49e+12*** | 9.73e+11 |
| **HadGEM3-GC31-HM** | *-5.13e+12* | *-3.02e+12* | ***4.40e+12*** | -3.28e+11 |
| **HadGEM3-GC31-HH** | ***-6.66e+12*** | *-2.28e+12* | 3.10e+11 | -9.49e+11 |
| **MPI-ESM1-2-HR** | -1.41e+12 | -6.15e+11 | *-8.17e+11* | *-8.53e+11* |
| **MPI-ESM1-2-XR** | ***-7.94e+12*** | -6.02e+10 | ***-4.06e+11*** | *2.32e+11* |
| **CMCC-CM2-HR4** | ***-5.19e+12*** | -1.08e+12 | ***-9.31e+12*** | *-3.97e+12* |
| **CMCC-CM2-VHR4** | -1.42e+12 | *-3.27e+12* | ***-1.20e+13*** | *-2.69e+12* |
| **CNRM-CM6.1** | ***-5.38e+11*** | 1.62e+11 | *8.61e+11* | *4.51e+11* |
| **CNRM-CM6.1-HR** | ***-6.91e+12*** | -3.06e+11 | *-1.01e+13* | *-1.05e+13* |
| **AWI-CM-1-0-LR** | *-1.50e+13* | *-1.68e+13* | -1.97e+12 | 6.80e+11 |
| **AWI-CM-1-0-HR** | -9.56e+11 | -4.98e+12 | 1.21e+12 | *-2.50e+12* |
| **EC-Earth3P** | 1.69e+11 | 0 | ***4.25e+11*** | *3.05e+10* |
| **EC-Earth3P-HR** | **-3.87e+11** | *9.22e+11* | -1.39e+09 | 3.04e+09 |

**Table 3: Trends in the DMV in the Labrador and Greenland Seas in the historical simulations and in the first 65 years of the 1950-control simulations. Trends that are significantly different from 0 at the 95%-confidence level are shown in italic, trends significantly different to the control-runs are bold, and trends significantly different to both 0 and the control-run are italic and bold.**





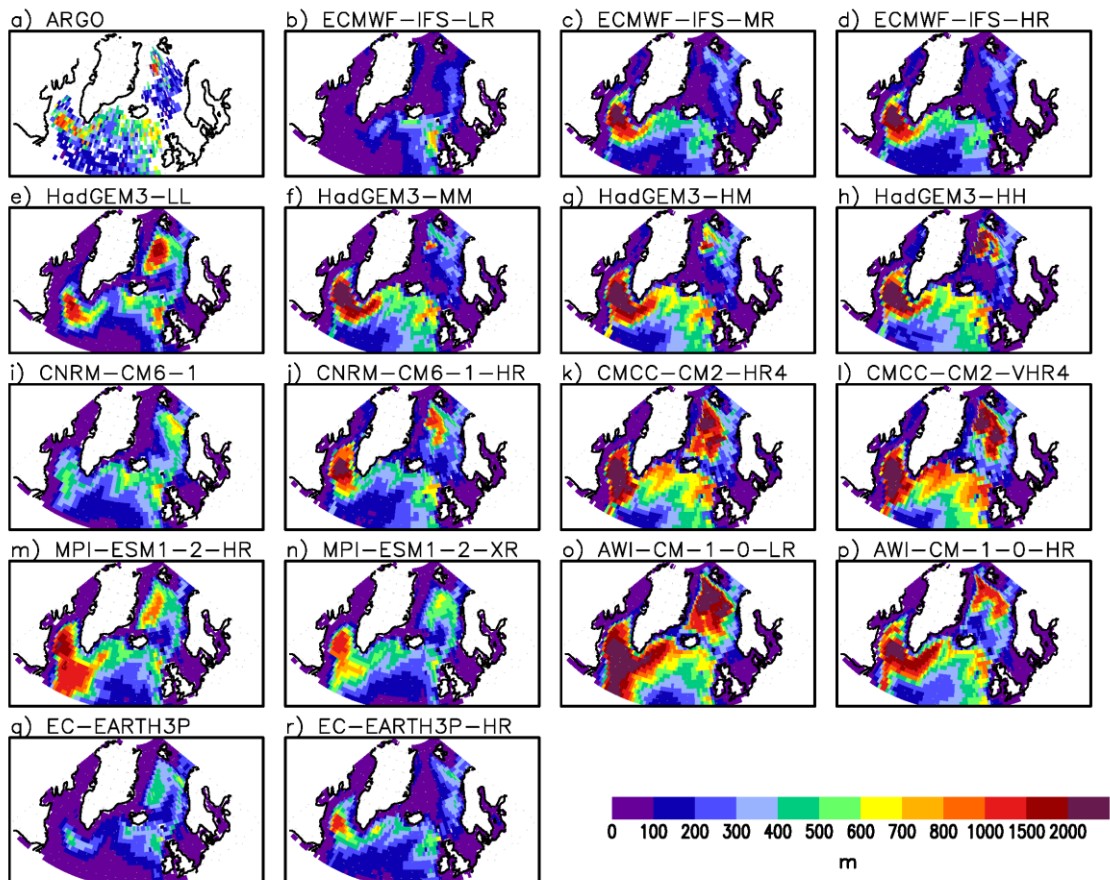

**Figure 1: Mixed layer depth in March in the ARGO-data, averaged over 2000-2015 (a) and in the historical low and high-resolution model simulations, averaged over 1950-2014 (b-r).**








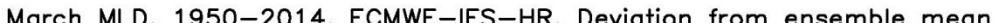

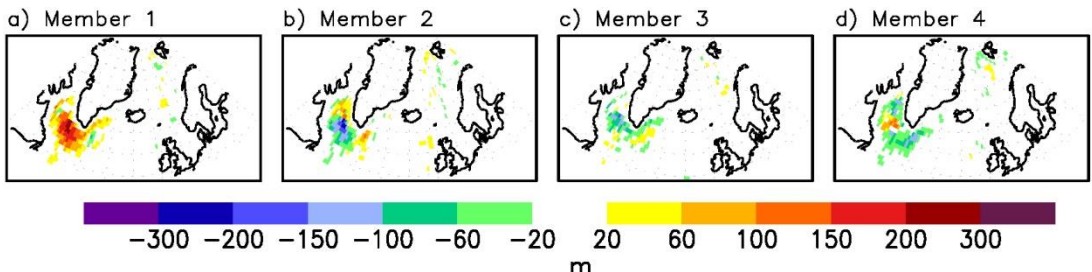

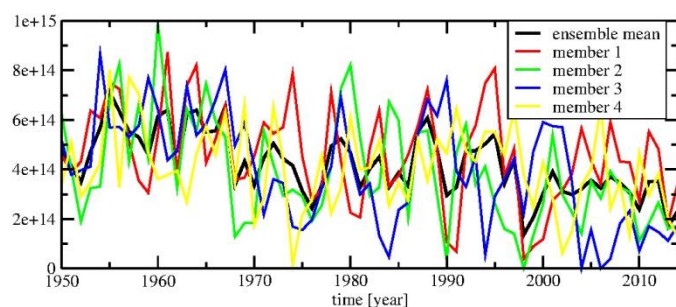

**Figure 2: a-d) Deviation of mixed layer depth in March in the ensemble members of ECMWF-IFS-HR from the ensemble mean of the four ECMWF-IFS-HR simulations for the time period 1950-2014. e) DMV in the Labrador Sea in the ensemble mean and single ensemble members of ECMWF-IFS-HR.**




**Figure 3: Deep Mixed Volume (DMV) in m³ using a critical depth of 1000 m in the Labrador Sea in March between**
**1950-2014. Note the different y-axis between models. For ECMWF-IFS, only member 1 is shown for better visual**
**comparison of the variability across resolutions.**





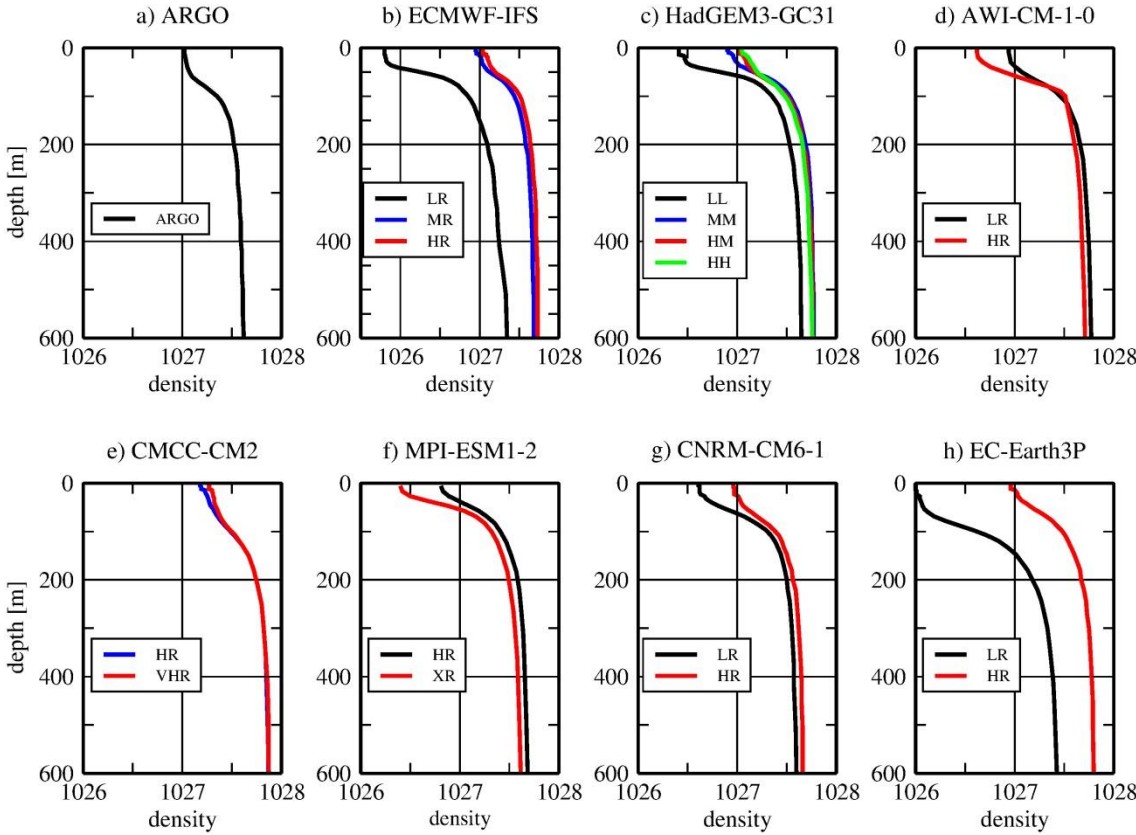


**Figure 4: Density (in kg/m$^3$) in the upper 600 m averaged over the Labrador Sea in November. Note that the scale of the x-axis differs in b) to capture the low densities of ECMWF-IFS-LR near the surface.**








**Figure 5: As Figure 3 but for the 100-year control simulation.**








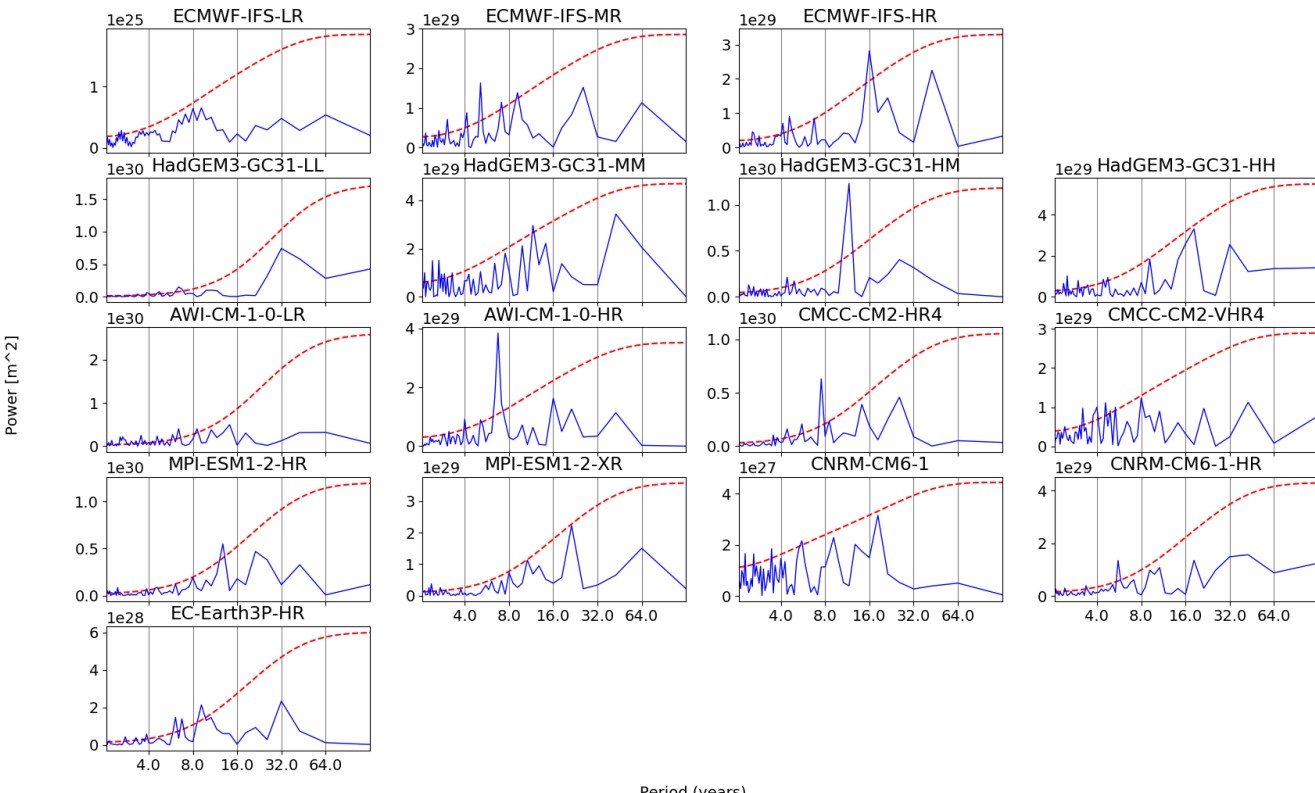

**Figure 6: Power spectrum of detrended and normalized March DMV time series of the 100-year control simulation in the Labrador Sea. The dashed red line shows the 95% significance level. Note, that no deep convection occurred in the 100-year period in EC-Earth3P.**






**Figure 7: As Figure 3 but for the Greenland Sea and a critical depth of 700 m.**







**Figure 8: As Figure 7 but for the 100-year 1950-control simulations.**







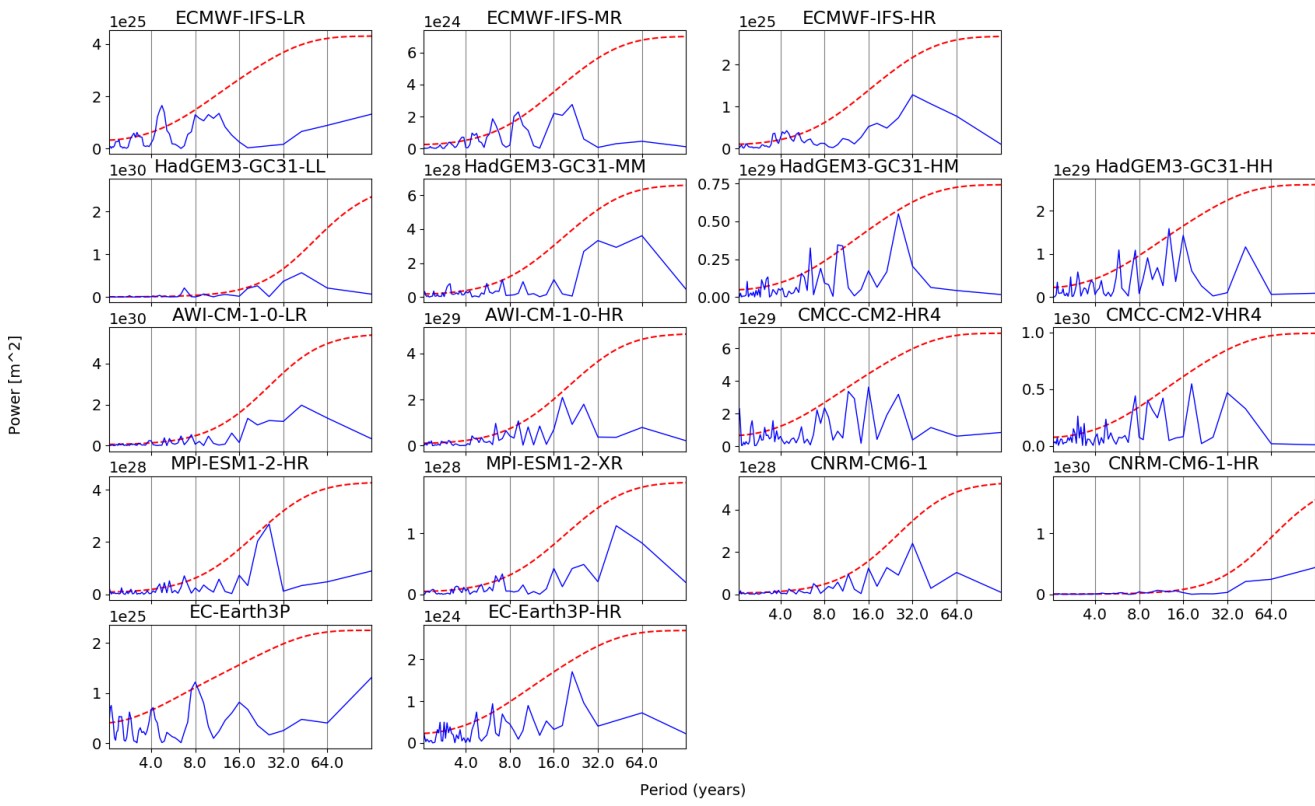

**Figure 9: Same as Figure 6 but for DMV in the Greenland Sea.**





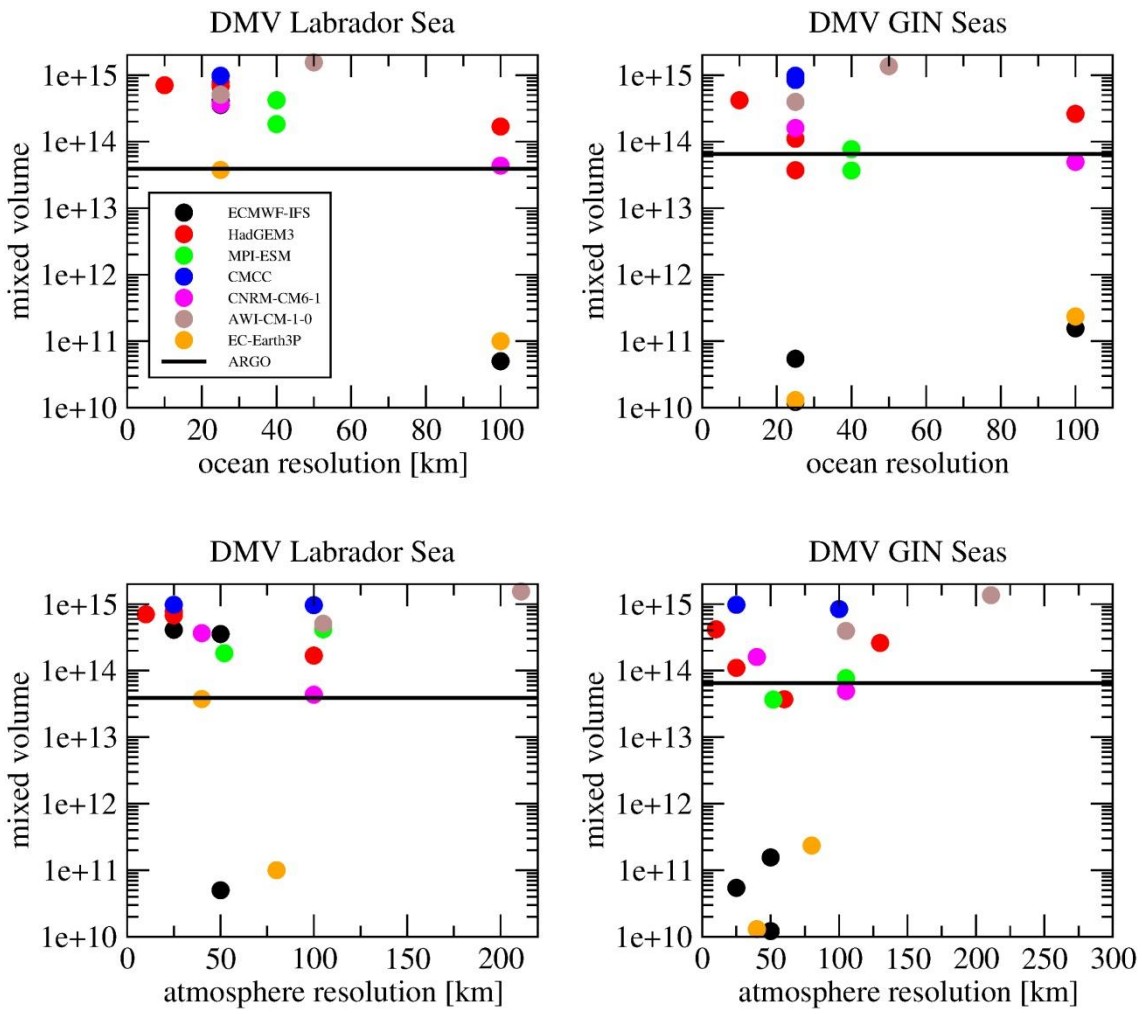

**Figure 10: DMV (in m$^3$) in Labrador Sea (left) and GIN-Seas (right) in March in dependence on the oceanic (top) and atmospheric (bottom) resolution. Average over 1950-2014.**





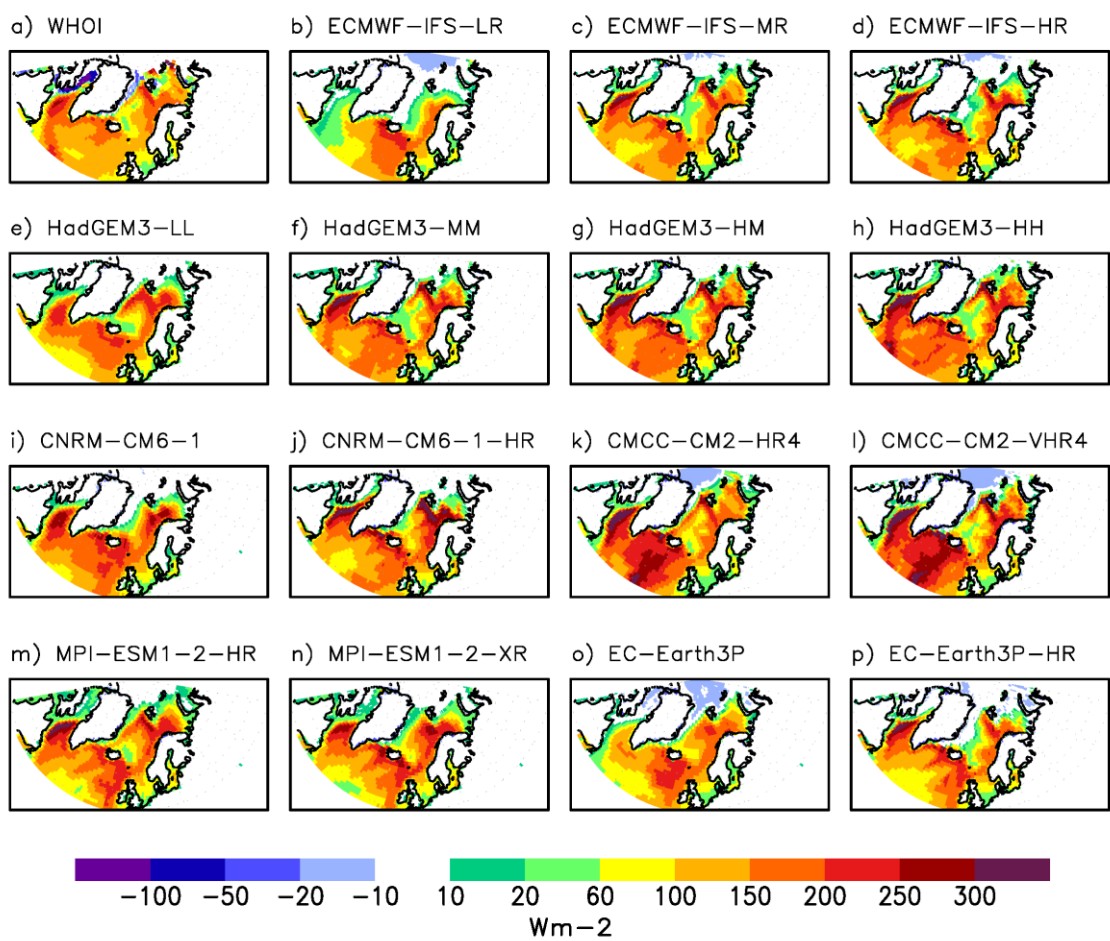

**Figure 11: Turbulent surface heat flux (January, February, March average) in 1950-2014 in the WHOI-OAFlux data and in the model simulations.**








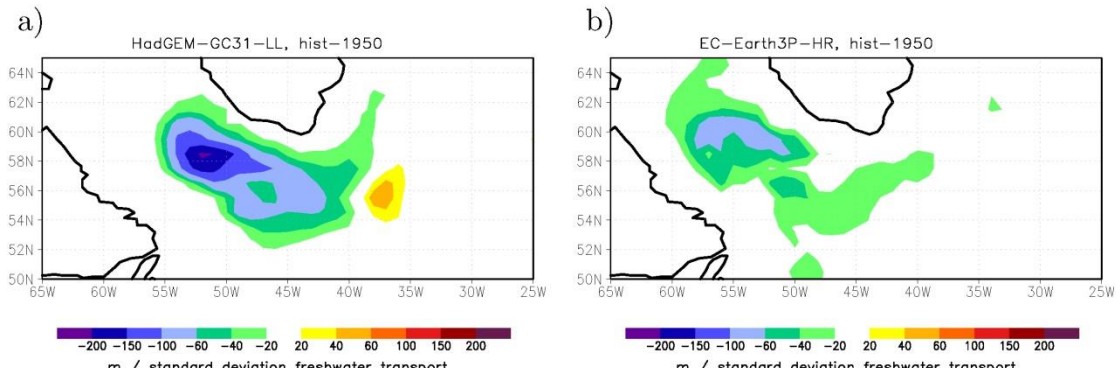

**Figure 12: Regression between annual mean freshwater transport through the Denmark Strait and mixed layer depth in the following March. a) HadGEM-CG31-LL and b) EC-Earth3P-HR. Data have been detrended before calculating** 900 **the regression. These two simulations show the largest correlation between Denmark Strait freshwater transport and DMV in the Labrador Sea (-0.4 and -0.35 for HadGEM-GC31-LL and EC-Earth3P-HR, respectively).**



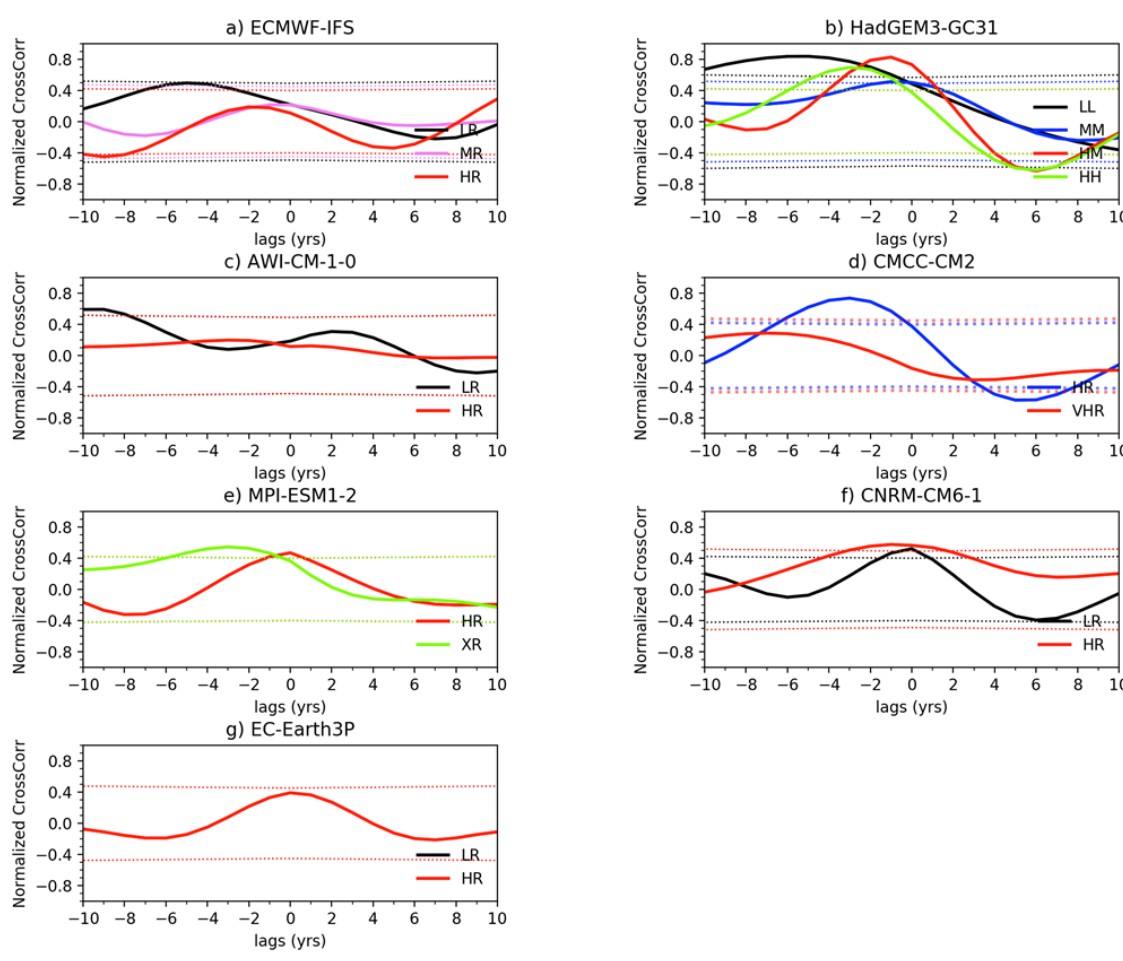


**Figure 13: Crosscorrelation between the DMV using a critical depth of 1000 m in the Labrador Sea in March and the AMOC index for the 100-year control simulation. Both timeseries were detrended and filtered with a 10 years low-pass filter. Area enclosed by dotted lines represents the 95% confidence calculated as 2/sqrt(N), where N is the number of independent data based on the time that takes autocorrelation to fall below 1/e. Positive lags mean AMOC**

**leads DMV, negative lags mean DMV leads AMOC. The low resolution version of EC-Earth3P does not produce any deep convection events in the control simulation.**





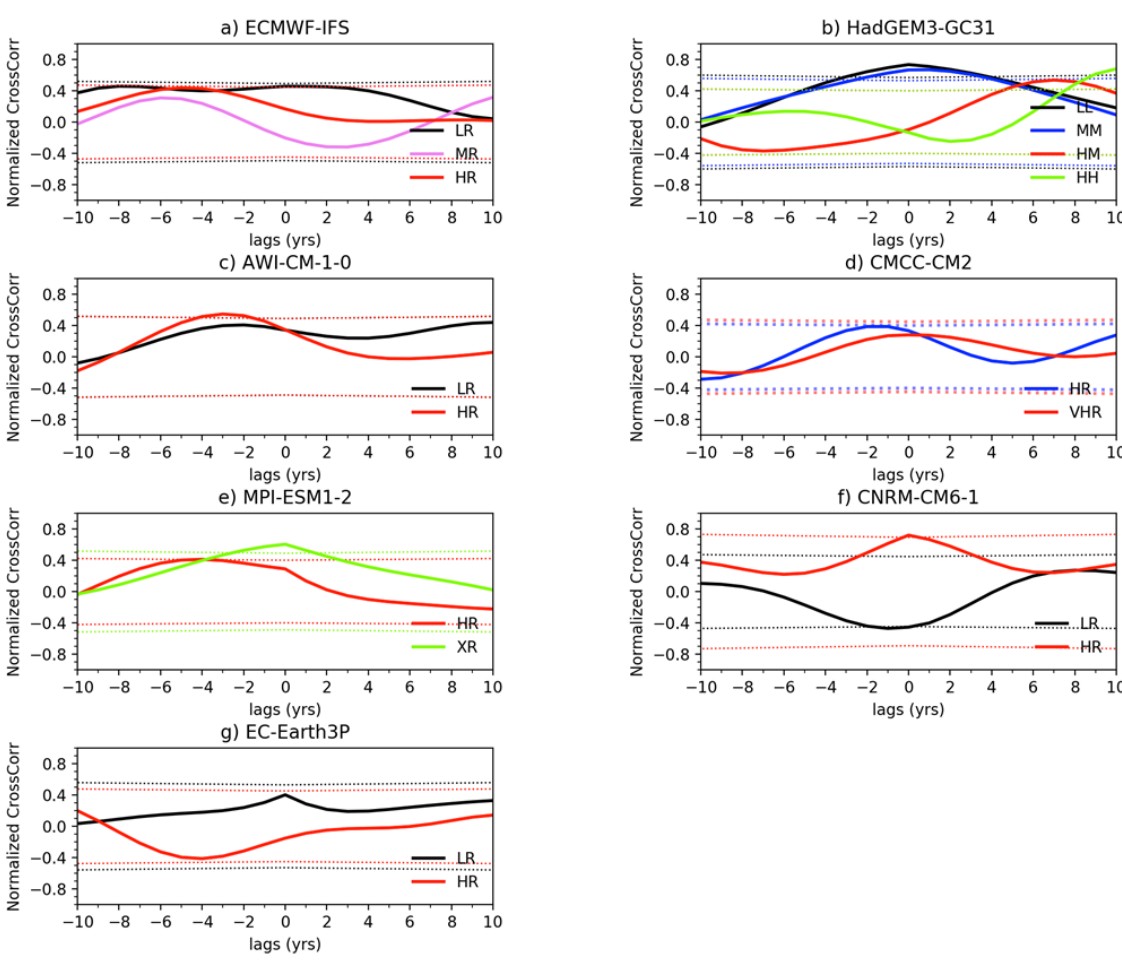

Figure 14: As Figure 12 but for the Greenland Sea and a critical depth of 700 m.