# Peer review of "Deep water formation in the North Atlantic Ocean in high resolution global coupled climate models"

_Ocean Science, 2020_

## Referee Comment (RC1) · Céline Heuzé (Referee) · 16 Jun 2020

Céline Heuzé (Referee)

celine.heuze@gu.se

The aim of this paper is to investigate the effect of increasing the ocean and/or atmosphere resolution on the modelled deep convection in the North Atlantic. Let's start with the major comments:

1) The results shown are far less clear than described.

The authors work with 7 models. The abstract indicates that for "most models, higher ocean resolution leads to increased deep convection". In the text (line 230), this already is reduced to "generally", NEMO models only (5/7). But tables and figures show that it is not the case for EC Earth or CMCC, so we are down to 3/7. ... I suggest a less exciting but more honest rephrasing of the results section, which clearly explains that

the results are model-dependent. And which investigates what in the ocean component (or in the atmosphere component, as you later suggest it is all set by the wind) causes it.

2) Section 4 is not robust, and its methods are not detailed.

Section 4.2 treats of the potential relationship between horizontal freshwater fluxes and deep convection. Nowhere are the freshwater calculations presented. All that we know comes from line 386, that the liquid component is the "vertical integrated liquid freshwater export". How did you do that? Was it on each model's grid, or interpolated onto a regular section? What is your reference salinity? Did you take the same for all models (e.g. use the same ref. salinity as in past literature), or have one per model (e.g. each model's mean deep salinity in the GIN seas)?

Moreover, you show no data / result for most of the section. At least, add the mean or min/max transport values to Table 2. Same for section 4.1: the SHF discussion is based on data you show, but not the NAO/wind one. Show it! Show where the NAO lies in these models (centres are most likely shifted depending on the resolution, see literature).

3) How is accuracy defined?

In section 3, your comparison observation - models was hard to follow as you never defined what an accurate model should be. For example, line 237, you write "the only simulation that shows similar values in the Labrador Sea as ARGO is EC-Earth3P-HR". ARGO is at 3.95; EC-Earth HR, at 0.95. But HadGEM-LL and MPI XR are at 4.3 and 4.6 respectively, and CNRM CM6.1 is at 1.09. What is wrong with these, that made you reject them in favour of EC Earth?! Same again line 298: ARGO is at 6.5 and AWI HR at 6.1, but this model is not mentioned. It feels like an extra criterion is required for you to accept a model as accurate, but you never specified it, so the reader is left with their confusion.

**4) Other methodological, rather major comments**

You purposely exclude the Irminger Sea, even though a lot of models convect here instead of the Labrador Sea. Why? And why not investigating whether deep convection shifts from the Irminger to the Labrador Sea depending on the resolution?

Line 200, you admit that comparing the models to the ARGO period would be better, but you do not. For the frequency analysis, I agree that you should not. But at least in table 2, you should.

Section 5 is introduced as having already been done in Roberts et al. (subm). So, why having section 5 at all? What is different from Roberts et al.?

|| To finish, here come some more minor comments, by order of appearance:

The introduction is mostly about the AMOC and its relationship to deep convection. But the AMOC is not the topic of this paper (apart from section 5, see above). Please modify the introduction, focussing on deep convection and its importance for the ocean and the climate in general, including the AMOC sure, but not only.

Throughout the text, you use Greenland Sea when you mean GIN or Nordic Seas. Please correct.

The figures really need to be improved. The most crucial ones are the line plots, especially Fig. 3, where black and blue, or magenta and red, or red and green, are on the same panel, with the same line style. First, as the lines are supposed to represent an increase in resolution, what about some colours that are more intuitive? With e.g. LR in blue, moving to green, then yellow (with black contours), then orange, and finishing at XR with red. Then, to help the reader distinguish the lines, vary their styles. Again, just an example, make every other line dashed, and/or vary their thicknesses.

Figs 6 and 9, as the aim is resolution comparison, give the same y-axis to all panels. And since in the text you comment on the "peak around 10 years", use a log10 x-axis instead of a log2.

Fig 11, the increase of SHF with resolution is not visible as the colour scales are saturated. This information can be retrieved from table 2, so up to you whether to also improve Fig 11.

Line 260 (and Fig 4), which density are you using?

Section 4.1, clarify whether SHF > 0 means heat lost or gained by the ocean (from the figures and results, I assume it's lost by the ocean).

Lines 377-378: spell out what the relationship means in practice, i.e. that larger DMV is associated with larger heat fluxes out of the ocean. Furthermore, comment on lagged relationship (calculate it if needed) to see which comes first.

— end —

---

## Referee Comment (RC2) · Anonymous Referee #2 · 8 Jul 2020

This paper represents important material for further development of global climate model. The authors do a decent job in getting an overview of the different models and their performance. My comments are mainy regarding the formulations and figures, which could be improved.

Main comments:

Introduction: in many places, the style is a fast shift between presenting 'settled knowledge' and presenting the moerations. This leads to very long sentences, which are difficult to read. For instance: L 72-74 (and how is it questioned?) and l. 75-79. The whole section must be revised according to this to become more clearly structured.

The introduction, or may be the models and simulations section, must discuss 'high

resolution' in relation to the Rossby radius and resolving eddies.

Power spectra calculated in Figs 6 and 9 and referred to in the text. As far as I can see, these 'spectra' are raw periodograms. They will have 100 independent points (for a 100 year simulation), and therefore, five points will exceed the 95% significance curve just by chance. So periodogram spectra don't tell much. There are ways to overcome this, see e.g. von Storch and Zwiers (1999). The spectral analysis must be improved along these lines. Also, it must be specified, how the red significance level is calculated. The spectra with red lines does not seem to describe the background spectra of the model data very well; this looks odd, and must be explained. Conclusions in text must be changed according to revised spectral analysis.

Minor comments:

L 1-2 (title): The paper is comparing what could be called (present-day) standard resolution with higher resolution. The title should reflect this. L. 82: '..future model simulations ..', write e.g: '.. model simulations of future climate...' L. 87: 'The question whether ...'. Why not write: 'It is still discussed ...' L. 107: '..important role.' For what? L. 110: 'Climate-related processes' Be more specific, please. L. 113: 'increasinng the hjorizontal resolution' .. of GCMs. L. 117: Here you refer to a study with an eddy-resolving model. It should be made clear, that the present paper is not about eddy-resolving models. L. 134: An outline of the HighResMip-protocol and a reference are needed here. L. 154-158: Some more description of these fluxes is needed. L. 161: I cannot find a detailed description on how MLD is calculated in observations and in models. This must be added. L. 220: Where do you see that? L. 231: Well , I se an around 50/50 split. L. 259-261: I don't understand this sentence. l. 274: What is the spinup period? L. 315: 'melting heat water fluxes'. What is that? L. 349 'high-resolution models'. In ocean or in atmosphere? L. 419: What does this sentence mean? L. 443: add '... non eddy-permitting...'

Table 3. Add columns with histroical trend – control trend.

Fig. 2: Dont you yellow lines. They are really difficult to see.

Add a Fig. 2 $\frac{1}{2}$ showing MLD for models + ARGO (analogous to Fig. 3)

Fig. 3: Labels like 5e+14 a really not nice to look at, please change them. Why is there only two lines in panel a). And please add a panel based on ARGO data.

Fig. 4: Instead of having a separate panel (a) for ARGO, put the argo stratification in the model-panels.

Fig. 10: Colors for HadGEM3 and CNRM-CM6 are indistinguishable, please change. Also the figure is hard to understand. May be points referring to the same model in different resolution can be connected with thin lines. You must experiment, and improve the figure.

---

## Author Comment (AC1) · 5 Aug 2020

Thank you for your time and the constructive comments. Below, we will respond to all comments. All changes we suggest to include in a revised manuscript (including changes in figures and tables) are highlighted in the "SuggestedChanges_Manus-Tables-Figures_4August2020.pdf" (see supplement). Although, we are aware that we are not supposed to include a revised manuscript in this reply, we found it more practical for both us as authors and you as reviewer to highlight the suggested improvements into the manuscript itself rather than attaching all figures and tables separately.

––––––––––––

The aim of this paper is to investigate the effect of increasing the ocean and/or atmo-

sphere resolution on the modelled deep convection in the North Atlantic. Let's start with the major comments: 1) The results shown are far less clear than described. The authors work with 7 models. The abstract indicates that for "most models, higher ocean resolution leads to increased deep convection". In the text (line 230), this already is reduced to "generally", NEMO models only (5/7). But tables and figures show that it is not the case for EC Earth or CMCC, so we are down to 3/7. ... I suggest a less exciting but more honest rephrasing of the results section, which clearly explains that the results are model-dependent. And which investigates what in the ocean component (or in the atmosphere component, as you later suggest it is all set by the wind) causes it.

Response: We modified the abstract to make clear that not all 7 models increase the ocean resolution but only 5 of them do (ECMWF, HadGEM, CNRM, EC-Earth, AWI). The other two (MPI-ESM, CMCC) only increase the atmospheric resolution (see Table 1). 4 out of these 5 models where the ocean resolution is increased show a clear increase of Labrador Sea convection with increased ocean resolution (even EC-Earth, see Figures 1 and 3 and table 2). The only model where ocean resolution does not lead to increased deep convection in the Labrador Sea is thus the AWI-model and here - as discussed at several places in the manuscript - this can be linked to a (unrealistic) strong reduction in the wind stress in the high resolution version. The MPI-model uses the same atmosphere and suffers from the same problem in the high-resolution model; thus the impact of increased atmosphere resolution in MPI on convection is very likely linked to this. In the CMCC model, increased atmosphere resolution has hardly any effect on the convection in the Labrador Sea. Thus, we find for all NEMO-models, increased deep convection in the Labrador Sea with increased ocean resolution. The effect of atmosphere resolution seems to be smaller or is dominated by an unrealistic decrease of wind forcing (AWI, MPI). The question remains if this is only a NEMO-feature. The relation between increased ocean resolution and reduced deep convection in the GIN Sea is not as robust. We modified this part of the sentence in the abstract to make this clear.
* * *
2) Section 4 is not robust, and its methods are not detailed. Section 4.2 treats of the potential relationship between horizontal freshwater fluxes and deep convection. Nowhere are the freshwater calculations presented. All that we know comes from line 386, that the liquid component is the "vertical integrated liquid freshwater export". How did you do that? Was it on each model's grid, or interpolated onto a regular section? What is your reference salinity? Did you take the same for all models (e.g. use the same ref. salinity as in past literature), or have one per model (e.g. each model's mean deep salinity in the GIN seas)? Moreover, you show no data / result for most of the section. At least, add the mean or min/max transport values to Table 2. Same for section 4.1: the SHF discussion is based on data you show, but not the NAO/wind one. Show it! Show where the NAO lies in these models (centres are most likely shifted depending on the resolution, see literature).

Response: To calculate the liquid freshwater transport, we used the model grid lines on the native grids of the models that are closest to the geographical landmarks that define Fram Strait (across 78°N), Northern Baffin Bay (78°N) and Denmark Strait (66°N). The freshwater has been defined as the amount of zero-salinity water required to reach the observed salinity of a seawater sample starting from a reference salinity. Specifically, liquid freshwater transport (fwt, in m3/s) is estimated as

fwt=∫ _p1p2 ∫ _Dη âŰŠãŰ((S-Sref)/Sref)ãŰ dzdx

(please find a better view of the formula in the supplement) for salinity S (in practical salinity units). Reference salinity Sref was considered as 34.80 psu. The integration along z is performed from the bottom at depth D to the sea surface at height $\eta$ (in this case $\eta$=0). p1 and p2 are the landmarks and the integration was done considering dx as the length (or depth for dz) between every grid point. The solid freshwater transport is calculated from the sea ice transports across the sections assuming a constant ice salinity of 5 psu. We added these explanations to the manuscript.

We added a new Table 4 showing the freshwater exports through Fram Strait, Baffin Bay and Denmark Strait and discuss the results in section 4.

We added a figure showing the NAO-winter pattern in all the models (new Figure 12): We performed a Pearson correlation between geopotential height at 500 hPa and North Atlantic Oscilation (NAO) index during winter (JFM mean) for ERA5 and PRIMAVERA models. The periods used, were 1979-2019 for ERA5 and 1950-2014 for PRIMAVERA models. The NAO index was considered as the leading EOF of geopotential height on the 500 hPa pressure surface over the European/Atlantic sector (80W-40E, 20-90N). While the pattern varies across models and across different resolutions of the same model, no clear effect of the resolution on the NAO-pattern can be found.

_______________-

3) How is accuracy defined? In section 3, your comparison observation - models was hard to follow as you never defined what an accurate model should be. For example, line 237, you write "the only simulation that shows similar values in the Labrador Sea as ARGO is EC-Earth3P-HR". ARGO is at 3.95; EC-Earth HR, at 0.95. But HadGEM-LL and MPI XR are at 4.3 and 4.6 respectively, and CNRM CM6.1 is at 1.09. What is wrong with these, that made you reject them in favour of EC Earth?! Same again line 298: ARGO is at 6.5 and AWI HR at 6.1, but this model is not mentioned. It feels like an extra criterion is required for you to accept a model as accurate, but you never specified it, so the reader is left with their confusion.

Response: We clarified the table description to make clearer that only the row with the observations (row 2) of table 2 shows absolute values, while the other rows show the ratio between modeled and observed values. A value of 1 would mean that the modelled DMV (or SHF) is the same as the observed one, 0.5 that it is half as large and 2 that it is twice as large. Thus, a value of 0.95 for EC-Earth means that the EC-Earth value is close to the ARGO value while a value of e.g. 6.1 for AWI-HR means that the DMV is 6.1 times as large as the ARGO value. Thus, comparing these two models,

EC-Earth is much closer to ARGO than AWI. This hopefully clarifies the confusion with the model-observation comparison in section 3. We fully agree that it is difficult to clearly formulate how large a model is allowed to deviate from ARGO and still can be called "realistic". It is very subjective to formulate a criteria for "accurate" and given the fact that the DMV results from ARGO are rather uncertain as well (see discussion in the manuscript), we prefer to not formulate such criteria. We thus replaced the statements that a model is well reproducing the observations by stating more specific how much the different models deviate from the observations.
* * *
4) Other methodological, rather major comments You purposely exclude the Irminger Sea, even though a lot of models convect here instead of the Labrador Sea. Why? And why not investigating whether deep convection shifts from the Irminger to the Labrador Sea depending on the resolution?

Response: Figure 1 shows that none of the models used in this study convect in the Irminger Sea instead of the Labrador Sea. Only the ECMWF-LR model shows the largest mixed layer depth in the Irminger Sea but this convection is not very strong either, and thus not compensating in any way for the missing convection in the Labrador Sea in ECMWF-LR. Some models have the largest convection not directly in the Labrador Sea but shifted to the southeast to the south of Greenland. However, the Labrador Sea box is chosen such that it also covers convections south of Greenland and not only directly in the Labrador Sea. Figure 1 shows that there is no clear shift of convection centres from Irminger Sea to Labrador Sea or vice versa with increased resolution. We focus on deep convection here, defined as convection potentially contributing to the AMOC, and the convection in the Irminger Sea can if it all only for short periods be classified as deep convection.
* * *
Line 200, you admit that comparing the models to the ARGO period would be better,

but you do not. For the frequency analysis, I agree that you should not. But at least in table 2, you should.

Response: We added a column to table 2 where we compare the period 2000-2014 of the modelled DMV to ARGO and added discussions of these results to the manuscript. For the Labrador Sea, generally smaller DMV values occur in 2000-2014 compared to 1950-2015 as expected and thus somewhat reduced overestimations for most models. The two models, which fit best to ARGO using 1950-2014 as comparison, underestimate ARGO in 2000-2014. For GIN Seas, both increases and decreases of the DMV in 2000-2014 compared to the entire period occur. For some simulations these differences are substantial. This is also due to the large natural variability of the DMV with decadal or longer periods with strong or weak convection activities. Since we can not assume that models are in the same phase of natural variability as the observed period 2000-2015, it is not clear if comparing only this period in the models is really reducing the uncertainty in the comparison (which was the reason that we initially only compared the entire 65 years).
* * *
Section 5 is introduced as having already been done in Roberts et al. (subm). So, why having section 5 at all? What is different from Roberts et al.?

Response: Roberts et al. have analysed only mean values from the historical runs by showing scatterplots of mean AMOC against mean DMV. Here, we analyse the variability by showing the cross-correlations of the AMOC against DMV in the control runs to isolate the unforced internal variability. Therefore, both studies are different and complementary. We clarify this issue in the manuscript by changing L415-427 of the original manuscript from: to "Roberts et al., (submitted) also analysed the relation between temporal mean values of the DMV and the average AMOC in the historical runs. They found that there is a strong relationship between DMV and the AMOC strength across models; models with more deep water production in the Labrador Sea have a

stronger AMOC. Also for all single models (apart from AWI- CM-1-0), simulations with larger DMV are linked to a stronger AMOC. This relationship is less robust between DMV in the Greenland Sea and the AMOC as expected from the reduced DMV with increased resolution in most of the models (see sections 3.2.2). To investigate the impact of variability in the deep water formation on the variability of the AMOC, we performed cross-correlation analyses between the DMV in Labrador and Greenland Seas and the AMOC... "

———————————————— To finish, here come some more minor comments, by order of appearance:

The introduction is mostly about the AMOC and its relationship to deep convection. But the AMOC is not the topic of this paper (apart from section 5, see above). Please modify the introduction, focussing on deep convection and its importance for the ocean and the climate in general, including the AMOC sure, but not only.

Response: We rewrote almost the entire introduction to better structure it and focus more on the deep convection itself. We first introduce deep water formation, focusing then on it is importance for water masses, followed by the effects on local and remote climate. This is followed by a longer section focusing on the AMOC and potential linkages between deep convection and AMOC before we shortly describe the aim of the study and discuss potential effects of high resolution.

———————————

Throughout the text, you use Greenland Sea when you mean GIN or Nordic Seas. Please correct.

Response: We changed it to GIN Seas.

———————————

The figures really need to be improved. The most crucial ones are the line plots, especially Fig. 3, where black and blue, or magenta and red, or red and green, are on the

same panel, with the same line style. First, as the lines are supposed to represent an increase in resolution, what about some colours that are more intuitive? With e.g. LR in blue, moving to green, then yellow (with black contours), then orange, and finishing at XR with red. Then, to help the reader distinguish the lines, vary their styles. Again, just an example, make every other line dashed, and/or vary their thicknesses.

Response: Since model grids are different and ocean and atmosphere resolution increase differently between different models, it is difficult to find a color scale, which shows the resolution of each model configuration across all models. We decided therefore to change the colors in all the time-series and line figures as follows: Blue (solid line) for the standard resolution for each single model. Red (solid line) for the high-resolution version of each model. For models with intermediate resolution versions (ECMWF, HadGEM), we added dashed green and orange lines.

————————————————

Figs 6 and 9, as the aim is resolution comparison, give the same y-axis to all panels. And since in the text you comment on the "peak around 10 years", use a log10 x-axis instead of a log2.

Response: We changed the x-axis in Figures 6 and 9 following your suggestions. We tested to use the same y-axis as well, however, prefer to keep different y-axis for the different models since the power varies strongly among models and using the same y-axis makes it very hard to see anything on the plots. However, we changed it to log-scale as well.

————————————————

Fig 11, the increase of SHF with resolution is not visible as the colour scales are saturated. This information can be retrieved from table 2, so up to you whether to also improve Fig 11. Response: We changed the color scale in Figure 11 to make it better visible.

———————————

Line 260 (and Fig 4), which density are you using?

Response: We used the density of standard sea water (i.e. at 1 atm), as given by Millero and Poisson (1981): Frank J.Millero, AlainPoisson 1981: International one-atmosphere equation of state of seawater. Deep Sea Research Part A. Oceanographic Research Papers 28 (6), 625-629, https://doi.org/10.1016/0198-0149(81)90122-9 We added this information to the text.

———————————

Section 4.1, clarify whether SHF > 0 means heat lost or gained by the ocean (from the figures and results, I assume it's lost by the ocean).

Response: It is a loss for the ocean, we clarified this in the text and in Figure 11.

———————————————

Lines 377-378: spell out what the relationship means in practice, i.e. that larger DMV is associated with larger heat fluxes out of the ocean. Furthermore, comment on lagged relationship (calculate it if needed) to see which comes first.

Response: Large ocean heat losses in the winter are linked to strong DMVs in the following March, indicating that large upward surface heat fluxes lead the DMV. We clarified this in the text and commented on lag/ lead relationship.

Please also note the supplement to this comment:
https://os.copernicus.org/preprints/os-2020-41/os-2020-41-AC1-supplement.pdf

———————————————————

[Figure]

**Supplement:**

**The effect of increasing the horizontal resolution on the Deep water formation in the North Atlantic Ocean in HighResMIP models**

Torben Koenigk[1,2], Ramon Fuentes-Franco[1], Virna L. Meccia[3], Oliver Gutjahr[4], Laura C. Jackson[5], Adrian L. New[6], Pablo Ortega[7], Christopher Roberts[8], Malcolm Roberts[5], Thomas Arsouze[7], Doroteaciro Iovino[9], Marie-Pierre Moine[10], Dmitry V. Sein[11]

[1] Rossby Centre, Swedish Meteorological and Hydrological Institute, Norrköping, Sweden
[2] Bolin Centre for Climate Research, Stockholm University, Sweden
[3] Istituto di Scienze dell'Atmosfera e del Clima (CNR-ISAC), Bologna, Italy
[4] Max Planck Institute for Meteorology, Hamburg, Germany
[5] Met Office, Exeter EX1 3PB, U.K
[6] The National Oceanography Centre Southampton, U.K.
[7] Barcelona Supercomputing Center – Centro Nacional de Supercomputación (BSC), Barcelona, Spain
[8] European Centre for Medium-Range Weather Forecasts (ECMWF), Reading, U.K
[9] Fondazione Centro Euro-Mediterraneo sui Cambiamenti Climatici (CMCC), Bologna, Italy
[10] CERFACS/CNRS, Toulouse, France
[11] Alfred Wegener Institute for Polar and Marine Research, Bremerhaven, Germany

*Correspondence to*: Torben Koenigk (torben.koenigk@smhi.se)

**Abstract.** Simulations from seven global coupled climate models performed at high and standard resolution as part of the High Resolution Model Intercomparison Project (HighResMIP) have been analyzed to study the impact of horizontal resolution in both ocean and atmosphere on deep ocean convection in the North Atlantic and to evaluate the robustness of the signal across models. The representation of convection varies strongly among models. Compared to observations from ARGO-floats, most models substantially overestimate deep water formation in the Labrador Sea. In the Greenland-Norwegian-Iceland (GIN) Seas, some models overestimate convection while others show too weak convection.

In four out of five  models with increased ocean resolution, higher ocean resolution leads to increased deep convection in the Labrador Sea. The effect of resolution on convection in the GIN Seas is less clear. Increasing the atmospheric resolution has a smaller effect on the deep convection than increasing the ocean resolution. Simulated convection in the Labrador Sea is largely governed by the release of heat from the ocean to the atmosphere. Higher resolution models show stronger surface heat fluxes than the standard resolution models in the convection areas, which promotes the stronger convection in the Labrador Sea. In the GIN Seas, the connection between high resolution and ocean heat release to the atmosphere is less robust and there is more variation across models in the relation between surface heat fluxes and convection. Simulated freshwater fluxes have less impact than surface heat fluxes on convection in both the GIN and Labrador Seas and this result is insensitive to model resolution. The mean strength of the Labrador Sea convection is important for the mean Atlantic Meridional Overturning Circulation (AMOC) and in around half of the models the variability of Labrador Sea convection is a significant contributor to the variability of the AMOC.

**1 Introduction**

Open-ocean deep convection is a rare phenomenon, occurring only at a few locations in the world's ocean. It provides a vertical link between properties of the surface ocean and the deep ocean. In the North Atlantic, the northward flowing warm water masses become denser because of large heat loss to the atmosphere and sink into the deep ocean. Deep convection ventilates the deep ocean with oxygen and plays an important role for the storage of carbon and heat.

The main convection sites in the North Atlantic are the Labrador and Irminger Seas as well as the Greenland-Iceland-Norwegian (GIN) Seas. Deep convection in the Labrador and Irminger Seas produces the deep water masses called Labrador Sea Water (LSW) (Clarke and Gascard, 1983), which together with the dense overflow waters coming through the Faroe Bank Channel and Denmark Strait (Dickson and Brown, 1994) form the North Atlantic Deep Water.

The deep convection in the Labrador Sea is mainly driven by wintertime buoyancy loss to the atmosphere (Latif et al., 2006; Frankignoul et al., 2009), which is strongly governed by the North Atlantic Oscillation (NAO). Also freshwater transports through Fram Strait contribute to the variability of deep convection in the Labrador Sea (Holland et al., 2001; Jungclaus et al., 2005; Koenigk et al., 2006). Labrador Sea convection varies strongly on interannual to decadal time scales (Yashayaev and Loder, 2016) and was rather shallow in the 1990s and 2000s, but recovered in recent years with mixing depths exceeding 2000m (Yashayaev and Loder, 2017). Convection in the GIN Seas occurred frequently down to the bottom until the 1980s. However, thereafter, no regularly occurring deep convection has been observed any more; between 1994 and 2002 mixing depths of 700-1600m have been monitored (Ronski and Budeus, 2005). Irminger Sea convection is generally weaker than in the Labrador and GIN Seas but in recent years mixing depth reached below 1000m depth (de Jong and de Steur, 2016; de Jong et al., 2018).

Variability and change of deep convection affects local and remote climate. Reduced convection in the Labrador Sea is linked to surface salinity, temperature and sea ice in Labrador Sea and Davis Strait (Deser et al. 2002) but has also remote effects on the atmospheric circulation (Koenigk et al. 2006). The deep convection has for long time been seen as the driving mechanism for the Atlantic Meridional Overturning Circulation (AMOC), which is fundamentally important for the climate in the North Atlantic and adjacent regions such as Western Europe and the Arctic (Manabe and Stouffer 1999; Mahajan et al., 2011, Koenigk et al., 2012; Jackson et al., 2015). However, more recently, the importance of deep convection for the AMOC has been questioned (Sayol et al., 2019). Kuhlbrodt et al. (2007) and Medhaug and Furevik (2011) identified wind-driven upwelling, gyre circulation, and wind and tidal vertical mixing as important processes sustaining the long-term strength of the AMOC, thus, a collapse of the deep convection would not necessarily lead to collapse of the AMOC (Gelderloos et al. 2012; Marotzke and Scott 1999). On the other hand, surface buoyancy forcing exerts a very strong control on exactly where and when the overturning occurs. Thus, even if mixing ultimately sets the strength of the global overturning, the deep water formation in the North Atlantic will play a key role in the strength of AMOC on decadal timescales.

Many studies have discussed the potential for a weakening AMOC as a response to global warming (Cheng et al., 2013; Swingedouw et al., 2007; Brodeau and Koenigk, 2016; Koenigk and Brodeau, 2017) and linked the weakening to a reduction of the deep water formation (Latif et al., 2006; Deshayes et al., 2007; Koenigk et al., 2007; Frankignoul et al., 2009; Langehaug et al. 2012). Recent studies indicate an ongoing weakening of the AMOC at 26.5°N (Smeed et al. 2014; Smeed et al. 2018; Thornalley et al., 2018; Caesar et al., 2018) although it is unclear if this observed weakening is caused by climate change or internal decadal variability (e.g. Jackson et al., 2016; Roberts et al., 2014; Robson et al. 2016). Observational evidence for a link between dense water formation in the Labrador Sea and AMOC is still missing (Lozier et al. 2017, Lozier et al. 2019) but short observation period and potential lags of several years between AMOC and convection (Brodeau and Koenigk, 2016; Roberts et al. 2013) make robust conclusions from observations difficult.

The deep convective process is temporally intermittent and spatially compact. This makes it difficult to observe this process, and state-of-the-art climate models, such as CMIP6, need to use parameterizations to represent convective processes (Fox-Kemper et al., 2019). Given the importance of the deep convection for climate and its future change, a reliable representation in climate models is highly important. However, Heuzé (2017) stated that "the majority of CMIP5 models convect too deeply, over too large an area, too often and too far south". While CMIP6 models seem to provide some improvements in the representation of bottom waters, more improvements are required (Heuzé 2020).

Going beyond the horizontal resolution of CMIP6 models, we analyse in this study whether high-resolution models from the CMIP6 High-Resolution Model Intercomparison Project (HighResMIP, Haarsma et al., 2016) improve the representation of deep convection in the subpolar North Atlantic. We use simulations from seven models participating in HighResMIP, which have been performed in the EU-H2020-project PRIMAVERA. Most of the high-resolution model versions of these seven models have ocean grid resolutions of around 10-25 km. Since the baroclinic Rossby radius of deformation is small in high latitudes (roughly 10 km in polar regions and 200 km in the tropics), mesoscale oceanic eddies are not resolved in the sub-polar convection regions in most of the HighResMIP models. We thus denote them as "eddy permitting" but not "eddy-resolving".

Recent studies found that the increased resolution in the HighResMIP models improved many aspects of the ocean including temperature and salinity biases (Gutjahr et al. 2019), the northward ocean heat transport (Grist et al., 2019), 
[revised manuscript text omitted]
., 2020). The set of HighResMIP experiments is divided into three tiers consisting of atmosphere-only and coupled runs and spanning the period 1950-2050 (details in Haarsma et al. 2016). Here, wWe use the Tier 2 historical coupled simulations from 1950-2014 and the 100-year control simulations (using constant 1950 forcing) from these seven models for our analysis. The control run used fixed 1950s forcing (GHG gases, including O3 and aerosol loading for a 1950s (~10 year mean) climatology). They will allow to evaluate potential model drifts in the historical simulations. Both control and historical runs are initialized from the end of 50 year spin-up simulations using 1950s-forcing. All models performed historical and controlthe simulations in at least two different resolutions. following the HighResMIP protocol. Changes in oceanic and atmospheric parameters are kept to a minimum between low and high resolution simulations, so that all changes can be directly attributed to the change in resolution.

[revised manuscript text omitted]

For calculation of the power spectrum, we used the method of Torrence and Compo (1998). Fourier transforms are calculated and red noise is used as background spectrum. To determine significance levels for the Fourier spectra the method of Torrence and Compo (1998) assumes that different realizations of the geophysical process will be randomly distributed

265 about this background spectrum, and the actual spectrum can be compared against this random distribution.

**3 Deep Convection in the North Atlantic**

This section analyzes first the MLD in March, the month with the strongest convection in both observations and models, in the North Atlantic in the different models and in ARGO. Then, we focus on the DMV in the models, its variability and potential trends in the historical simulations.

270 **3.1 Mixed layer depth**

Figure 1 shows the averaged March MLD from ARGO and from all historical model simulations. In the period 2000-2015, the ARGO data suggest average MLDs of about 1000 m in the Labrador and GIN sSeas. In the models, the MLD differs greatly and shows a strong dependence on the spatial resolution. While ECMWF-IFS-LR and EC-Earth3P show no or very shallow MLD in the main convection areas of the North Atlantic, many of the other simulations strongly

275 overestimate the MLD compared to ARGO. Some models simulate much too strong convection in the Labrador Sea but do not show any deep convection in the GIN Seas while other models overestimate the MLD in both seas. In contrast, in the Irminger Sea, the MLD is more consistent across models and agrees better with ARGO. Note that we here compare models' MLD averaged over 1950-2014 with ARGO-data from 2000-2015. As we will discuss later more in detail, some of the models show a weakening of the convection with time, particularly in the Labrador Sea.

280

Although the convection centre varies somewhat across models, we do not find any clear linkage to resolution. The models with particularly deep mixed layers show also the largest convection areas. Thus, compared to ARGO, they do not only overestimate the depth of mixed layer but also the area of deep convection.

285

The MLD deepens with increasing ocean resolution in all models, except for AWI-CM-1-0. However, the models showing deepening MLDs are not fully independent, because they share NEMO as the ocean component, whereas AWI-CM-1-0 has FESOM as ocean component. On the other hand, even the  models with NEMO3.6 as ocean component (compare HadGEM3-GC31, CNRM-CM6.1, CMCC-CM2 and EC-Earth3P) differ considerably. This discrepancy suggests that either the different atmospheric components or the choice of ocean parameters have a strong influence on the convection. In contrast, the MLD differs little when the atmosphere resolution is increased  (compare ECMWF-IFS-MR and ECMWF-IFS-HR, HadGEM3-GC31-MM with HadGEM3-GC31-HM, CCCM-CM2-HR4 with CCCM-CM2-VHR4). An exception is  MPI-ESM1-2, where an increased atmospheric resolution  reduces  MLD. This MLD reduction can  be linked to too weak wind forcing in MPI-ESM1-2-XR (Putrasahan et al., 2019).

To investigate the impact of natural variability on the mean March MLD in the historical period and to quantify  the potential contribution of natural variations to the differences in MLD with changing resolution, we use an ensemble of historical simulations with the ECMWF-IFS model. The MLD in the low resolution version ECMWF-IFS-LR is very shallow in all 6 ensemble members and there is no deep convection in the historical and control simulations (not shown). Thus, we concentrate in the following on the four members of ECMWF-IFS-HR, which all exhibit pronounced deep mixing, particularly in the Labrador Sea. These four ECMWF-IFS-HR members underlie  a considerable natural variability (Figure 2 a-d). The averaged March MLD (1950-2014) deviates in individual ensemble members up to about 200m from the ensemble mean MLD. Although, this is a considerable amount, given the relatively long averaging period, the MLD differences due to increased resolution from 1° to a 0.25° in the NEMO-models are larger (compare Figure 2 to differences between 1° and 0.25° simulations in Figure 1).  Although the DMV varies considerably between the ensemble member, the general amplitude, frequency and trends are similar.
Even though four members are not sufficient to capture the total natural variability, these results suggest that natural variability cannot explain the differences in MLD due to a change in spatial resolution.

**3.2 Deep Mixed Volume**

In the following, in order to consider the horizontal extension of convection patterns and discard shallow convection events that have limited impact on the oceanic circulation such as the AMOC, we will concentrate on the DMV index to investigate the deep convection in the Labrador and GIN Seas in more detail.

**3.2.1 Labrador Sea**

Figure 3 shows the DMV in the Labrador Sea in March in the historical model simulations. In agreement with Figure 1, increasing the ocean resolution from around 1° to 0.25°  generally leads to an increased DMV in all models using NEMO (ECMWF-IFS, HadGEM3-GC31, CNRM-CM6-1, EC-Earth3P, see Table 2.1), while the opposite is true for AWI-CM-1-0. creas in ocean resolution further to 1/12 ° in HadGEM3-GC31-HH does not further increase the DMV. The DMV varies strongly among models: ECMWF-IFS-LR does not show any deep convection events in the entire historical period, CNRM-CM6.1 and EC-Earth3P simulate only a few events with deep convection and AWI-CM-1-0-LR and CMCC-CM2 simulate strong deep convection every winter.

Table 2 compares the average DMV in the historical model simulations with that of ARGO in the period 2000-2015. We compare both the entire period 1950-2014 and the period 2000-2014 to ARGO. Generally, the simulated DMV in the Labrador Sea is smaller in 2000-2014 compared to the entire period. On the other hand, natural variability of the DMV is high and thus a 15-year period is probably too short for a comparison.  The DMV in EC-Earth3P-HR and CNRM-CM6.1 are closest to ARGO when considering the entire 1950-2014 period into account, however, they underestimate ARGO in 2000-2014. As discussed above, EC-Earth3P and ECMWF-IFS-LR show no or rather little deep convection in the Labrador Sea while the other simulations  overestimate the ARGO-based DMV with factors of four to almost 40 when taking the entire time period into account (3-25 in the period 2000-2014). The MPI-ESM1.2-XR overestimates ARGO in 1950-2014 but substantially underestimates it in 2000-2014. This indicates the difficulties when comparing short time periods with trends. Despite the uncertainties in the comparison to ARGO, it is clear that the models seem to have problems to realistically simulate the convection in the Labrador Sea. If deep convection occurs, the ocean is often mixed down to the bottom in the models, wherea deep convection rarely exceeds 2000m in the observations (Yashayaev and Loder, 2016; Yashayaev and Loder, 2017).

If we use a critical depth of $z_{crit}$=0 m instead of 1000 m in the Labrador Sea and thus consider the total mixed layer depth, the relative deviation of the DMV in the models from ARGO is reduced as expected (not shown). However, AWI-CM-1-0-LR and CMCC-CM2 still overestimate the DMV based on ARGO by a factor of three and two, respectively. On the other hand, ECMWF-IFS-LR simulates only 20% of the mixed volume compared to ARGO. The comparison between $z_{crit0}$ and $z_{crit1000}$ reveals also some non-linearites in the deep convection. While CNRM-CM6.1-HR has a nine times higher DMV ($z_{crit1000}$) compared to ARGO, it is only 16% higher for $z_{crit0}$, whereas the DMV ($z_{crit1000}$) for MPI-ESM1-2-XR is 4.6 times higher compared to ARGO but 14% smaller for $z_{crit0}$.

The interannual variability of the DMV is large in all models (Figure 3). Some of the simulations (EC-Earth3P, HadGEM3-GC31-LL, MPI-ESM1-2, CNRM-CM6.1 and ECMWF-IFS-MR) also indicate substantial variability at decadal or longer periods where phases with and without convection alternate. Such intermittent deep convection was also suggested based

on observations (Lazier et al., 2002; Yashayaev and Loder, 2016). . However, in most of the model simulations, deep convection occurs in almost every winter or not at all.

The strength of the deep convection in March is reflected in the vertical density distribution in the Labrador Sea. To calculate the density, we used the definition of density, following Millero and Poisson (1981). Naturally, the models with more frequent and deeper convection show a much weaker vertical stratification than the models that do not exhibit deep convection. We therefore analyse density profiles in the Labrador Sea in November that are not influenced by convection to explain why the mixed layer depth is overestimated in the following winter.  Figure 4 shows the vertical density stratification of the upper 600m in the Labrador Sea. All models show a near surface low density layer, mainly due to a combination of low surface salinity and relatively (compared to late winter) warm water near the surface in November. Generally, the models with lower ocean resolution show a stronger stratification in the upper ocean than models with higher resolution (except for AWI-CM-1-0). The two model simulations, which do not simulate any deep convection, ECMWF-IFS-LR and EC-Earth3P, show particularly strong upper ocean density gradients. Consequently, a large buoyancy flux would be needed during winter until deep convection could set in in these two models. MPI-ESM1-2 and AWI-CM-1-0 show a more stratified upper ocean in November with increased atmospheric resolution, which  agrees with a weaker convection in their higher resolution versions. The density profiles of the high ocean resolution models agree  well with the observed one from ARGO, although the near surface low density layer is too shallow in most of these models. This shallower surface layer requires less heat to be eroded, which might explain the overestimation of the deep convection in late winter in these models (compare Figures 1 and 3). However, this is probably not the only reason as will be further discussed in section 4.

Twelve of 19 simulations indicate a significantly negative trend of the DMV in the historical period (Figure 3, Table 3). To investigate whether this trend is  due to external forcing or due to model drift , we compared the DMV in the historical simulations with that from the 100-year 1950-control simulations (Figure 5 and Table 3). Most of the control simulations do not show any significant trend and in 9 out of 17 historical simulations, the DMV trends in the historical simulations are significantly more negative compared to the first 65 years of the control simulations. This indicate external forcing as major cause for the DMV reduction in the historical simulations. Furthermore, the negative trends become larger with higher ocean resolution.

A reduction of DMV in the historical period would be in line with some recent studies by Caesar et al. (2018), Thornalley et al. (2018) and Brodeau and Koenigk (2016).

To investigate the predominant variability frequency of the DMV, we calculated the power spectrum of detrended DMV time series of the Labrador Sea from the 100-year long control simulations

385  (Figure 6).  The dominant time scale varies across simulations and models. Many of the models  (ECMWF-MR, HadGEM3-GC31-MM, HadGEM3-GC31-HM, , EC-Earth3P-HR,  MPI-ESM1-2-HR) show a  peak in the spectrum at around 10 years. In ECMWFS-IFS-
390 HR, HadGEM3-GC31-HH and MPI-ESM1-2-XR with further increased resolution in either ocean or atmosphere, the main peak in the spectrum seems to shift towards somewhat longer time periods. A shorter frequency of 7 years is found for CMCC-CM2-HR  and AWI-CM-1-0-HR . The lower ocean-resolution models, AWI-CM-1-0-LR, HadGEM3-GC31-LL and ECMWF-IFS-LR (but very weak DMV), show less dominate peaks than their higher resolution versions. CNRM-CM6.1 shows peaks at similar periods as CNRM-CM6.1-HR
395 but the amplitudes differ.

**3.2.2 GIN Seas**

The DMV in the GIN Seas shows also a large spread across models (Figure 7, Table 2). As for the Labrador Sea, AWI-CM-1-0-LR and the two CMCC-CM2 simulations show the strongest deep convection while ECMWF-IFS and
400 EC-Earth3P simulate rather weak convection (note the different order of magnitude in the vertical scales of Figure 7).  MPI-ESM1-2-HR and CNRM-CM6.1 show the smallest deviations  from the ARGO observations. HadGEM3-GC31-HM overestimates MLD by 70% when comparing the full historical period to ARGO (400% when comparing only 2000-2014) and MPI-ESM1-2-XR underestimates MLD by 50%. EC-Earth3P and ECMWF-IFS strongly underestimate deep water formation
405 in the GIN Seas while the DMV is  strongly overestimated by AWI-CM-1-0 and CMCC-CM2.
The resolution dependency of  DMV in the GIN Seas differs substantially from the Labrador Sea. There is no robust relation between MLD and resolution. All models with NEMO show either no or shallower MLD with increased resolution. ~~The NEMO models do not show any deepening of convection with increased ocean resolution. In contrast, the high resolution versions of ECMWF-IFS, EC-Earth3P and CNRM-CM6.1 (after 1980) show very shallow or no deep
410 convection. Moving from HadGEM3-GC31-LL to HadGEM3-GC31-MM leads to substantially smaller DMV. However, when increasing the atmospheric resolution from HadGEM3-GC31-MM to HadGEM3-GC31-HM, DMV increases, and increases further in HadGEM3-GC31-HH, where both ocean and atmosphere resolution is increased. Thus, unlike for the Labrador Sea, there is no robust relation between the resolution in ocean or atmosphere and DMV in the Greenland Sea in the global models with NEMO as ocean component. The two other models,~~ MPI-ESM1-2 and AWI-CM-1-0 show, as for
415 the Labrador Sea, a reduction of the DMV with increased resolution.

The trends of DMV in the GIN Seas  do not agree across models in the historical period (Figure 7, Table 3). While HadGEM3-GC31-MM, MPI-ESM1-2-XR and the CMCC-CM2 simulations show a significantly negative trend, HadGEM3-GC31-HM and EC-Earth3P show positive trends. This discrepancy could be due to the competing effects from global warming, which are represented differently in each model: on one hand,  reduced sea ice extent  enabl a larger surface for deep convection, while on the other hand  melt water  and warm surface water enhance the stratification and thus impede convection. The high DMV in CNRM-CM6.1-HR before 1970, and the decrease thereafter occurs similarly in the first decades of the 1950-control-simulation indicating that this trend is caused by a model balance adjustment and not due to external forcings (Figure 8, Table. 3). Similarly, the strong negative trends in CMCC-CM2 can partly be explained by similar drifts in the control simulations, although the reduction in the historical runs is significantly larger than in the control runs.

Some of the historical and control simulations show strong decadal or longer-term variations (Figure 9). HadGEM3-GC31-HM, MPI-ESM1-2-HR, AWI-CM-1-0-HR and EC-Earth3P-HR show dominant variability at periods of around 20-25 years; ECMWF-IFS-MR, HadGEM3-GC31-HH, CMCC-CM2 and CNRM-CM6-1 simulations at time scales of 10-15 years. In addition, some of the simulations indicate variability at time-scales below 8 years. Overall, there is no clear dependence of the variability on the resolution.

Figure 10 summarizes the results from sections 3.2.1 and 3.2.2. on the resolution dependency of the deep convection across the models in Labrador and GIN Seas. Each single model shows a clear dependence of the DMV in the Labrador Sea on the oceanic resolution. The differences across models are large, and as discussed before, even models using the same version of the NEMO as ocean model exhibit a wide range of solutions. Models with coarse resolution (~100 km) produce no or only shallow convection. Models with a resolution of 50km and higher in the ocean, however, overestimate deep convection compared to ARGO.

Increasing the atmosphere resolution has a minor impact on the DMV in the Labrador Sea, except for MPI-ESM1.2 and AWI-CM-1-0, where DMV is reduced with increased resolution.

The resolution dependency of the DMV in the GIN Seas in single models is smaller than in the Labrador Sea. However, all the models, except for CNRM-CM6-1, show a decreased DMV when increasing the resolution to around 0.25°. The response to increased atmosphere resolution is not robust across models.

**4 The impact of heat and freshwater fluxes on the deep convection in the North Atlantic**

Deep convection depends strongly on the buoyancy of the ocean surface layer in the convection regions - the heat loss to the atmosphere and the influx of fresh water into the convection regions.

**4.1 Surface heat fluxes**

Brodeau and Koenigk (2016) showed that the turbulent surface heat flux (SHF) is the main driver for interannual variability in the DMV. Thus, in the following, we will mainly focus on the SHF.

Figure 11 shows the winter (January, February, March) SHF in each of the model simulations. The WHOI-OAFlux data show the largest SHF (from the ocean to the atmosphere) up to more than 200 W/m$^2$ from the ice edge in the Labrador Sea extending to the southern part of the subpolar gyre, south of Iceland and along the southeast coast of Greenland, and in the northern Norwegian-Greenland Seas and Barents Sea. The large-scale features of this pattern are reproduced by most of the models. ECMWF-IFS-LR and to a lesser degree EC-Earth3P, both simulating too weak convection, strongly underestimate the SHF in the Labrador Sea. Increased ocean resolution improves the representation of the observed SHF pattern. In particularly,  the extension of high SHF from the Labrador Sea into the southwestern branch of the sub-polar gyre and the high SHF in the northern Greenland and Norwegian Seas is better simulated. A number of models ( both CMCC-CM2 versions, HadGEM3-GC31-HH and CNRM-CM6.1) overestimate the SHF in the sub-polar gyre. In addition, the SHF west and northwest of Scotland is too high in most of the models.

In the Labrador Sea, all high-resolution models with NEMO as the ocean component simulate increased SHF (averaged over the same box as used for calculation of the DMV) compared to their lower-resolution counterparts (Table 2). In contrast, MPI-ESM1-2 shows reduced SHF with increased atmospheric resolution in line with the reduced convection.  In all models, the interannual variations of winter SHF  is significantly positively correlated with the DMV in March. hus, large ocean heat losses in the winter are linked to strong DMVs in the following March, indicating that large upward surface heat fluxes lead the DMV. The correlation coefficient varies from 0.48 in CNRM-CM6.1 to slightly above 0.7 in ECMWF-IFS-MR, EC-Earth3P and CMCC-CM2-HR4. The relation between SHF and DMV is neither resolution nor model dependent.

The winter SHF in the Labrador Sea itself is governed by the atmospheric circulation (not shown). In all model simulations northerly to northwesterly winds, which advect cold air from the Arctic sea ice towards the Labrador Sea, lead to strong surface heat fluxes, which overcome the stratification of the ocean (Ortega et al. 2011), and increased convection. These north-to-northwesterly winds are further linked to a large scale atmospheric circulation pattern, which is similar to the positive phase of the North Atlantic Oscillation (NAO. ) defined as the leading EOF of geopotential height on the 500 hPa

pressure surface over the European/Atlantic sector (80°W-40°E, 20°-90°N)). The spatial imprint of the NAO-index on the 500-hpa geopotential height is shown in Figure 12. All models reproduce the NAO-pattern of the ERA5-reanalysis data well. However, the position of the negative pole over Iceland-Greenland and the extension of the positive pole towards Eurasia vary slightly among models. The NAO-index itself, which we define here as the difference of the normalized winter sea level pressure anomalies over the Azores and Iceland, is significantly positively correlated with the DMV$z_{crit1000}$ in the Labrador Sea in all simulations except for the low resolution simulations with EC-Earth3P and ECMWF-IFS. These are the simulations with no or only little deep convection and which have a strongly stratified ocean. The other model simulations show correlations between 0.38 (HadGEM-GC31-LL) and 0.67 (HadGEM-GC31-HM and CMCC-CM2-HR4). The NAO is not only important for interannual variations of the DMV but also on the decadal scale. Correlations of 10-year running means of NAO and DMV reach between 0.3 and 0.57. A spectral analysis of the NAO resembles most of the peaks in the spectrum of the DMV (not shown) although the NAO shows relatively more energy at shorter time scales compared to the DMV in the Labrador Sea.

As in the Labrador Sea, northerly winds are the main cause for large oceanic surface heat loss to the atmosphere in the Greenland Sea. The northerly winds are connected to low pressure anomalies over northern Scandinavia and the Barents Sea. As in the Labrador Sea, tThe DMV in the Greenland SeaGIN Seas is correlated to the SHF as well and ocean heat loss is linked to a large DMV. Northerly winds are the main cause for large oceanic surface heat loss to the atmosphere in the GIN Seas. However, here we find a strongerer model dependency of the correlation than in the Labrador Seais relation. The correlation is weak to moderate in HadGEM3-GC31 (r=0.22 for LL; r=0.5 for HH) and in CNRM-CM6.1 (r=0.35; r=0.5 for HH) but high correlation is found for ECWMF-IFS (r=0.64 for HR; r=0.85 in LR) and EC-Earth3P (r=0.61; r=0.69 for HR). As for the Labrador Sea, the relation between SHF and DMV shows no clear resolution dependency.

**4.2 Freshwater and sea ice exports**

A number of studies have previously discussed the effect of Arctic freshwater export, especially through Fram Strait, as a potential source of variability of the deep water convection in the Labrador Sea (Holland et al., 2001; Jungclaus et al., 2005; Koenigk et al., 2006). Here, we analyze the correlations between freshwater transports across different sections (Fram Strait, Denmark Strait, northern Baffin Bay) and deep convection in the Labrador Sea as well as betweennd transports through Fram Strait and deep convection in the Labrador Sea and in the Greenland SeaGIN Seas convection(transports through Fram Strait) in the historical simulations of the models.

To calculate the liquid freshwater transport, we used the model grid lines on the native grids of the models that are closest to the geographical landmarks that define Fram Strait (across 78°N), Northern Baffin Bay (78°N) and Denmark Strait (66°N). The freshwater has been defined as the amount of zero-salinity water required to reach the observed salinity of a seawater sample starting from a reference salinity. Specifically, liquid freshwater transport (fwt, in $m^3/s$) is estimated as

$$fwt = \int\limits_{p1}^{p2} \int\limits_{D}^{\eta} \left(\frac{S - Sref}{Sref}\right) dz dx$$

for salinity S (in practical salinity units). As reference salinity Sref we used 34.80 psu for all models. The integration along z is performed from the bottom at depth D to the sea surface at height η (in this case η=0). p1 and p2 are the landmarks and the integration was done considering dx as the length (or depth for dz) between every grid point.

The solid freshwater transport is calculated from the sea ice transports across the sections assuming a constant ice salinity of 5 psu.

Table 4 shows the freshwater exports out of the Arctic into the North Atlantic through Fram Strait and Baffin Bay and through the Denmark Strait. Although differences between models are large, the exports through Fram Strait are generally larger than through Baffin Bay. The total freshwater exports through Fram Strait (liquid + solid export) varies between around 80000 m$^3$/s in the two CNRM-CM6.1 models and 160000 m$^3$/s in ECMWF-IFS-LR and HadGEM3-GC31-MM. The distribution between liquid and solid export through Fram Strait differs strongly across the simulations. While in HadGEM3-GC31-LL and all ECMWF-IFS and EC-Earth3P simulations most of the freshwater leaves the Arctic in the form of sea ice, liquid and solid parts are of similar size in the other models. In CNRM-CM6.1-HR, the liquid part is even larger than the solid part. The amount of freshwater that passes the Denmark Strait is reduced compared to Fram Strait in all models except for ECMWF-IFS-LR and MPI-ESM1.2-XR, and the liquid part is dominating. Large parts of the ice melt in the East Greenland Current on its way from Fram Strait to Denmark Strait.

The low resolution versions of ECMWF-IFS, HadGEM3-GC31, CNRM-CM6.1 show a larger fraction of solid exports through Fram Strait and larger liquid transports through Baffin Bay (EC-Earth3P as well for Baffin Bay) compared to their higher resolution counter parts. The sum of freshwater exports through Fram Strait and Baffin Bay differs more between the models than between different versions of single models. Despite the large differences in mean Arctic freshwater exports into the North Atlantic, there is no clear linkage to the mean DMV in GIN and Labrador Seas. Only for ECMWF-IFS-LR, we speculate that the very large freshwater fluxes, particularly in form of sea ice, through Denmark Strait contribute to the low surface density in the Labrador Sea (compare Figure 4) and consequently to suppress any deep convection activities in the Labrador Sea.

In order to further investigate if the variability of freshwater exports affects the deep convection in GIN and Labrador Seas, we correlate the solid and liquid transports across all sections with the DMV.  In all model simulations, the annual mean southward transport of both liquid and solid freshwater across Fram Strait and the liquid transport across Denmark Strait are weakly negatively correlated with the deep convection in the Labrador Sea in March ( range

Formaterat: Teckensnitt:(Standard +Rubriker (Times New Roman)

Formaterat: Teckensnitt:(Standard +Rubriker (Times New Roman)

Formaterat: Teckensnitt:(Standard +Rubriker (Times New Roman)

Formaterat: Upphöjd

Formaterat: Upphöjd

[revised manuscript text omitted]

Docquier, D., Fuentes-Franco, R., Koenigk, T., and Fichefet, T.: Sea ice - ocean interactions in the Barents Sea modeled at different resolutions, Front. Earth Sci. 8:172, doi: 10.3389/feart.2020.00172.

Eden, C., and Willebrand, J.: Mechanism of interannual to decadal variability of the North Atlantic circulation, J. Clim., 14(10), 2266–2280, doi:10.1175/1520-0442(2001)014, 2001.

Fox-Kemper, B., Adcroft, A., Böning, C. W., Chassignet, E. P., Curchitser, E., Danabasoglu, G. Eden, C. and co-authors: Challenges and Prospects in Ocean Circulation Models, Front. Mar. Sci., 26, https://doi.org/10.3389/fmars.2019.00065, 2019.

[revised manuscript text omitted]

**Formaterad tabell**

| | | | | | | |
|---|---|---|---|---|---|---|
| **CMCC-CM2-HR4** | 24.4 (21.0) | 1.22 | 13.0 (7.8) | 1.22 | 0.72 | 0.52 |
| **CMCC-CM2-VHR4** | 24.8 (25.6) | 1.34 | 15.0 (13.1) | 1.14 | 0.59 | 0.58 |
| **CNRM-CM6.1** | 1.09 (0.13) | 1.15 | 1.10 (1.30) | 1.07 | 0.53 | 0.45 |
| **CNRM-CM6.1-HR** | 9.3 (3.2) | 1.18 | 2.47 (0.04) | 1.35 | 0.48 | 0.35 |
| **MPI-ESM1-2-HR** | 10.6 (7.1) | 1.14 | 1.2 (0.69) | 1.25 | 0.61 | 0.44 |
| **MPI-ESM1-2-XR** | 4.6 (0.30) | 0.98 | 0.6 (0.47) | 1.16 | 0.64 | 0.64 |
| **AWI-C-1-0-LR** | 39.5 (24.9) | no | 20.9 (18.8) | no | no | no |
| **AWI-CM-1-0-HR** | 12.8 (10.1) | data | 6.1 (4.8) | data | data | data |
| **EC-Earth3P** | 0.26 (0.05) | 0.63 | 0.24 (0.38) | 1.02 | 0.72 | 0.69 |
| **EC-Earth3P-HR** | 0.95 (0.21) | 1.07 | 0. (0) | 0.79 | 0.50 | 0.61 |

**Table 2: Observed and modeled DMV and SHF in the Labrador and GIN sSeas, the ratio between model and observed values and  correlations between SHF and DMV. Row 2: DMV and SHF in observations, shown are absolute values. Rows 3-9: Ratio of modeled and observed DMV- and SHF (Model values divided through observational values: DMV$_{model}$/ DMV$_{obs}$ and SHF$_{model}$/ SHF$_{obs}$). For ECMWF-IFS, ensemble means are shown. For the DMV (columns 2, 4), the first number compares the mean of the entire historical simulation (1950-2014) to ARGO. The number in brackets only the years 2000-2014. Columns 6, 7: Correlation between winter SHF and March DMV in the respective boxes of the Labrador and GIN Seas. For the correlations $z_{crit0}$ has been used to avoid complications with periods without any deep convection; the correlations based on $z_{crit1000}$ and $z_{crit700}$ in LAB and GIN Seas are generally similar for models with deep water formation in every winter but much lower in the models with no or very few deep convection events (ECMWF-LR, EC-Earth3P).**

| Model | Trend/year historical 1950-2014 DMV-Lab | Trend/year control-1950 year 1-65 DMV-Lab | Trend-difference: hist – control DMV-Lab | Trend/year historical 1950-2014 DMV-GIN | Trend/year control-1950 year 1-65 DMV-GIN | Trend-difference: hist – control DMV -GIN |
|---|---|---|---|---|---|---|
| **ECMWF-IFS-LR** **ECMWF-IFS-MR** **ECMWF-IFS-HR** | 0.003 -0.5 *-3.93e+12* *-3.40e+12* | -0.01 -0.43 | -0.02 *-3.39* *-2.97* | 0.01 0 0.002 | 0.05 0.005 0.01 | 0.06 -0.005 -0.012 |

| | | | | 09 | | |
|---|---|---|---|---|---|---|
| **HadGEM3-GC31-LL** | 0.41.09e+11 | 1.58e+12 | -1.17 | 6.62e+12 | 5.59e+12 | 1.03 |
| **HadGEM3-GC31-MM** | *-4.43e+12* | -1.11e+12 | *-3.32* | *-1.49e+12* | 0.9.73e+11 | -2.46 |
| **HadGEM3-GC31-HM** | *-5.13.3e+12* | -3.02e+12 | *-2.11* | *4.40e+12* | -0.33.28e+11 | *4.73* |
| **HadGEM3-GC31-HH** | *-6.66e+12* | -2.28e+12 | *-4.38* | 0.3.10e+11 | -0.95.49e+11 | 1.25 |
| **MPI-ESM1-2-HR** | -1.41e+12 | -0.62.15e+11 | -0.79 | - | *-0.8.553e+11* | -0.02 |
| **MPI-ESM1-2-XR** | *-7.94e+12* | -0.06.02e+10 | *-7.88* | 0.82.17e+11 *-0.41.06e+11* | 0.23.32e+11 | -0.64 |
| **CMCC-CM2-HR4** | *-5.19e+12* | -1.08e+12 | *-4.11* | *-9.31e+12* | -3.97e+12 | *-5.34* |
| **CMCC-CM2-VHR4** | -1.42e+12 | -3.27e+12 | 1.85 | *-1.2.0e+13* | -2.69e+12 | *-14.69* |
| **CNRM-CM6.1** | *-0.54.38e+11* | 0.1.62e+11 | *-0.70* | 0.8.61e+11 | 0.4.51e+11 | 0.41 |
| **CNRM-CM6.1-HR** | *-6.91e+12* | -0.31.06e+11 | *-6.60* | *-1.0.1e+13* | *-1.0.5e+13* | 0.4 |
| **AWI-CM-1-0-LR** | *-1.5.0e+13* | *-1.6.88e+13* | 1.80 | -1.97e+12 | 0.6.80e+11 | -2.65 |
| **AWI-CM-1-0-HR** | *-0.96.56e+11* | -4.98e+12 | 4.02 | 1.21e+12 | *-2.50e+12* | 3.71 |
| **EC-Earth3P** | 0.17.69e+11 | 0 | 0.17 | *0.43.25e+11* | *0.033.05e+10* | *0.40* |
| **EC-Earth3P-HR** | *-0.39.87e+11* | 0.9.22e+11 | *-1.31* | - 0.001.1.39e+09 | 0.003.04e+09 | -0.002 |

**Table 3: Trends in the DMV in the Labrador and GIN Seas in the historical simulations and in the first 65 years of the 1950-control simulations (in 10$^{12}$ m$^3$/year). Trends that are significantly different from 0 at the 95%-confidence level are shown in italic, trends significantly different to the control-runs are bold, and trends**

920 **significantly different to both 0 and the control-run are italic and bold.**

| Freshwater fluxes in m$^3$/s | Fram Strait liquid | Fram Strait solid | Denmark Strait liquid | Denmark Strait solid | North Baffin Bay liquid | North Baffin Bay solid | Sum Fram Strait + Baffin Bay |
|---|---|---|---|---|---|---|---|
| **ECMWF-IFS-LR** | 12694 | 154000 | 60517 | 92093 | 21688 | 7181 | 195563 |
| **ECMWF-IFS-MR** | 58595 | 85108 | 69694 | 12635 | 17597 | 7031 | 168331 |
| **ECMWF-IFS-HR** | 47578 | 107470 | 68940 | 22533 | 14479 | 7367 | 176894 |
| **HadGEM3-GC31-LL** | 30394 | 85642 | 25304 | 28543 | 55002 | 7043 | 178081 |
| **HadGEM3-GC31-MM** | 81239 | 82197 | 69489 | 21907 | 19993 | 6614 | 190043 |
| **HadGEM3-GC31-HM** | 72294 | 73470 | 61834 | 13905 | 18728 | 10309 | 174801 |
| **HadGEM3-GC31-HH** | no data | 56979 | no data | 12845 | no data | 17117 | no data |
| **CMCC-CM2-HR4** | 70915 | no data | 60545 | no data | 14854 | no data | no data |
| **CMCC-CM2-VHR4** | 61063 | no data | 13785 | no data | 6239 | no data | no data |
| **CNRM-CM6.1** | 39699 | 43906 | 33344 | 8689 | 35078 | 3902 | 122585 |
| **CNRM-CM6.1-HR** | 52321 | 29243 | 41327 | 15409 | 22297 | 8540 | 112401 |
| **MPI-ESM1-2-HR** | 66843 | 54540 | 92680 | 8453 | 1276 | 2232 | 124891 |
| **MPI-ESM1-2-XR** | 54834 | 62348 | 112003 | 12408 | 2231 | 3145 | 122558 |
| **EC-Earth3P** | 26843 | 96230 | 30439 | 28277 | 25317 | 5813 | 154203 |

| EC-Earth3P-HR | 22096 | 105890 | 54655 | 27370 | 1776 | 3488 | 133250 |
|---|---|---|---|---|---|---|---|

**Table 4:** Liquid and solid freshwater fluxes through Fram Strait, Denmark Strait and northern Baffin Bay in the historical simulations averaged over 1950-2014. Positive values mean freshwater exports out of the Arctic. The last column shows the sum of liquid and solid exports through Fram Strait and Baffin Bay. No data were available to calculate transports in the AWI-CM-1.0 model, liquid freshwater transports in HadGEM3-GC3-HH and sea ice transports in the CMCC-simulations.

[Figure]

**Figure 1: Mixed layer depth in March in the ARGO-data, averaged over 2000-2015 (a) and in the historical low and high-resolution model simulations, averaged over 1950-2014 (b-r).**

940

945

950

[Figure]

March MLD, 1950−2014, ECMWF−IFS−HR. Deviation from ensemble mean
a) Member 1   b) Member 2   c) Member 3   d) Member 4

−300 −200 −150 −100 −60 −20   20  60  100  150  200  300
m

e) DMV in the Labrador Sea in ECMWF-IFS-HR

[Figure]

[Figure]

**Figure 2: a-d) Deviation of mixed layer depth in March in the ensemble members of ECMWF-IFS-HR from the ensemble mean of the four ECMWF-IFS-HR simulations for the time period 1950-2014. e) DMV in the Labrador Sea (in $10^{15}$ m$^3$) in the ensemble mean and single ensemble members of ECMWF-IFS-HR.**

**Formaterat:** Upphöjd

**Formaterat:** Upphöjd

[Figure]

[Figure]

970

**Figure 3: Deep Mixed Volume (DMV) in $10^{15}$ m$^3$ using a critical depth of 1000 m in the Labrador Sea in March between 1950-2014. Note the different y-axis between models. For ECMWF-IFS, only member 1 is shown for better visual comparison of the variability across resolutions.**

975

[Figure]

[Figure]

**Figure 4: Density (in kg/m³) in the upper 600 m averaged over the Labrador Sea in November.**

[Figure]

[Figure]

**Figure 5: As Figure 3 but for the 100-year control simulation.**

990

995

[Figure]

[Figure]

**Figure 6:** Power spectrum of detrended and normalized March DMV time series of the 100-year control simulation in the Labrador Sea. The dashed red line shows the 95% significance level. The y-axis uses a different scale depending on the model. Note, that no deep convection occurred in the 100-year period in EC-Earth3P.

**Formaterat:** Teckensnitt:Inte Fet

[Figure]

[Figure]

**Figure 7: As Figure 3 but for the GIN Seas and a critical depth of 700 m.**

1020

1025

[Figure]

[Figure]

**Figure 8: As Figure 7 but for the 100-year 1950-control simulations.**

1035

1040

[Figure]

[Figure]

**Figure 9: Same as Figure 6 but for DMV in the GIN Seas.**

[Figure]

[Figure]

**Figure 10: DMV (average over 1950-2014 in 1012 m3) in Labrador Sea (left) and GIN-Seas (right) in March in dependence on the oceanic (top) and atmospheric (bottom) resolution. Thin lines connect model versions with different resolution of the same model.**

**Formaterat:** Upphöjd

[Figure]

[Figure]

**Figure 11: Turbulent surface heat flux (January, February, March average) in 1950-2014 in the WHOI-OAFlux data and in the model simulations. Positive values mean flux from the ocean to the atmosphere.**

[Figure]

 **Figure 12:** Pearson correlation between geopotential height at 500 hPa and North Atlantic Oscilation (NAO) index during winter (JFM mean) for ERA5 and the models. The periods used were 1979-2019 for ERA5 and 1950-2014 for the models.

1085

[Figure]

1090

Figure 1 2: Regression between annual mean freshwater transport through the Denmark Strait and mixed layer depth in the following March. a) HadGEM-CG31-LL and b) EC-Earth3P-HR. Data have been detrended before calculating the regression. These two simulations show the largest correlation between Denmark Strait freshwater transport and DMV in the Labrador Sea (-0.4 and -0.35 for HadGEM-GC31-LL and EC-Earth3P-HR, respectively).

1095

**AMOC and DMV in the Labrador Sea (control-1950)**

[Figure]

[Figure]

[Figure]

Figure 143: Crosscorrelation between the DMV using a critical depth of 1000 m in the Labrador Sea in March and the AMOC index for the 100-year control simulation. Both timeseries were detrended and filtered with a 10 years low-pass filter. Area enclosed by dotted lines represents the 95% confidence calculated as 2/sqrt(N), where N is the number of independent data based on the time that takes autocorrelation to fall below 1/e. Positive lags mean AMOC leads DMV, negative lags mean DMV leads AMOC. The low resolution version of EC-Earth3P does not produce any deep convection events in the control simulation.

**AMOC and DMV in the Greenland Sea (control-1950)**

[Figure]

[Figure]

**Formaterat:** Svenska (Sverige)

[Figure]

**Figure 14: As Figure 12 but for the GIN Seas and a critical depth of 700 m.**

---

## Author Comment (AC2) · 5 Aug 2020

Response to Reviewer 2:
This paper represents important material for further development of global climate model. The authors do a decent job in getting an overview of the different models and their performance. My comments are mainy regarding the formulations and figures, which could be improved.

[Figure]

Response: Thank you for your time and your useful comments. Below, please find point-to-point responses. All changes we suggest to include in a revised manuscript (including changes in figures and tables) are highlighted in the supplement file. Although, we are aware that we are not supposed to include a revised manuscript in this reply, we found it more practical for both us as authors and you as reviewer to highlight the suggested improvements into the manuscript itself rather than attaching all figures and tables separately.
* * *
Main comments: Introduction: in many places, the style is a fast shift between presenting 'settled knowledge' and presenting the moerations. This leads to very long sentences, which are difficult to read. For instance: L 72-74 (and how is it questioned?) and l. 75-79. The whole section must be revised according to this to become more clearly structured. The introduction, or may be the models and simulations section, must discuss 'high resolution' in relation to the Rossby radius and resolving eddies.

Response: We rewrote almost the entire introduction to better structure it. We now first introduce deep water formation, focusing then on it is importance for water masses, followed by the effects on local and remote climate. This is followed by a longer section focusing on the AMOC and potential linkages between deep convection and AMOC before we describe the aim of the study, discuss high resolution (setting it also in relation to the Rossby radius) and potential effects of high resolution.
* * *
Power spectra calculated in Figs 6 and 9 and referred to in the text. As far as I can see, these 'spectra' are raw periodograms. They will have 100 independent points (for a 100 year simulation), and therefore, five points will exceed the 95% significance curve just by chance. So periodogram spectra don't tell much. There are ways to overcome this, see e.g. von Storch and Zwiers (1999). The spectral analysis must be improved along these lines. Also, it must be specified, how the red significance level is calculated. The

spectra with red lines does not seem to describe the background spectra of the model data very well; this looks odd, and must be explained. Conclusions in text must be changed according to revised spectral analysis.

Response: The power spectra were calculated using a standard python program (https://pycwt.readthedocs.io/en/latest/index.html#) following the suggestions by Torrence and Compo (1998), which is widely used. The power spectrums were calculated with a Fourier transform, and red noise was used as background spectrum. To determine significance levels for Fourier spectra the method of Torrence and Compo (1998) assumes that different realizations of the geophysical process will be randomly distributed about this mean or expected background, and the actual spectrum can be compared against this random distribution. Afterwards Torrence and Compo derive the theoretical red noise wavelet power spectra and compared them to Monte Carlo results. These spectra are used to establish a null hypothesis for the significance of a peak in the power spectrum. The method by Torrence and Compo (1998) agrees with the one suggested by von Storch (see page 223 of the book). The theory for the Fourier transform, and for red noise (and its spectrum) are exactly the same (compare equation 11.23 of von Storch with equation 16 of Torrence and Compo (1998)). We added information on the calculation of the power spectra to the method section in the manuscript. We also, following suggestions from reviewer 1, changed the scale of the x, and y-axis.

––––––––––––––––––––––––––-

Minor comments: L 1-2 (title): The paper is comparing what could be called (present-day) standard resolution with higher resolution. The title should reflect this.

Response: We changed the title to " The effect of increasing the horizontal resolution on the deep water formation in the North Atlantic Ocean in HighResMIP models" to reflect this.

––––––––––––––––––––––––––
L. 82: '..future model simulations ..', write e.g: '.. model simulations of future climate. . .'

Response: changed
* * *
L. 87: 'The question whether . . .'. Why not write: 'It is still discussed . . .'

Response: changed
* * *
L. 107: '..important role.' For what?

Response: For the AMOC. We added this to the sentence.
* * *
L. 110: 'Climate-related processes' Be more specific, please.

Response: We replaced "…climate-related processes…" by "…and local and remote climate processes such as local sea ice cover or the large scale oceanic circulation"
* * *
L. 113: 'increasinng the hjorizontal resolution' .. of GCMs.

Response: Corrected
* * *
L. 117: Here you refer to a study with an eddy-resolving model. It should be made clear, that the present paper is not about eddy-resolving models.

Response: We rewrote large parts of the introduction and clarified this now, please see also our answer to main comment one.
* * *
L. 134: An outline of the HighResMip-protocol and a reference are needed here.

Response: We added more details from the HighResMIP-protocol in section 2.1, being more specific about forcings of the coupled simulations that have been used in this study. We added also a short description of the spin-up. For further details we refer to Haarsma et al. 2016.

———————————————-

L. 161: I cannot find a detailed description on how MLD is calculated in observations and in models. This must be added.

Response: The de Boyer Montégut et al. (2004) (0.03 kg m$-3$) variable density thresh-old is used to calculate the mixed layer depth in ARGO-data – see section 2.2. The ocean mixed layer thickness in the models is defined by the sigma t-criterion (Levitus, 1982) (variable mlotst following CMIP6-conventions). The sigma-t (density) criterion used in Levitus uses the depth at which a change from the surface sigma-t of 0.125 has occurred. We added this to section 2.3.

———————————————-

L. 220: Where do you see that?

Response: You can see this if you compare Figure 2 to differences between 1° and 0.25° simulations in Figure 1 and 2 e to Figure 3. We clarified this in the text. Dif-ference between DMV in ECMWF-LR (1°) (zero almost all the time) and ECMWF-MR (0.25°) is much larger than differences between different members. Same is true for the differences between 1 and 0.25° versions in the other NEMO-models compared to the spread across the ECMWF-ensemble.

———————————————

L. 231: Well , I se an around 50/50 split.

Response: All models with NEMO as ocean component where the ocean resolution

is increased (ECMWF-IFS, HadGEM3-GC31, CNRM-CM6-1, EC-Earth3P, see Table 2.1) show an increase in the DMV of the Labrador Sea. Note that MPI (not using NEMO) and CMCC (using NEMO) do not change the ocean resolution and AWI (as mentioned in the text; not using NEMO) show a reduction of the DMV with increased ocean resolution. We now specified the models and point to Table 2 .1 to clarify the statement.

————————————————

L. 259-261: I don't understand this sentence.

Response: The density profile in late winter is dominated by the convection itself; if we have convection, the density profile will strongly reflect if convection takes place or not. Thus, showing the late winter or March density profile does not really provide new information. However, if we analyse early winter/ late autumn density profiles (e.g. in November), we can see if models already before the period with strongest surface heat loss show large biases in the stratification, which can explain the strength of the convection in late winter, or if processes during the convection period itself explain potential biases in the convections. We reformulated the sentence to: "Naturally, the models with more frequent and deeper convection show a much weaker vertical stratification than the models that do not exhibit deep convection. We therefore analyse density profiles in the Labrador Sea in November that are not influenced by convection to explain why the mixed layer depth is overestimated in the following winter.

————————————————-

l. 274: What is the spinup period?

Response: It is 50 years using 1950-forcing. We added this to section 2.1.

————————————————-

L. 315: 'melting heat water fluxes'. What is that?
Response: We rephrased the sentence: This discrepancy could be due to the competing effects from global warming, which are represented differently in each model: on one hand, reduced sea ice extent enables a larger surface for deep convection, while on the other hand melt water and warmer surface water enhance the stratification and thus impede convection.

_______________________________-

L. 349 'high-resolution models'. In ocean or in atmosphere?

Response: In the ocean. We changed the sentence to: "Increased ocean resolution improves the representation of the observed SHF pattern."
* * *
L. 419: What does this sentence mean?

Response: "Thus, the observations show a stronger AMOC with a lower DMV compared to the models, indicating other shortcomings in the representation of processes that govern the AMOC in the models.": It means that observations can have a stronger AMOC despite a lower DMV compared to the models, or that the models with an AMOC, which is similar to the observed one, have a higher DMV than observed. The conclusion from this is that there have to be other shortcomings in the models, which might partly compensate the bad representation of the DMV. We reformulate the sentence to: "Thus, the linkage between mean values of AMOC and DMV in the models is not consistent with the observations. This indicates that other shortcomings in the representation of processes that govern the AMOC in the models exist. "

_______________________-

L. 443: add '. . . non eddy-permitting. . .'

Response: Some of the high-resolution models are eddy-permitting (maybe not eddy-resolving, although HadGEMGC31-3-HH might be called eddy-resolving), thus we

think it is misleading to add "non eddy permitting" here. However, we now clarified in the introduction that most of the models in HighResMIP and used in this study are not eddy-resolving.

——————————————

Table 3. Add columns with histroical trend – control trend.

Response: We added columns with the difference between historical and control trend to Table 3 and we show data in 1012 m3/s now for better readability.

——————————————

Fig. 2: Dont you yellow lines. They are really difficult to see.

Response: We replaced the yellow line with orange in Figure 2.

——————————————-

Add a Fig. 2 showing MLD for models + ARGO (analogous to Fig. 3)

Response: We are not entirely sure if we correctly understood what you suggest. However, as described in the text, the ARGO mixed layer data consist of two data sets: the climatological mean MLD in each grid point in March in the years 2000-2015; second, the maximum (mean over the two largest observed values in the period 2000-2015) MLD in each grid-point. In many grid-points only a few ARGO-profiles exist and in some no profiles at all. Thus, it is unfortunately not possible to show any DMV-time-series from the ARGO data and we only calculated the mean DMV over the 2000-2015 period.

——————————————-

Fig. 3: Labels like 5e+14 a really not nice to look at, please change them. Why is there only two lines in panel a). And please add a panel based on ARGO data.

Response: We now changed the y-axis in figure 3 and the other time-series figures

to avoid such kind of labels. With respect to ARGO data, please see response to the comment before. The DMV in ECMWF-LR is 0 almost all the time, thus it is hard to see the third line in panel a because it is almost the entire time placed on the 0-line. However, following suggestions from reviewer 1, we changed the colors in all time-series figures and now this line is blue and you should see it a bit better.
* * *
Fig. 4: Instead of having a separate panel (a) for ARGO, put the argo stratification in the model-panels.

Response: We added the ARGO profile to the model panels and deleted panel (a) as suggested.

____________________________-

Fig. 10: Colors for HadGEM3 and CNRM-CM6 are indistinguishable, please change. Also the figure is hard to understand. May be points referring to the same model in different resolution can be connected with thin lines. You must experiment, and improve the figure.

Response: We connected different resolutions of the same model with lines now to make the effect of resolution clearer, and we chose a different color for CNRM-CM6.

Please also note the supplement to this comment:
https://os.copernicus.org/preprints/os-2020-41/os-2020-41-AC2-supplement.pdf

**Supplement:**

**The effect of increasing the horizontal resolution on the Deep water formation in the North Atlantic Ocean in HighResMIP models**

Torben Koenigk[1,2], Ramon Fuentes-Franco[1], Virna L. Meccia[3], Oliver Gutjahr[4], Laura C. Jackson[5], Adrian L. New[6], Pablo Ortega[7], Christopher Roberts[8], Malcolm Roberts[5], Thomas Arsouze[7], Doroteaciro Iovino[9], Marie-Pierre Moine[10], Dmitry V. Sein[11]

[1] Rossby Centre, Swedish Meteorological and Hydrological Institute, Norrköping, Sweden
[2] Bolin Centre for Climate Research, Stockholm University, Sweden
[3] Istituto di Scienze dell'Atmosfera e del Clima (CNR-ISAC), Bologna, Italy
[4] Max Planck Institute for Meteorology, Hamburg, Germany
[5] Met Office, Exeter EX1 3PB, U.K
[6] The National Oceanography Centre Southampton, U.K.
[7] Barcelona Supercomputing Center – Centro Nacional de Supercomputación (BSC), Barcelona, Spain
[8] European Centre for Medium-Range Weather Forecasts (ECMWF), Reading, U.K
[9] Fondazione Centro Euro-Mediterraneo sui Cambiamenti Climatici (CMCC), Bologna, Italy
[10] CERFACS/CNRS, Toulouse, France
[11] Alfred Wegener Institute for Polar and Marine Research, Bremerhaven, Germany

*Correspondence to*: Torben Koenigk (torben.koenigk@smhi.se)

**Abstract.** Simulations from seven global coupled climate models performed at high and standard resolution as part of the High Resolution Model Intercomparison Project (HighResMIP) have been analyzed to study the impact of horizontal resolution in both ocean and atmosphere on deep ocean convection in the North Atlantic and to evaluate the robustness of the signal across models. The representation of convection varies strongly among models. Compared to observations from ARGO-floats, most models substantially overestimate deep water formation in the Labrador Sea. In the Greenland-Norwegian-Iceland (GIN) Seas, some models overestimate convection while others show too weak convection.

In four out of five  models with increased ocean resolution, higher ocean resolution leads to increased deep convection in the Labrador Sea. The effect of resolution on convection in the GIN Seas is less clear. Increasing the atmospheric resolution has a smaller effect on the deep convection than increasing the ocean resolution. Simulated convection in the Labrador Sea is largely governed by the release of heat from the ocean to the atmosphere. Higher resolution models show stronger surface heat fluxes than the standard resolution models in the convection areas, which promotes the stronger convection in the Labrador Sea. In the GIN Seas, the connection between high resolution and ocean heat release to the atmosphere is less robust and there is more variation across models in the relation between surface heat fluxes and convection. Simulated freshwater fluxes have less impact than surface heat fluxes on convection in both the GIN and Labrador Seas and this result is insensitive to model resolution. The mean strength of the Labrador Sea convection is important for the mean Atlantic Meridional Overturning Circulation (AMOC) and in around half of the models the variability of Labrador Sea convection is a significant contributor to the variability of the AMOC.

**1 Introduction**

Open-ocean deep convection is a rare phenomenon, occurring only at a few locations in the world's ocean. It provides a vertical link between properties of the surface ocean and the deep ocean. In the North Atlantic, the northward flowing warm water masses become denser because of large heat loss to the atmosphere and sink into the deep ocean. Deep convection ventilates the deep ocean with oxygen and plays an important role for the storage of carbon and heat.

The main convection sites in the North Atlantic are the Labrador and Irminger Seas as well as the Greenland-Iceland-Norwegian (GIN) Seas. Deep convection in the Labrador and Irminger Seas produces the deep water masses called Labrador Sea Water (LSW) (Clarke and Gascard, 1983), which together with the dense overflow waters coming through the Faroe Bank Channel and Denmark Strait (Dickson and Brown, 1994) form the North Atlantic Deep Water.

The deep convection in the Labrador Sea is mainly driven by wintertime buoyancy loss to the atmosphere (Latif et al., 2006; Frankignoul et al., 2009), which is strongly governed by the North Atlantic Oscillation (NAO). Also freshwater transports through Fram Strait contribute to the variability of deep convection in the Labrador Sea (Holland et al., 2001; Jungclaus et al., 2005; Koenigk et al., 2006). Labrador Sea convection varies strongly on interannual to decadal time scales (Yashayaev and Loder, 2016) and was rather shallow in the 1990s and 2000s, but recovered in recent years with mixing depths exceeding 2000m (Yashayaev and Loder, 2017). Convection in the GIN Seas occurred frequently down to the bottom until the 1980s. However, thereafter, no regularly occurring deep convection has been observed any more; between 1994 and 2002 mixing depths of 700-1600m have been monitored (Ronski and Budeus, 2005). Irminger Sea convection is generally weaker than in the Labrador and GIN Seas but in recent years mixing depth reached below 1000m depth (de Jong and de Steur, 2016; de Jong et al., 2018).

Variability and change of deep convection affects local and remote climate. Reduced convection in the Labrador Sea is linked to surface salinity, temperature and sea ice in Labrador Sea and Davis Strait (Deser et al. 2002) but has also remote effects on the atmospheric circulation (Koenigk et al. 2006). The deep convection has for long time been seen as the driving mechanism for the Atlantic Meridional Overturning Circulation (AMOC), which is fundamentally important for the climate in the North Atlantic and adjacent regions such as Western Europe and the Arctic (Manabe and Stouffer 1999; Mahajan et al., 2011, Koenigk et al., 2012; Jackson et al., 2015). However, more recently, the importance of deep convection for the AMOC has been questioned (Sayol et al., 2019). Kuhlbrodt et al. (2007) and Medhaug and Furevik (2011) identified wind-driven upwelling, gyre circulation, and wind and tidal vertical mixing as important processes sustaining the long-term strength of the AMOC, thus, a collapse of the deep convection would not necessarily lead to collapse of the AMOC (Gelderloos et al. 2012; Marotzke and Scott 1999). On the other hand, surface buoyancy forcing exerts a very strong control on exactly where and when the overturning occurs. Thus, even if mixing ultimately sets the strength of the global overturning, the deep water formation in the North Atlantic will play a key role in the strength of AMOC on decadal timescales.

Many studies have discussed the potential for a weakening AMOC as a response to global warming (Cheng et al., 2013; Swingedouw et al., 2007; Brodeau and Koenigk, 2016; Koenigk and Brodeau, 2017) and linked the weakening to a reduction of the deep water formation (Latif et al., 2006; Deshayes et al., 2007; Koenigk et al., 2007; Frankignoul et al., 2009; Langehaug et al. 2012). Recent studies indicate an ongoing weakening of the AMOC at 26.5°N (Smeed et al. 2014; Smeed et al. 2018; Thornalley et al., 2018; Caesar et al., 2018) although it is unclear if this observed weakening is caused by climate change or internal decadal variability (e.g. Jackson et al., 2016; Roberts et al., 2014; Robson et al. 2016). Observational evidence for a link between dense water formation in the Labrador Sea and AMOC is still missing (Lozier et al. 2017, Lozier et al. 2019) but short observation period and potential lags of several years between AMOC and convection (Brodeau and Koenigk, 2016; Roberts et al. 2013) make robust conclusions from observations difficult.

The deep convective process is temporally intermittent and spatially compact. This makes it difficult to observe this process, and state-of-the-art climate models, such as CMIP6, need to use parameterizations to represent convective processes (Fox-Kemper et al., 2019). Given the importance of the deep convection for climate and its future change, a reliable representation in climate models is highly important. However, Heuzé (2017) stated that "the majority of CMIP5 models convect too deeply, over too large an area, too often and too far south". While CMIP6 models seem to provide some improvements in the representation of bottom waters, more improvements are required (Heuzé 2020).

Going beyond the horizontal resolution of CMIP6 models, we analyse in this study whether high-resolution models from the CMIP6 High-Resolution Model Intercomparison Project (HighResMIP, Haarsma et al., 2016) improve the representation of deep convection in the subpolar North Atlantic. We use simulations from seven models participating in HighResMIP, which have been performed in the EU-H2020-project PRIMAVERA. Most of the high-resolution model versions of these seven models have ocean grid resolutions of around 10-25 km. Since the baroclinic Rossby radius of deformation is small in high latitudes (roughly 10 km in polar regions and 200 km in the tropics), mesoscale oceanic eddies are not resolved in the sub-polar convection regions in most of the HighResMIP models. We thus denote them as "eddy permitting" but not "eddy-resolving".

Recent studies found that the increased resolution in the HighResMIP models improved many aspects of the ocean including temperature and salinity biases (Gutjahr et al. 2019), the northward ocean heat transport (Grist et al., 2019), 
[revised manuscript text omitted]
., 2020). The set of HighResMIP experiments is divided into three tiers consisting of atmosphere-only and coupled runs and spanning the period 1950-2050 (details in Haarsma et al. 2016). Here, wWe use the Tier 2 historical coupled simulations from 1950-2014 and the 100-year control simulations (using constant 1950 forcing) from these seven models for our analysis. The control run used fixed 1950s forcing (GHG gases, including O3 and aerosol loading for a 1950s (~10 year mean) climatology). They will allow to evaluate potential model drifts in the historical simulations. Both control and historical runs are initialized from the end of 50 year spin-up simulations using 1950s-forcing. All models performed historical and controlthe simulations in at least two different resolutions. following the HighResMIP protocol. Changes in oceanic and atmospheric parameters are kept to a minimum between low and high resolution simulations, so that all changes can be directly attributed to the change in resolution.

[revised manuscript text omitted]

For calculation of the power spectrum, we used the method of Torrence and Compo (1998). Fourier transforms are calculated and red noise is used as background spectrum. To determine significance levels for the Fourier spectra the method of Torrence and Compo (1998) assumes that different realizations of the geophysical process will be randomly distributed

265 about this background spectrum, and the actual spectrum can be compared against this random distribution.

**3 Deep Convection in the North Atlantic**

This section analyzes first the MLD in March, the month with the strongest convection in both observations and models, in the North Atlantic in the different models and in ARGO. Then, we focus on the DMV in the models, its variability and potential trends in the historical simulations.

270 **3.1 Mixed layer depth**

Figure 1 shows the averaged March MLD from ARGO and from all historical model simulations. In the period 2000-2015, the ARGO data suggest average MLDs of about 1000 m in the Labrador and GIN sSeas. In the models, the MLD differs greatly and shows a strong dependence on the spatial resolution. While ECMWF-IFS-LR and EC-Earth3P show no or very shallow MLD in the main convection areas of the North Atlantic, many of the other simulations strongly

275 overestimate the MLD compared to ARGO. Some models simulate much too strong convection in the Labrador Sea but do not show any deep convection in the GIN Seas while other models overestimate the MLD in both seas. In contrast, in the Irminger Sea, the MLD is more consistent across models and agrees better with ARGO. Note that we here compare models' MLD averaged over 1950-2014 with ARGO-data from 2000-2015. As we will discuss later more in detail, some of the models show a weakening of the convection with time, particularly in the Labrador Sea.

280

Although the convection centre varies somewhat across models, we do not find any clear linkage to resolution. The models with particularly deep mixed layers show also the largest convection areas. Thus, compared to ARGO, they do not only overestimate the depth of mixed layer but also the area of deep convection.

285

The MLD deepens with increasing ocean resolution in all models, except for AWI-CM-1-0. However, the models showing deepening MLDs are not fully independent, because they share NEMO as the ocean component, whereas AWI-CM-1-0 has FESOM as ocean component. On the other hand, even the  models with NEMO3.6 as ocean component (compare HadGEM3-GC31, CNRM-CM6.1, CMCC-CM2 and EC-Earth3P) differ considerably. This discrepancy suggests that either the different atmospheric components or the choice of ocean parameters have a strong influence on the convection. In contrast, the MLD differs little when the atmosphere resolution is increased  (compare ECMWF-IFS-MR and ECMWF-IFS-HR, HadGEM3-GC31-MM with HadGEM3-GC31-HM, CCCM-CM2-HR4 with CCCM-CM2-VHR4). An exception is  MPI-ESM1-2, where an increased atmospheric resolution  reduces  MLD. This MLD reduction can  be linked to too weak wind forcing in MPI-ESM1-2-XR (Putrasahan et al., 2019).

To investigate the impact of natural variability on the mean March MLD in the historical period and to quantify  the potential contribution of natural variations to the differences in MLD with changing resolution, we use an ensemble of historical simulations with the ECMWF-IFS model. The MLD in the low resolution version ECMWF-IFS-LR is very shallow in all 6 ensemble members and there is no deep convection in the historical and control simulations (not shown). Thus, we concentrate in the following on the four members of ECMWF-IFS-HR, which all exhibit pronounced deep mixing, particularly in the Labrador Sea. These four ECMWF-IFS-HR members underlie  a considerable natural variability (Figure 2 a-d). The averaged March MLD (1950-2014) deviates in individual ensemble members up to about 200m from the ensemble mean MLD. Although, this is a considerable amount, given the relatively long averaging period, the MLD differences due to increased resolution from 1° to a 0.25° in the NEMO-models are larger (compare Figure 2 to differences between 1° and 0.25° simulations in Figure 1).  Although the DMV varies considerably between the ensemble member, the general amplitude, frequency and trends are similar.
Even though four members are not sufficient to capture the total natural variability, these results suggest that natural variability cannot explain the differences in MLD due to a change in spatial resolution.

**3.2 Deep Mixed Volume**

In the following, in order to consider the horizontal extension of convection patterns and discard shallow convection events that have limited impact on the oceanic circulation such as the AMOC, we will concentrate on the DMV index to investigate the deep convection in the Labrador and GIN Seas in more detail.

**3.2.1 Labrador Sea**

Figure 3 shows the DMV in the Labrador Sea in March in the historical model simulations. In agreement with Figure 1, increasing the ocean resolution from around 1° to 0.25°  generally leads to an increased DMV in all models using NEMO (ECMWF-IFS, HadGEM3-GC31, CNRM-CM6-1, EC-Earth3P, see Table 2.1), while the opposite is true for AWI-CM-1-0. creas in ocean resolution further to 1/12 ° in HadGEM3-GC31-HH does not further increase the DMV. The DMV varies strongly among models: ECMWF-IFS-LR does not show any deep convection events in the entire historical period, CNRM-CM6.1 and EC-Earth3P simulate only a few events with deep convection and AWI-CM-1-0-LR and CMCC-CM2 simulate strong deep convection every winter.

Table 2 compares the average DMV in the historical model simulations with that of ARGO in the period 2000-2015. We compare both the entire period 1950-2014 and the period 2000-2014 to ARGO. Generally, the simulated DMV in the Labrador Sea is smaller in 2000-2014 compared to the entire period. On the other hand, natural variability of the DMV is high and thus a 15-year period is probably too short for a comparison.  The DMV in EC-Earth3P-HR and CNRM-CM6.1 are closest to ARGO when considering the entire 1950-2014 period into account, however, they underestimate ARGO in 2000-2014. As discussed above, EC-Earth3P and ECMWF-IFS-LR show no or rather little deep convection in the Labrador Sea while the other simulations  overestimate the ARGO-based DMV with factors of four to almost 40 when taking the entire time period into account (3-25 in the period 2000-2014). The MPI-ESM1.2-XR overestimates ARGO in 1950-2014 but substantially underestimates it in 2000-2014. This indicates the difficulties when comparing short time periods with trends. Despite the uncertainties in the comparison to ARGO, it is clear that the models seem to have problems to realistically simulate the convection in the Labrador Sea. If deep convection occurs, the ocean is often mixed down to the bottom in the models, wherea deep convection rarely exceeds 2000m in the observations (Yashayaev and Loder, 2016; Yashayaev and Loder, 2017).

If we use a critical depth of $z_{crit}$=0 m instead of 1000 m in the Labrador Sea and thus consider the total mixed layer depth, the relative deviation of the DMV in the models from ARGO is reduced as expected (not shown). However, AWI-CM-1-0-LR and CMCC-CM2 still overestimate the DMV based on ARGO by a factor of three and two, respectively. On the other hand, ECMWF-IFS-LR simulates only 20% of the mixed volume compared to ARGO. The comparison between $z_{crit0}$ and $z_{crit1000}$ reveals also some non-linearites in the deep convection. While CNRM-CM6.1-HR has a nine times higher DMV ($z_{crit1000}$) compared to ARGO, it is only 16% higher for $z_{crit0}$, whereas the DMV ($z_{crit1000}$) for MPI-ESM1-2-XR is 4.6 times higher compared to ARGO but 14% smaller for $z_{crit0}$.

The interannual variability of the DMV is large in all models (Figure 3). Some of the simulations (EC-Earth3P, HadGEM3-GC31-LL, MPI-ESM1-2, CNRM-CM6.1 and ECMWF-IFS-MR) also indicate substantial variability at decadal or longer periods where phases with and without convection alternate. Such intermittent deep convection was also suggested based

on observations (Lazier et al., 2002; Yashayaev and Loder, 2016). . However, in most of the model simulations, deep convection occurs in almost every winter or not at all.

The strength of the deep convection in March is reflected in the vertical density distribution in the Labrador Sea. To calculate the density, we used the definition of density, following Millero and Poisson (1981). Naturally, the models with more frequent and deeper convection show a much weaker vertical stratification than the models that do not exhibit deep convection. We therefore analyse density profiles in the Labrador Sea in November that are not influenced by convection to explain why the mixed layer depth is overestimated in the following winter.  Figure 4 shows the vertical density stratification of the upper 600m in the Labrador Sea. All models show a near surface low density layer, mainly due to a combination of low surface salinity and relatively (compared to late winter) warm water near the surface in November. Generally, the models with lower ocean resolution show a stronger stratification in the upper ocean than models with higher resolution (except for AWI-CM-1-0). The two model simulations, which do not simulate any deep convection, ECMWF-IFS-LR and EC-Earth3P, show particularly strong upper ocean density gradients. Consequently, a large buoyancy flux would be needed during winter until deep convection could set in in these two models. MPI-ESM1-2 and AWI-CM-1-0 show a more stratified upper ocean in November with increased atmospheric resolution, which  agrees with a weaker convection in their higher resolution versions. The density profiles of the high ocean resolution models agree  well with the observed one from ARGO, although the near surface low density layer is too shallow in most of these models. This shallower surface layer requires less heat to be eroded, which might explain the overestimation of the deep convection in late winter in these models (compare Figures 1 and 3). However, this is probably not the only reason as will be further discussed in section 4.

Twelve of 19 simulations indicate a significantly negative trend of the DMV in the historical period (Figure 3, Table 3). To investigate whether this trend is  due to external forcing or due to model drift , we compared the DMV in the historical simulations with that from the 100-year 1950-control simulations (Figure 5 and Table 3). Most of the control simulations do not show any significant trend and in 9 out of 17 historical simulations, the DMV trends in the historical simulations are significantly more negative compared to the first 65 years of the control simulations. This indicate external forcing as major cause for the DMV reduction in the historical simulations. Furthermore, the negative trends become larger with higher ocean resolution.

A reduction of DMV in the historical period would be in line with some recent studies by Caesar et al. (2018), Thornalley et al. (2018) and Brodeau and Koenigk (2016).

To investigate the predominant variability frequency of the DMV, we calculated the power spectrum of detrended DMV time series of the Labrador Sea from the 100-year long control simulations

385  (Figure 6).  The dominant time scale varies across simulations and models. Many of the models  (ECMWF-MR, HadGEM3-GC31-MM, HadGEM3-GC31-HM, , EC-Earth3P-HR,  MPI-ESM1-2-HR) show a  peak in the spectrum at around 10 years. In ECMWFS-IFS-
390 HR, HadGEM3-GC31-HH and MPI-ESM1-2-XR with further increased resolution in either ocean or atmosphere, the main peak in the spectrum seems to shift towards somewhat longer time periods. A shorter frequency of 7 years is found for CMCC-CM2-HR  and AWI-CM-1-0-HR . The lower ocean-resolution models, AWI-CM-1-0-LR, HadGEM3-GC31-LL and ECMWF-IFS-LR (but very weak DMV), show less dominate peaks than their higher resolution versions. CNRM-CM6.1 shows peaks at similar periods as CNRM-CM6.1-HR
395 but the amplitudes differ.

**3.2.2 GIN Seas**

The DMV in the GIN Seas shows also a large spread across models (Figure 7, Table 2). As for the Labrador Sea, AWI-CM-1-0-LR and the two CMCC-CM2 simulations show the strongest deep convection while ECMWF-IFS and
400 EC-Earth3P simulate rather weak convection (note the different order of magnitude in the vertical scales of Figure 7).  MPI-ESM1-2-HR and CNRM-CM6.1 show the smallest deviations  from the ARGO observations. HadGEM3-GC31-HM overestimates MLD by 70% when comparing the full historical period to ARGO (400% when comparing only 2000-2014) and MPI-ESM1-2-XR underestimates MLD by 50%. EC-Earth3P and ECMWF-IFS strongly underestimate deep water formation
405 in the GIN Seas while the DMV is  strongly overestimated by AWI-CM-1-0 and CMCC-CM2.
The resolution dependency of  DMV in the GIN Seas differs substantially from the Labrador Sea. There is no robust relation between MLD and resolution. All models with NEMO show either no or shallower MLD with increased resolution. ~~The NEMO models do not show any deepening of convection with increased ocean resolution. In contrast, the high resolution versions of ECMWF-IFS, EC-Earth3P and CNRM-CM6.1 (after 1980) show very shallow or no deep
410 convection. Moving from HadGEM3-GC31-LL to HadGEM3-GC31-MM leads to substantially smaller DMV. However, when increasing the atmospheric resolution from HadGEM3-GC31-MM to HadGEM3-GC31-HM, DMV increases, and increases further in HadGEM3-GC31-HH, where both ocean and atmosphere resolution is increased. Thus, unlike for the Labrador Sea, there is no robust relation between the resolution in ocean or atmosphere and DMV in the Greenland Sea in the global models with NEMO as ocean component. The two other models,~~ MPI-ESM1-2 and AWI-CM-1-0 show, as for
415 the Labrador Sea, a reduction of the DMV with increased resolution.

The trends of DMV in the GIN Seas  do not agree across models in the historical period (Figure 7, Table 3). While HadGEM3-GC31-MM, MPI-ESM1-2-XR and the CMCC-CM2 simulations show a significantly negative trend, HadGEM3-GC31-HM and EC-Earth3P show positive trends. This discrepancy could be due to the competing effects from global warming, which are represented differently in each model: on one hand,  reduced sea ice extent  enabl a larger surface for deep convection, while on the other hand  melt water  and warm surface water enhance the stratification and thus impede convection. The high DMV in CNRM-CM6.1-HR before 1970, and the decrease thereafter occurs similarly in the first decades of the 1950-control-simulation indicating that this trend is caused by a model balance adjustment and not due to external forcings (Figure 8, Table. 3). Similarly, the strong negative trends in CMCC-CM2 can partly be explained by similar drifts in the control simulations, although the reduction in the historical runs is significantly larger than in the control runs.

Some of the historical and control simulations show strong decadal or longer-term variations (Figure 9). HadGEM3-GC31-HM, MPI-ESM1-2-HR, AWI-CM-1-0-HR and EC-Earth3P-HR show dominant variability at periods of around 20-25 years; ECMWF-IFS-MR, HadGEM3-GC31-HH, CMCC-CM2 and CNRM-CM6-1 simulations at time scales of 10-15 years. In addition, some of the simulations indicate variability at time-scales below 8 years. Overall, there is no clear dependence of the variability on the resolution.

Figure 10 summarizes the results from sections 3.2.1 and 3.2.2. on the resolution dependency of the deep convection across the models in Labrador and GIN Seas. Each single model shows a clear dependence of the DMV in the Labrador Sea on the oceanic resolution. The differences across models are large, and as discussed before, even models using the same version of the NEMO as ocean model exhibit a wide range of solutions. Models with coarse resolution (~100 km) produce no or only shallow convection. Models with a resolution of 50km and higher in the ocean, however, overestimate deep convection compared to ARGO.

Increasing the atmosphere resolution has a minor impact on the DMV in the Labrador Sea, except for MPI-ESM1.2 and AWI-CM-1-0, where DMV is reduced with increased resolution.

The resolution dependency of the DMV in the GIN Seas in single models is smaller than in the Labrador Sea. However, all the models, except for CNRM-CM6-1, show a decreased DMV when increasing the resolution to around 0.25°. The response to increased atmosphere resolution is not robust across models.

**4 The impact of heat and freshwater fluxes on the deep convection in the North Atlantic**

Deep convection depends strongly on the buoyancy of the ocean surface layer in the convection regions - the heat loss to the atmosphere and the influx of fresh water into the convection regions.

**4.1 Surface heat fluxes**

Brodeau and Koenigk (2016) showed that the turbulent surface heat flux (SHF) is the main driver for interannual variability in the DMV. Thus, in the following, we will mainly focus on the SHF.

Figure 11 shows the winter (January, February, March) SHF in each of the model simulations. The WHOI-OAFlux data show the largest SHF (from the ocean to the atmosphere) up to more than 200 W/m$^2$ from the ice edge in the Labrador Sea extending to the southern part of the subpolar gyre, south of Iceland and along the southeast coast of Greenland, and in the northern Norwegian-Greenland Seas and Barents Sea. The large-scale features of this pattern are reproduced by most of the models. ECMWF-IFS-LR and to a lesser degree EC-Earth3P, both simulating too weak convection, strongly underestimate the SHF in the Labrador Sea. Increased ocean resolution improves the representation of the observed SHF pattern. In particularly,  the extension of high SHF from the Labrador Sea into the southwestern branch of the sub-polar gyre and the high SHF in the northern Greenland and Norwegian Seas is better simulated. A number of models ( both CMCC-CM2 versions, HadGEM3-GC31-HH and CNRM-CM6.1) overestimate the SHF in the sub-polar gyre. In addition, the SHF west and northwest of Scotland is too high in most of the models.

In the Labrador Sea, all high-resolution models with NEMO as the ocean component simulate increased SHF (averaged over the same box as used for calculation of the DMV) compared to their lower-resolution counterparts (Table 2). In contrast, MPI-ESM1-2 shows reduced SHF with increased atmospheric resolution in line with the reduced convection.  In all models, the interannual variations of winter SHF  is significantly positively correlated with the DMV in March. hus, large ocean heat losses in the winter are linked to strong DMVs in the following March, indicating that large upward surface heat fluxes lead the DMV. The correlation coefficient varies from 0.48 in CNRM-CM6.1 to slightly above 0.7 in ECMWF-IFS-MR, EC-Earth3P and CMCC-CM2-HR4. The relation between SHF and DMV is neither resolution nor model dependent.

The winter SHF in the Labrador Sea itself is governed by the atmospheric circulation (not shown). In all model simulations northerly to northwesterly winds, which advect cold air from the Arctic sea ice towards the Labrador Sea, lead to strong surface heat fluxes, which overcome the stratification of the ocean (Ortega et al. 2011), and increased convection. These north-to-northwesterly winds are further linked to a large scale atmospheric circulation pattern, which is similar to the positive phase of the North Atlantic Oscillation (NAO. ) defined as the leading EOF of geopotential height on the 500 hPa

pressure surface over the European/Atlantic sector (80°W-40°E, 20°-90°N)). The spatial imprint of the NAO-index on the 500-hpa geopotential height is shown in Figure 12. All models reproduce the NAO-pattern of the ERA5-reanalysis data well. However, the position of the negative pole over Iceland-Greenland and the extension of the positive pole towards Eurasia vary slightly among models. The NAO-index itself, which we define here as the difference of the normalized winter sea level pressure anomalies over the Azores and Iceland, is significantly positively correlated with the DMV$z_{crit1000}$ in the Labrador Sea in all simulations except for the low resolution simulations with EC-Earth3P and ECMWF-IFS. These are the simulations with no or only little deep convection and which have a strongly stratified ocean. The other model simulations show correlations between 0.38 (HadGEM-GC31-LL) and 0.67 (HadGEM-GC31-HM and CMCC-CM2-HR4). The NAO is not only important for interannual variations of the DMV but also on the decadal scale. Correlations of 10-year running means of NAO and DMV reach between 0.3 and 0.57. A spectral analysis of the NAO resembles most of the peaks in the spectrum of the DMV (not shown) although the NAO shows relatively more energy at shorter time scales compared to the DMV in the Labrador Sea.

As in the Labrador Sea, northerly winds are the main cause for large oceanic surface heat loss to the atmosphere in the Greenland Sea. The northerly winds are connected to low pressure anomalies over northern Scandinavia and the Barents Sea. As in the Labrador Sea, tThe DMV in the Greenland SeaGIN Seas is correlated to the SHF as well and ocean heat loss is linked to a large DMV. Northerly winds are the main cause for large oceanic surface heat loss to the atmosphere in the GIN Seas. However, here we find a strongerer model dependency of the correlation than in the Labrador Seais relation. The correlation is weak to moderate in HadGEM3-GC31 (r=0.22 for LL; r=0.5 for HH) and in CNRM-CM6.1 (r=0.35; r=0.5 for HH) but high correlation is found for ECWMF-IFS (r=0.64 for HR; r=0.85 in LR) and EC-Earth3P (r=0.61; r=0.69 for HR). As for the Labrador Sea, the relation between SHF and DMV shows no clear resolution dependency.

**4.2 Freshwater and sea ice exports**

A number of studies have previously discussed the effect of Arctic freshwater export, especially through Fram Strait, as a potential source of variability of the deep water convection in the Labrador Sea (Holland et al., 2001; Jungclaus et al., 2005; Koenigk et al., 2006). Here, we analyze the correlations between freshwater transports across different sections (Fram Strait, Denmark Strait, northern Baffin Bay) and deep convection in the Labrador Sea as well as betweennd transports through Fram Strait and deep convection in the Labrador Sea and in the Greenland SeaGIN Seas convection(transports through Fram Strait) in the historical simulations of the models.

To calculate the liquid freshwater transport, we used the model grid lines on the native grids of the models that are closest to the geographical landmarks that define Fram Strait (across 78°N), Northern Baffin Bay (78°N) and Denmark Strait (66°N). The freshwater has been defined as the amount of zero-salinity water required to reach the observed salinity of a seawater sample starting from a reference salinity. Specifically, liquid freshwater transport (fwt, in $m^3/s$) is estimated as

$$fwt = \int\limits_{p1}^{p2} \int\limits_{D}^{\eta} \left(\frac{S - Sref}{Sref}\right) dz dx$$

for salinity S (in practical salinity units). As reference salinity Sref we used 34.80 psu for all models. The integration along z is performed from the bottom at depth D to the sea surface at height η (in this case η=0). p1 and p2 are the landmarks and the integration was done considering dx as the length (or depth for dz) between every grid point.

The solid freshwater transport is calculated from the sea ice transports across the sections assuming a constant ice salinity of 5 psu.

Table 4 shows the freshwater exports out of the Arctic into the North Atlantic through Fram Strait and Baffin Bay and through the Denmark Strait. Although differences between models are large, the exports through Fram Strait are generally larger than through Baffin Bay. The total freshwater exports through Fram Strait (liquid + solid export) varies between around 80000 m$^3$/s in the two CNRM-CM6.1 models and 160000 m$^3$/s in ECMWF-IFS-LR and HadGEM3-GC31-MM. The distribution between liquid and solid export through Fram Strait differs strongly across the simulations. While in HadGEM3-GC31-LL and all ECMWF-IFS and EC-Earth3P simulations most of the freshwater leaves the Arctic in the form of sea ice, liquid and solid parts are of similar size in the other models. In CNRM-CM6.1-HR, the liquid part is even larger than the solid part. The amount of freshwater that passes the Denmark Strait is reduced compared to Fram Strait in all models except for ECMWF-IFS-LR and MPI-ESM1.2-XR, and the liquid part is dominating. Large parts of the ice melt in the East Greenland Current on its way from Fram Strait to Denmark Strait.

The low resolution versions of ECMWF-IFS, HadGEM3-GC31, CNRM-CM6.1 show a larger fraction of solid exports through Fram Strait and larger liquid transports through Baffin Bay (EC-Earth3P as well for Baffin Bay) compared to their higher resolution counter parts. The sum of freshwater exports through Fram Strait and Baffin Bay differs more between the models than between different versions of single models. Despite the large differences in mean Arctic freshwater exports into the North Atlantic, there is no clear linkage to the mean DMV in GIN and Labrador Seas. Only for ECMWF-IFS-LR, we speculate that the very large freshwater fluxes, particularly in form of sea ice, through Denmark Strait contribute to the low surface density in the Labrador Sea (compare Figure 4) and consequently to suppress any deep convection activities in the Labrador Sea.

In order to further investigate if the variability of freshwater exports affects the deep convection in GIN and Labrador Seas, we correlate the solid and liquid transports across all sections with the DMV.  In all model simulations, the annual mean southward transport of both liquid and solid freshwater across Fram Strait and the liquid transport across Denmark Strait are weakly negatively correlated with the deep convection in the Labrador Sea in March ( range

Formaterat: Teckensnitt:(Standard +Rubriker (Times New Roman)

Formaterat: Teckensnitt:(Standard +Rubriker (Times New Roman)

Formaterat: Teckensnitt:(Standard +Rubriker (Times New Roman)

Formaterat: Upphöjd

Formaterat: Upphöjd

[revised manuscript text omitted]

Docquier, D., Fuentes-Franco, R., Koenigk, T., and Fichefet, T.: Sea ice - ocean interactions in the Barents Sea modeled at different resolutions, Front. Earth Sci. 8:172, doi: 10.3389/feart.2020.00172.

Eden, C., and Willebrand, J.: Mechanism of interannual to decadal variability of the North Atlantic circulation, J. Clim., 14(10), 2266–2280, doi:10.1175/1520-0442(2001)014, 2001.

Fox-Kemper, B., Adcroft, A., Böning, C. W., Chassignet, E. P., Curchitser, E., Danabasoglu, G. Eden, C. and co-authors: Challenges and Prospects in Ocean Circulation Models, Front. Mar. Sci., 26, https://doi.org/10.3389/fmars.2019.00065, 2019.

[revised manuscript text omitted]

**Formaterad tabell**

| | | | | | | |
|---|---|---|---|---|---|---|
| **CMCC-CM2-HR4** | 24.4 (21.0) | 1.22 | 13.0 (7.8) | 1.22 | 0.72 | 0.52 |
| **CMCC-CM2-VHR4** | 24.8 (25.6) | 1.34 | 15.0 (13.1) | 1.14 | 0.59 | 0.58 |
| **CNRM-CM6.1** | 1.09 (0.13) | 1.15 | 1.10 (1.30) | 1.07 | 0.53 | 0.45 |
| **CNRM-CM6.1-HR** | 9.3 (3.2) | 1.18 | 2.47 (0.04) | 1.35 | 0.48 | 0.35 |
| **MPI-ESM1-2-HR** | 10.6 (7.1) | 1.14 | 1.2 (0.69) | 1.25 | 0.61 | 0.44 |
| **MPI-ESM1-2-XR** | 4.6 (0.30) | 0.98 | 0.6 (0.47) | 1.16 | 0.64 | 0.64 |
| **AWI-C-1-0-LR** | 39.5 (24.9) | no | 20.9 (18.8) | no | no | no |
| **AWI-CM-1-0-HR** | 12.8 (10.1) | data | 6.1 (4.8) | data | data | data |
| **EC-Earth3P** | 0.26 (0.05) | 0.63 | 0.24 (0.38) | 1.02 | 0.72 | 0.69 |
| **EC-Earth3P-HR** | 0.95 (0.21) | 1.07 | 0. (0) | 0.79 | 0.50 | 0.61 |

**Table 2: Observed and modeled DMV and SHF in the Labrador and GIN sSeas, the ratio between model and observed values and  correlations between SHF and DMV. Row 2: DMV and SHF in observations, shown are absolute values. Rows 3-9: Ratio of modeled and observed DMV- and SHF (Model values divided through observational values: DMV$_{model}$/ DMV$_{obs}$ and SHF$_{model}$/ SHF$_{obs}$). For ECMWF-IFS, ensemble means are shown. For the DMV (columns 2, 4), the first number compares the mean of the entire historical simulation (1950-2014) to ARGO. The number in brackets only the years 2000-2014. Columns 6, 7: Correlation between winter SHF and March DMV in the respective boxes of the Labrador and GIN Seas. For the correlations $z_{crit0}$ has been used to avoid complications with periods without any deep convection; the correlations based on $z_{crit1000}$ and $z_{crit700}$ in LAB and GIN Seas are generally similar for models with deep water formation in every winter but much lower in the models with no or very few deep convection events (ECMWF-LR, EC-Earth3P).**

| Model | Trend/year historical 1950-2014 DMV-Lab | Trend/year control-1950 year 1-65 DMV-Lab | Trend-difference: hist – control DMV-Lab | Trend/year historical 1950-2014 DMV-GIN | Trend/year control-1950 year 1-65 DMV-GIN | Trend-difference: hist – control DMV -GIN |
|---|---|---|---|---|---|---|
| **ECMWF-IFS-LR** **ECMWF-IFS-MR** **ECMWF-IFS-HR** | 0.003 -0.5 *-3.93e+12* *-3.40e+12* | -0.01 -0.43 | -0.02 *-3.39* *-2.97* | 0.01 0 0.002 | 0.05 0.005 0.01 | 0.06 -0.005 -0.012 |

| | | | | 09 | | |
|---|---|---|---|---|---|---|
| **HadGEM3-GC31-LL** | 0.41.09e+11 | 1.58e+12 | -1.17 | 6.62e+12 | 5.59e+12 | 1.03 |
| **HadGEM3-GC31-MM** | *-4.43e+12* | -1.11e+12 | *-3.32* | *-1.49e+12* | 0.9.73e+11 | -2.46 |
| **HadGEM3-GC31-HM** | *-5.13.3e+12* | -3.02e+12 | *-2.11* | *4.40e+12* | -0.33.28e+11 | *4.73* |
| **HadGEM3-GC31-HH** | *-6.66e+12* | -2.28e+12 | *-4.38* | 0.3.10e+11 | -0.95.49e+11 | 1.25 |
| **MPI-ESM1-2-HR** | -1.41e+12 | -0.62.15e+11 | -0.79 | - | *-0.8.553e+11* | -0.02 |
| **MPI-ESM1-2-XR** | *-7.94e+12* | -0.06.02e+10 | *-7.88* | 0.82.17e+11 *-0.41.06e+11* | 0.23.32e+11 | -0.64 |
| **CMCC-CM2-HR4** | *-5.19e+12* | -1.08e+12 | *-4.11* | *-9.31e+12* | -3.97e+12 | *-5.34* |
| **CMCC-CM2-VHR4** | -1.42e+12 | -3.27e+12 | 1.85 | *-1.2.0e+13* | -2.69e+12 | *-14.69* |
| **CNRM-CM6.1** | *-0.54.38e+11* | 0.1.62e+11 | *-0.70* | 0.8.61e+11 | 0.4.51e+11 | 0.41 |
| **CNRM-CM6.1-HR** | *-6.91e+12* | -0.31.06e+11 | *-6.60* | *-1.0.1e+13* | *-1.0.5e+13* | 0.4 |
| **AWI-CM-1-0-LR** | *-1.5.0e+13* | *-1.6.88e+13* | 1.80 | -1.97e+12 | 0.6.80e+11 | -2.65 |
| **AWI-CM-1-0-HR** | *-0.96.56e+11* | -4.98e+12 | 4.02 | 1.21e+12 | *-2.50e+12* | 3.71 |
| **EC-Earth3P** | 0.17.69e+11 | 0 | 0.17 | *0.43.25e+11* | *0.033.05e+10* | *0.40* |
| **EC-Earth3P-HR** | *-0.39.87e+11* | 0.9.22e+11 | *-1.31* | - 0.001.1.39e+09 | 0.003.04e+09 | -0.002 |

**Table 3: Trends in the DMV in the Labrador and GIN Seas in the historical simulations and in the first 65 years of the 1950-control simulations (in 10$^{12}$ m$^3$/year). Trends that are significantly different from 0 at the 95%-confidence level are shown in italic, trends significantly different to the control-runs are bold, and trends**

920 **significantly different to both 0 and the control-run are italic and bold.**

| Freshwater fluxes in m$^3$/s | Fram Strait liquid | Fram Strait solid | Denmark Strait liquid | Denmark Strait solid | North Baffin Bay liquid | North Baffin Bay solid | Sum Fram Strait + Baffin Bay |
|---|---|---|---|---|---|---|---|
| **ECMWF-IFS-LR** | 12694 | 154000 | 60517 | 92093 | 21688 | 7181 | 195563 |
| **ECMWF-IFS-MR** | 58595 | 85108 | 69694 | 12635 | 17597 | 7031 | 168331 |
| **ECMWF-IFS-HR** | 47578 | 107470 | 68940 | 22533 | 14479 | 7367 | 176894 |
| **HadGEM3-GC31-LL** | 30394 | 85642 | 25304 | 28543 | 55002 | 7043 | 178081 |
| **HadGEM3-GC31-MM** | 81239 | 82197 | 69489 | 21907 | 19993 | 6614 | 190043 |
| **HadGEM3-GC31-HM** | 72294 | 73470 | 61834 | 13905 | 18728 | 10309 | 174801 |
| **HadGEM3-GC31-HH** | no data | 56979 | no data | 12845 | no data | 17117 | no data |
| **CMCC-CM2-HR4** | 70915 | no data | 60545 | no data | 14854 | no data | no data |
| **CMCC-CM2-VHR4** | 61063 | no data | 13785 | no data | 6239 | no data | no data |
| **CNRM-CM6.1** | 39699 | 43906 | 33344 | 8689 | 35078 | 3902 | 122585 |
| **CNRM-CM6.1-HR** | 52321 | 29243 | 41327 | 15409 | 22297 | 8540 | 112401 |
| **MPI-ESM1-2-HR** | 66843 | 54540 | 92680 | 8453 | 1276 | 2232 | 124891 |
| **MPI-ESM1-2-XR** | 54834 | 62348 | 112003 | 12408 | 2231 | 3145 | 122558 |
| **EC-Earth3P** | 26843 | 96230 | 30439 | 28277 | 25317 | 5813 | 154203 |

| EC-Earth3P-HR | 22096 | 105890 | 54655 | 27370 | 1776 | 3488 | 133250 |
|---|---|---|---|---|---|---|---|

**Table 4:** Liquid and solid freshwater fluxes through Fram Strait, Denmark Strait and northern Baffin Bay in the historical simulations averaged over 1950-2014. Positive values mean freshwater exports out of the Arctic. The last column shows the sum of liquid and solid exports through Fram Strait and Baffin Bay. No data were available to calculate transports in the AWI-CM-1.0 model, liquid freshwater transports in HadGEM3-GC3-HH and sea ice transports in the CMCC-simulations.

[Figure]

**Figure 1: Mixed layer depth in March in the ARGO-data, averaged over 2000-2015 (a) and in the historical low and high-resolution model simulations, averaged over 1950-2014 (b-r).**

940

945

950

[Figure]

March MLD, 1950−2014, ECMWF−IFS−HR. Deviation from ensemble mean
a) Member 1   b) Member 2   c) Member 3   d) Member 4

−300 −200 −150 −100 −60 −20   20  60  100  150  200  300
m

e) DMV in the Labrador Sea in ECMWF-IFS-HR

[Figure]

[Figure]

**Figure 2: a-d) Deviation of mixed layer depth in March in the ensemble members of ECMWF-IFS-HR from the ensemble mean of the four ECMWF-IFS-HR simulations for the time period 1950-2014. e) DMV in the Labrador Sea (in $10^{15}$ m$^3$) in the ensemble mean and single ensemble members of ECMWF-IFS-HR.**

**Formaterat:** Upphöjd

**Formaterat:** Upphöjd

[Figure]

[Figure]

970

**Figure 3: Deep Mixed Volume (DMV) in $10^{15}$ m$^3$ using a critical depth of 1000 m in the Labrador Sea in March between 1950-2014. Note the different y-axis between models. For ECMWF-IFS, only member 1 is shown for better visual comparison of the variability across resolutions.**

975

[Figure]

[Figure]

**Figure 4: Density (in kg/m³) in the upper 600 m averaged over the Labrador Sea in November.**

[Figure]

[Figure]

**Figure 5: As Figure 3 but for the 100-year control simulation.**

990

995

[Figure]

[Figure]

**Figure 6:** Power spectrum of detrended and normalized March DMV time series of the 100-year control simulation in the Labrador Sea. The dashed red line shows the 95% significance level. The y-axis uses a different scale depending on the model. Note, that no deep convection occurred in the 100-year period in EC-Earth3P.

**Formaterat:** Teckensnitt:Inte Fet

[Figure]

[Figure]

**Figure 7: As Figure 3 but for the GIN Seas and a critical depth of 700 m.**

1020

1025

[Figure]

[Figure]

**Figure 8: As Figure 7 but for the 100-year 1950-control simulations.**

1035

1040

[Figure]

[Figure]

**Figure 9: Same as Figure 6 but for DMV in the GIN Seas.**

[Figure]

[Figure]

**Figure 10: DMV (average over 1950-2014 in 1012 m3) in Labrador Sea (left) and GIN-Seas (right) in March in dependence on the oceanic (top) and atmospheric (bottom) resolution. Thin lines connect model versions with different resolution of the same model.**

**Formaterat:** Upphöjd

[Figure]

[Figure]

**Figure 11: Turbulent surface heat flux (January, February, March average) in 1950-2014 in the WHOI-OAFlux data and in the model simulations. Positive values mean flux from the ocean to the atmosphere.**

[Figure]

 **Figure 12:** Pearson correlation between geopotential height at 500 hPa and North Atlantic Oscilation (NAO) index during winter (JFM mean) for ERA5 and the models. The periods used were 1979-2019 for ERA5 and 1950-2014 for the models.

1085

[Figure]

1090

Figure 1 2: Regression between annual mean freshwater transport through the Denmark Strait and mixed layer depth in the following March. a) HadGEM-CG31-LL and b) EC-Earth3P-HR. Data have been detrended before calculating the regression. These two simulations show the largest correlation between Denmark Strait freshwater transport and DMV in the Labrador Sea (-0.4 and -0.35 for HadGEM-GC31-LL and EC-Earth3P-HR, respectively).

1095

**AMOC and DMV in the Labrador Sea (control-1950)**

[Figure]

[Figure]

[Figure]

Figure 143: Crosscorrelation between the DMV using a critical depth of 1000 m in the Labrador Sea in March and the AMOC index for the 100-year control simulation. Both timeseries were detrended and filtered with a 10 years low-pass filter. Area enclosed by dotted lines represents the 95% confidence calculated as 2/sqrt(N), where N is the number of independent data based on the time that takes autocorrelation to fall below 1/e. Positive lags mean AMOC leads DMV, negative lags mean DMV leads AMOC. The low resolution version of EC-Earth3P does not produce any deep convection events in the control simulation.

**AMOC and DMV in the Greenland Sea (control-1950)**

[Figure]

[Figure]

**Formaterat:** Svenska (Sverige)

[Figure]

**Figure 14: As Figure 12 but for the GIN Seas and a critical depth of 700 m.**